# Macrophage-fibroblast JAK/STAT dependent crosstalk promotes liver metastatic outgrowth in pancreatic cancer

Meirion Raymant [1], Yuliana Astuti[1], Laura Alvaro-Espinosa[2], Daniel Green[1], Valeria Quaranta[1], Gaia Bellomo[1], Mark Glenn [1], Vatshala Chandran-Gorner[1], Daniel H. Palmer[1], Christopher Halloran[1], Paula Ghaneh[1], Neil C. Henderson [3,4], Jennifer P. Morton [5], Manuel Valiente [2], Ainhoa Mielgo [1] & Michael C. Schmid [1]✉

Pancreatic ductal adenocarcinoma (PDAC) is a highly metastatic disease for which better therapies are urgently needed. Fibroblasts and macrophages are heterogeneous cell populations able to enhance metastasis, but the role of a macrophage-fibroblast crosstalk in regulating their pro-metastatic functions remains poorly understood. Here we deconvolve how macrophages regulate metastasis-associated fibroblast (MAF) heterogeneity in the liver. We identify three functionally distinct MAF populations, among which the generation of pro-metastatic and immunoregulatory myofibroblastic-MAFs (myMAFs) critically depends on macrophages. Mechanistically, myMAFs are induced through a STAT3-dependent mechanism driven by macrophage-derived progranulin and cancer cell-secreted leukaemia inhibitory factor (LIF). In a reciprocal manner, myMAF secreted osteopontin promotes an immunosuppressive macrophage phenotype resulting in the inhibition of cytotoxic T cell functions. Pharmacological blockade of STAT3 or myMAF-specific genetic depletion of STAT3 restores an anti-tumour immune response and reduces metastases. Our findings provide molecular insights into the complex macrophage–fibroblast interactions in tumours and reveal potential targets to inhibit PDAC liver metastasis.

Pancreatic ductal adenocarcinoma (PDAC) is an aggressive and lethal disease with a 5-year survival rate of 38% for the minority of patients (20%) diagnosed at a local stage, eligible for curative surgical resection with adjuvant chemotherapy[1]. However, the majority of PDAC patients are diagnosed with distant hepatic metastatic disease and the lack of effective treatments inhibiting metastasis results in a devastating 5-year survival rate of only 3% for these patients[1,2]. A better

understanding of the molecular pathology of metastatic PDAC is essential for the development of novel therapeutic strategies targeting this lethal disease[3]. Colonisation of a distant organ is a rate limiting step for metastatic progression and is a process critically facilitated by non-cancerous cells that are both recruited and resident to the distant metastatic organ. These non-cancerous cells establish an inflammatory-fibrotic tumour microenvironment (TME) that supports

[1]Department of Molecular and Clinical Cancer Medicine, University of Liverpool, Ashton Street, Liverpool L69 3GE, UK. [2]Brain Metastasis Group, Spanish National Cancer Research Centre (CNIO), Madrid, Spain. [3]Centre for Inflammation Research, The Queen's Medical Research Institute, Edinburgh BioQuarter, University of Edinburgh, Edinburgh EH16 4TJ, UK. [4]MRC Human Genetics Unit, Institute of Genetics and Cancer, University of Edinburgh, Edinburgh, UK. [5]Cancer Research-UK Scotland Institute and School of Cancer Sciences, University of Glasgow, Switchback Road, Glasgow G61 1BD, UK. ✉e-mail: mschmid@liv.ac.uk

the survival and growth of disseminated cancer cells in the distant organ[3–6]. The emergence of single cell RNA sequencing (scRNAseq) technology has revealed the existence of transcriptionally diverse cancer-associated fibroblast (CAF) populations among many solid tumours[7–13]. CAFs with an inflammatory gene expression signature have been defined as inflammatory CAFs (iCAF), while CAFs with an extracellular matrix rich gene expression signature resembling smooth muscle actin (αSMA) high myofibroblasts were named myofibroblastic CAFs (myCAF)[7,8]. In primary PDAC tumours, besides iCAF and myCAF, an additional antigen presenting CAF (apCAF) subpopulation has been described[12–14]. In murine PDAC liver metastasis, three distinct metastasis-associated fibroblast (MAF) populations have been described, resembling transcriptional phenotypes of iCAF, myCAF and a mesothelial-like CAF (mesCAF) subtype[15]. However, there is emerging evidence that CAF subtypes with similar transcriptional phenotypes may have functionally distinct roles in different organs. In PDAC, the depletion of αSMA+ myofibroblastic CAFs in the pancreas accelerates disease progression, whereas depletion of αSMA+ myofibroblastic MAFs in the liver reduces metastatic disease progression, highlighting organ-specific functional discrepancies among myofibroblasts[15,16].

CAFs and MAFs arise from distinct pools of tissue-resident mesenchymal cells unique to the primary and metastatic site[12,15,17]. Diversity of cellular origin shapes organ-specific CAF and MAF functional heterogeneity and contributes to the lack of available pan-fibroblast markers. Consequently, CAFs and MAFs are commonly identified by reporter models of lineage markers appropriate to the organ in question[15,17]. Following extensive characterisation, podoplanin has been identified as an efficient pan-marker of CAFs at the primary pancreatic tumour site, but labels only a minor population of hepatic MAFs[12,13,15,18]. In the liver of human and mouse, three distinct mesenchymal cell populations expressing the mesenchymal lineage marker PDGFRβ exist: i) fibroblasts (CD34+), ii) hepatic stellate cells (HStCs, Lrat+), and iii) vascular smooth muscle cells (VSMCs, Myh11+)[19,20].

Solid tumours are often highly infiltrated by macrophages and emerging studies indicate that a dynamic relationship between macrophages and fibroblasts exist, and that macrophages and fibroblasts regulate each other's functions[21–23]. However, in metastatic PDAC, the cellular origin of MAFs and whether and how MAF heterogeneity and function is regulated by macrophages remains poorly understood. A better molecular and cellular understanding of the complex interactions between macrophages and fibroblasts within the metastatic TME in PDAC is critical to improving therapies for this difficult to treat disease.

Here, we show that macrophages promote the activation of pSTAT3+myMAFs, which facilitates the outgrowth of disseminated pancreatic cancer cells in the liver. Mechanistically, pSTAT3+myMAF-derived osteopontin contributes to an immunosuppressive metastatic tumour microenvironment. Pharmacological blockade or genetic depletion of STAT3 restores an anti-tumour immune response and reduces metastatic outgrowth.

## Results

### Metastasis-associated macrophages regulate fibroblast heterogeneity in metastatic PDAC

To evaluate the presence of mesenchymal PDGFRβ+ cells in PDAC liver metastasis, we initially examined tissue sections obtained from liver biopsies of healthy and advanced metastatic PDAC patients using immunofluorescent staining. As expected, metastatic liver tumours, identified by the presence of CK19+ cancer cells, showed a significant expansion of PDGFRβ+ mesenchymal cells and a marked increase of CD68+ macrophages compared to healthy liver (Fig.1A, B). To study mesenchymal cell diversity in liver metastasis, we next utilised the mesenchymal reporter mouse model *Pdgfrb*-GFP, where fibroblasts,

VSMCs, and HStCs in the liver are efficiently labelled with the fluorescence reporter protein GFP[19].

Metastatic tumour burden was induced by intrasplenic injection of KPC-derived pancreatic cancer cells isolated from the genetically engineered mouse model of PDAC (*Kras*[G12D]; *TrpS3*[R172H]; *Pdx1-Cre*)[24]. Similarly, immunofluorescent staining of metastasis-bearing *Pdgfrb*-GFP mice revealed an expansion of GFP+ mesenchymal cells and rich infiltration of F4/80+ macrophages surrounding CK19+ metastatic cancer cells, recapitulating the TME of patient-derived biopsies (Fig. 1C, D). Co-expression of GFP+ mesenchymal cells with the fibroblast activation markers PDGFRα and αSMA confirmed that the *Pdgfrb*-GFP reporter mouse model efficiently labels metastasis associated mesenchymal cells in PDAC liver metastasis (Supplementary Fig. 1A, B).

Having validated our model, we next aimed to investigate how the presence, or absence, of macrophages impacts the mesenchymal landscape during liver metastasis progression (Fig. 1E). To deplete macrophages, mice were administered with αCSF1R neutralising antibody at day 7 post intrasplenic injection of KPC cells, a time point in which initial colonisation of the liver by KPC cells has already occurred[21,22,25]. Successful macrophage depletion was confirmed by flow cytometric analysis of viable, CD45+, CD11b+, F4/80+ cells, revealing a 65% reduction in metastasis-associated macrophages in αCSF1R treated, compared to IgG control, mice (Supplementary Fig. 1C, D). In addition to depleting macrophages, αCSF1R therapy significantly reduced both metastatic burden and liver fibrosis (Supplementary Fig. 1E–H), consistent with previous reports[21–23].

To confirm that our observations were not limited to the experimental metastasis model, we interrogated the stromal composition of spontaneous liver metastases derived from autochthonous KPC mice with advanced disease that were treated with AZD7507, a pharmacological inhibitor of CSF1R (CSF1Ri) (Supplementary Fig. 1I). Consistent with our results above, and in agreement with a previously reported survival benefit[23], CSF1Ri significantly reduced the overall metastatic burden in KPC mice, in comparison to control mice (Supplementary Fig. 1J, K). Further interrogation of metastasis-bearing livers revealed successful macrophage depletion and reduced fibrosis, among CK19+ lesions, in CSF1Ri-treated mice (Supplementary Fig. 1L–P).

To understand how macrophages regulate fibrosis, we analysed the gene expression profile of GFP+ mesenchymal cells isolated from αCSF1R, and IgG treated mice at a single cell level using the Chromium single-cell RNA sequencing platform (Fig. 1E). To distinguish mesenchymal cell clusters from those that are present under non-pathological conditions we also sequenced *Pdgfrb*-GFP+ cells from healthy livers. Enrichment of GFP+ cells was achieved by mechanical and enzymatic dissociation of livers into a single-cell suspension, followed by fluorescence-activated cell sorting (FACS) of viable, CD45-, Epcam-, CD31-, GFP+ cells to exclude immune, epithelial, and endothelial cells, respectively (Supplementary Fig. 2A). Unsupervised clustering analysis and Uniform Manifold Approximation and Projection (UMAP) of the healthy liver segregated cell clusters into three distinct cell populations that enriched for genes expressed by fibroblasts (*Gsn, Cd34, Pi16*), HStCs (*Reln, Lrat, Ecm1*), and VSMCs (*Myh11, Acta2, Myl9*) (Supplementary Fig. 2B, C), strongly aligning with previously defined populations of the hepatic mesenchyme (Supplementary Fig. 2D)[19].

Combined unsupervised clustering analysis of GFP+ cells from metastatic and healthy liver revealed 18 distinct clusters (Supplementary Fig. 2E). To robustly separate metastasis associated mesenchymal cells from the healthy mesenchyme, cell clusters were only defined as metastasis associated if they were unique to the metastasis-bearing dataset, meaning that <1% of cells from that cluster originated from the healthy mesenchyme (Supplementary Fig. 2E, F). Since upregulation of common fibrosis markers *Col1a1, Col1a2*, and *Col3a1* was found in all metastasis specific cell clusters, indicative for an activated fibroblast

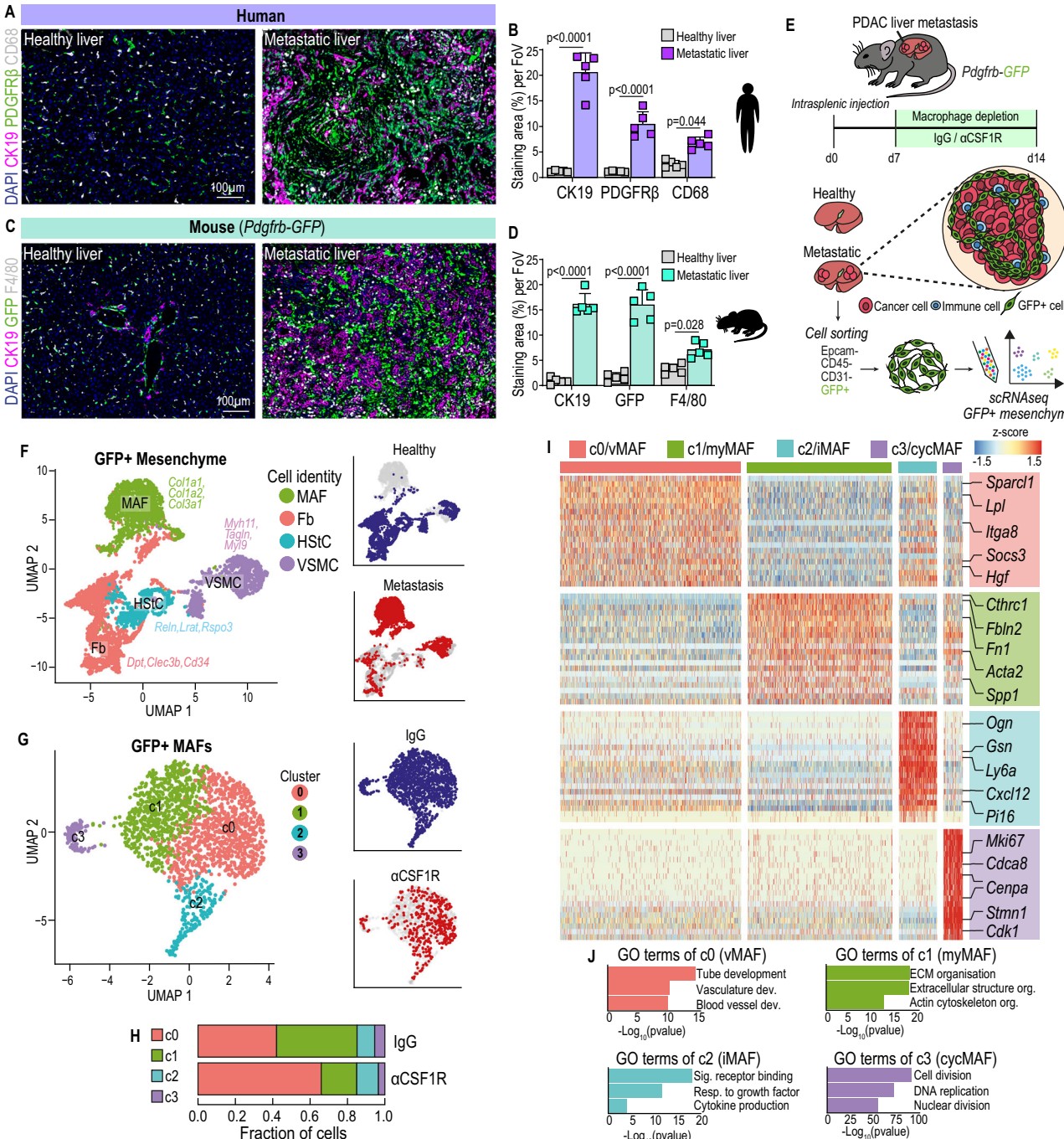

**Fig. 1 | Tumour associated macrophages regulate fibroblast heterogeneity in metastatic PDAC.** Representative immunofluorescence images (**A**) and quantification of cancer cells (CK19+), mesenchyme (PDGFRβ+), and macrophages (CD68+) (**B**) in healthy liver or PDAC liver metastasis (*n* = 5 patients per group). Scale bars, 100 μm. Data presented as mean percentage area. Error bars, SD. *P* values, two-way ANOVA with Tukey's multiple comparisons. Representative immunofluorescence images (**C**) and quantification of cancer cells (CK19+), mesenchyme (GFP+), and macrophages (F4/80+) (**D**) in healthy liver or PDAC liver metastasis in *Pdgfrb*-GFP mice (*n* = 5 mice per group). Scale bars, 100 μm. Data presented as mean percentage area, Error bars, SD. *P* values, two-way ANOVA with Tukey's multiple comparisons. **E** Schematic of scRNA sequencing on GFP+ cells enriched from metastatic livers of *Pdgfrb*-GFP mice intrasplenically implanted with

KPC-derived cancer cells and treated with IgG or αCSF1R. *n* = 4 mice were pooled per group. **F** UMAP plot of sequenced GFP+ cells coloured by (left) cell identity, or (right) by origin to healthy or metastatic tissue. **G** UMAP plot of GFP+ MAFs coloured by (left) cluster identity, or (right) treatment origin. **H** Bar chart depicting distribution of GFP + MAF clusters in metastatic livers of mice treated with IgG or αCSF1R. **I** Heat map depicting relative average expression of the top upregulated differentially expressed genes in each GFP + MAF cluster, compared to all other clusters in the scRNA-seq dataset. Representative genes are labelled for each cluster. Z-score distribution from −1.5 (blue) to 1.5 (red). **J** Gene ontology pathway enrichment analysis of discriminative marker genes of GFP + MAF clusters from **G**. Statistical enrichment analyses were performed using Fisher's exact test on g:Profiler. Source data are provided as a Source Data file.

state (Supplementary Fig. 2G)[12,26], these clusters were defined as metastasis associated fibroblasts (MAFs) (Fig. 1F). Meanwhile, undetectable expression of *Ptprc*, *Epcam*, or *Pecam1* confirmed no contamination from immune, epithelial, or endothelial cells, respectively

(Supplementary Fig. 2H). To further focus our analysis on transcriptional diversity, MAFs were extracted and reclustered using a shared nearest neighbour (SNN) graph constructed across the first 30 principal components of the dataset, revealing a total of four separate MAF

clusters (c0,1,2,3) with distinct transcriptional profiles and abundance (Fig. 1G and Supplementary Data 1). In liver metastases from IgG treated control mice, subcluster c0 and c1 represented the majority of MAFs with 42% and 43% abundance, respectively, whereas c2 (9.5%) and c3 (5.5%) were found to be less abundant clusters among all MAFs (Fig. 1G, H). Strikingly, beyond overall reduced fibrosis (Supplementary Fig. 1C), macrophage depletion also resulted in dramatic redistribution of the MAF landscape. The abundance of subcluster c0 among all MAFs increased from 42% to 65.6%, whereas subcluster c1 eroded by greater than half, from 43% to only 19% of all MAFs (Fig. 1H). Meanwhile, subcluster c2 increased from 9.5% to 11.6% and c3 decreased from 5.5% to 3.46%.

Subcluster c0 abundantly expressed *Sparcl1, Lpl, Socs3,* and *Hgf,* but lower levels of *Acta2* (encoding αSMA) and extracellular matrix (ECM) genes (Fig. 1I and Supplementary Fig. 3A). Gene ontology (GO) analysis of genes upregulated in c0 indicated significant enrichment for tube-, vascular- and blood vessel development (Fig. 1J), while expression of the common pericyte markers *Cspg4* and *Mcam* was absent (Fig. 1F and Supplementary Fig. 3B, C). Expression of *Pdgfra*, a common marker of activated fibroblasts, was enriched within c0-MAFs, but not among *Cspg4*+ nor *Mcam*+ cells (Supplementary Fig. 3B–D), suggesting that c0-MAFs are distinct from Cspg4+ or Mcam+ cells in the liver. In agreement with our transcriptional analysis, immunofluorescent staining of healthy livers revealed an enrichment of MCAM+ GFP+ cells surrounding blood vessel-like structures, while MCAM+ GFP+ cells were only sparely found within metastatic tumours (Supplementary Fig. 3E, F). Taken together, our data shows that c0-MAFs display a transcriptional phenotype of vascular remodelling and are present in tumour bearing, but not healthy, livers. Accordingly, c0-MAFs were named vascular-associated MAFs (vMAFs).

Subcluster c1 expressed high levels of *Acta2* and enriched for an ECM signature rich in Fibronectin (*Fn1*), Periostin (*Postn*), Versican (*Vcan*), and collagen-associated molecules (*Cthrc1, Col1a1,* and *Col1a2*) (Fig. 1I and Supplementary Fig. 3A). GO terms of c1-MAFs strongly associated with ECM and actin cytoskeleton organisation, similar to previously defined myCAFs in pancreatic tumours[12,18]. For consistent nomenclature, c1-MAFs were designated myofibroblastic MAFs (myMAFs).

Subcluster c2 expressed low levels of *Acta2*, but high levels of cytokines and growth factors including *Ogn, Gsn, Gas6,* and *Cxcl12* (Fig. 1I and Supplementary Fig. 3A). c2-MAFs enriched for the GO terms Signalling receptor binding, Response to growth factor, and Cytokine production (Fig. 1J), consistent with the iCAF subtype identified in pancreatic tumours[12,18]. Accordingly, c2-MAFs were named inflammatory MAFs (iMAFs).

Finally, subcluster c3 expressed a strong proliferative signature, including *Mki67, Cdca8,* and *Cdk1*, and enriched for the GO terms Cell division, DNA replication, and Nuclear division, indicative of a proliferative subset of MAFs (Fig. 1I, J and Supplementary Fig. 3A). Thereby, c3-MAFs were designated as cycling MAFs (cycMAF). Notably, retained expression of *Acta2* and ECM genes led us to conclude that cycMAFs are a subpopulation of proliferating myMAFs. A 36% reduction in cycMAF abundance in αCSF1R-treated metastatic tumours suggests that myMAFs are highly proliferative, and the absence of macrophages hampers myMAF proliferation, consistent with reduced overall fibrosis (Fig. 1H and Supplementary Fig. 1E, F).

Given the resemblance of myMAFs and iMAFs to previously defined CAF populations, we compared our annotated MAF clusters to the expression signatures of podoplanin+ myCAF and iCAF populations, derived from pancreatic tumours of KPC mice[12]. Heatmap projection and gene set enrichment analysis (GSEA) confirmed that myMAFs and iMAFs significantly enriched for myCAF ($p = 0.0437$) and iCAF ($p = 0.0014$) gene signatures, respectively, whereas vMAFs did not enrich with either (Supplementary Fig. 3G, H).

Alternative approaches of annotating CAF populations in pancreatic tumours have also been described, including a population of immune-suppressive *Lrrc15*+ myofibroblasts[13,27], and Endoglin (*Eng*, encoding CD105) positive and negative CAFs, with the latter associated with anti-tumour immunity[28]. While *Lrrc15* was undetectable in our transcriptomic analysis (Supplementary Fig. 3I), a gradient of *Eng* expression was observed with greater abundance in vMAFs and iMAFs, compared to myMAFs and cycMAFs (Supplementary Fig. 3J).

Finally, we explored the cellular origin of each MAF subcluster (c0-4) based on the expression of HStC (*Lrat*), fibroblast (*Ly6a*), and VSMC (*Myh11*) lineage markers. vMAFs, myMAFs, and cycMAFs highly expressed *Lrat*, suggesting HStC origin, whereas iMAFs exclusively expressed *Ly6a*, indicative of fibroblast origin (Supplementary Fig. 3K). The contribution of VSMC cells to the four MAF clusters was very minor to absent (Supplementary Fig. 3K). Taken together, our data reveals that, in PDAC liver metastases, MAFs exist as distinct populations that arise from a pool of liver-resident HStCs (vMAF, myMAF, and cycMAF) and fibroblasts (iMAF), and that macrophage-depletion leads to less myMAFs.

## Transcriptional and spatially diverse MAF populations co-exist in PDAC liver metastasis

We next sought to validate the presence of the defined MAF populations in liver tissue sections and understand whether they are conserved across both murine and human disease. Since our transcriptional analysis revealed cycMAFs as dividing cells among myMAFs, we focused our further analysis on the three functionally distinct MAF clusters: vMAF, myMAF, and iMAF. To enable immunofluorescent detection of our identified MAF subpopulations, we aimed to define common fibroblast markers that were enriched within each cluster and that would not likely be present in other non-mesenchymal cells, in line with published guidance[29]. Among these three MAF clusters, we observed that common fibroblast markers displayed gradated expression patterns rather than complete presence or absence (Fig. 2A). *Acta2* (αSMA) was expressed among all MAFs, but was particularly higher in myMAFs, whereas *Pdgfra* showed higher expression levels in vMAFs and iMAFs (Fig. 2A and Supplementary Fig. 4A). *Ly6a* (Sca-1), identifying MAFs of fibroblast origin, lacks a human ortholog, therefore *Cd34*, which similarly displayed gradated expression, was utilised as a cross-species marker of iMAFs (Fig. 2A).

Accordingly, we applied the gradated expression of αSMA, PDGFRα, and CD34 as a surrogate strategy for in-situ detection of vMAFs (αSMA^{low}PDGFRα^{high}CD34^{low}), myMAFs (αSMA^{high}PDGFRα^{low}CD34^{low}), and iMAFs αSMA^{low}PDGFRα^{high}CD34^{high}) (Supplementary Fig. 4B). Using this approach, all three MAF subpopulations were detectable in metastatic tumours of *Pdgfrb*-GFP mice (Supplementary Fig. 4C), in liver metastasis of autochthonous KPC mice (Fig. 2B), and patient-derived liver biopsies (Supplementary Fig. 4D). Notably, some αSMA^{high}PDGFRα^{high} double positive cells were also detectable, which could represent cells at the interphase between clusters. We next attempted to define the spatial localisation of MAFs in liver metastasis. Given the enrichment for GO terms associated with vascular development (Fig. 1J), we hypothesised that vMAFs reside proximal to intra-metastatic vasculature and used antibodies against CD31, identifying endothelial cells, to define regions high and low in intra-metastatic vascularisation (Supplementary Fig. 4E). We observed no significant difference in the number of GFP+ cells among CD31-rich or -poor regions (Supplementary Fig. 4F), suggesting that MAFs reside in both rich and poorly vascularised regions. However, when vMAF and myMAF subtypes were distinguished by the divergent expression of αSMA and PDGFRα, we observed that αSMA^{low}PDGFRα^{high} cells enriched in CD31-rich regions, whereas αSMA^{high}PDGFRα^{low} cells enriched in CD31-poor regions (Supplementary Fig. 4G, H). Together, these results suggest that vMAFs may reside proximal to tumour vascularisation, whereas myMAFs are found in less vascularised regions.

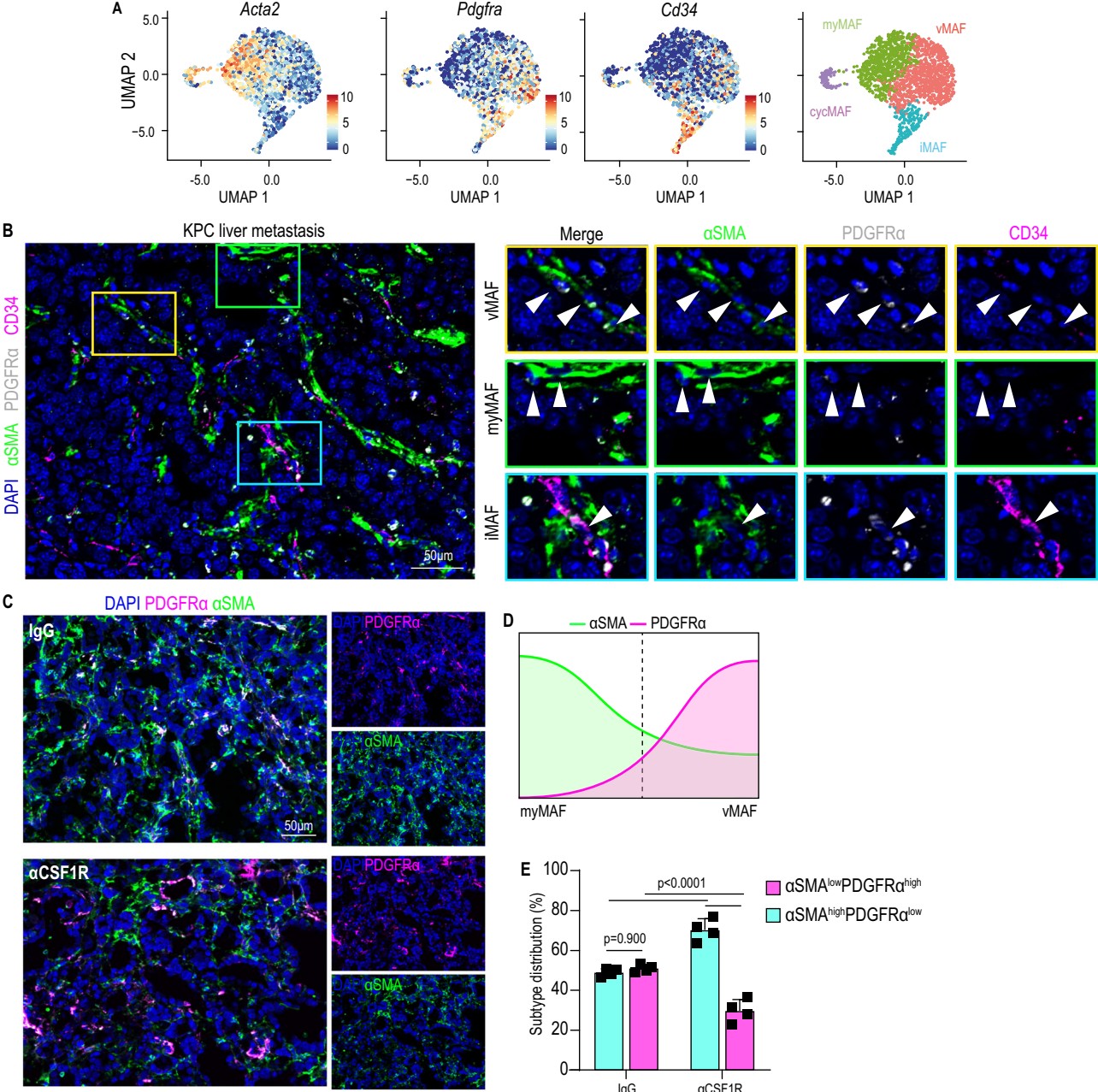

**Fig. 2 | Transcriptional and spatially diverse MAF populations co-exist in PDAC liver metastasis. A** UMAP plot depicting expression of *Acta2*, *Pdgfra*, and *Cd34* in vMAF, myMAF, iMAF, and cycMAF clusters. **B** Representative immunofluorescence image of annotated MAF subtypes in liver metastasis of KPC mice. Individual MAF subtypes are defined on the representative image in coloured regions: vMAF (αSMA$^{Low}$, PDGFRα$^{High}$, CD34$^{Low}$) – red; myMAF (αSMA$^{High}$, PDGFRα$^{Low}$, CD34$^{Low}$) – green; iMAF (αSMA$^{Low}$, PDGFRα$^{High}$, CD34$^{High}$) – blue. Arrowheads indicate the defined MAF subtype. *N* = 3 mice. Scale bar, 50 μm. **C** Representative immunofluorescence image for the distribution of PDGFRα and and αSMA to discern vMAFs and myMAFs in metastatic tumours derived from *Pdgfrb*-GFP mice treated with IgG or αCSF1R neutralising antibody. To the right, separated channel images are shown. Scale bar, 50 μm. **D** Representative schematic depicting how the gradated expression of αSMA and PDGFRα separates across vMAF and myMAF subpopulations, as observed in **A**, is utilised as a strategy to analyse changes in abundance between both subpopulations. **E** Quantification of the distribution of αSMA$^{low}$PDGFRα$^{high}$ and αSMA$^{high}$PDGFRα$^{low}$ cells in **C**. Data is presented as mean percentage distribution averaged from *n* = 4 mice per group. Error bars, SD. *P* values, two-way ANOVA with Tukey's multiple comparisons test. Source data are provided as a Source Data file.

Finally, using our established markers, we aimed to visualise and validate the concomitant increase in vMAFs (αSMA$^{low}$PDGFRα$^{high}$) and reduction in myMAFs (αSMA$^{high}$PDGFRα$^{low}$) observed in αCSF1R-treated mice, compared to IgG control, on a transcriptional level (Fig. 1G, H). In agreement with our transcriptional analysis, a similar ratio of αSMA$^{low}$PDGFRα$^{high}$ vMAFs to αSMA$^{high}$PDGFRα$^{low}$ myMAFs in IgG-treated mice was altered towards an abundance of vMAFs and fewer myMAFs in αCSF1R-treated mice (Fig. 2C–E).

Corroborating these observations, quantification of picrosirius red staining revealed a significant reduction in the deposition of collagen rich extracellular matrix, primarily expressed by myMAFs, in αCSF1R-treated tumours (Supplementary Fig. 5A, B). Interestingly, αCSF1R-treated tumours also associated with an increase in CD31+ intra-tumoural blood vessels (Supplementary Fig. 5C, D), suggesting that an imbalance of vMAF/myMAF ratio, and/or depletion of collagen deposition, may facilitate vascularisation of the metastatic TME.

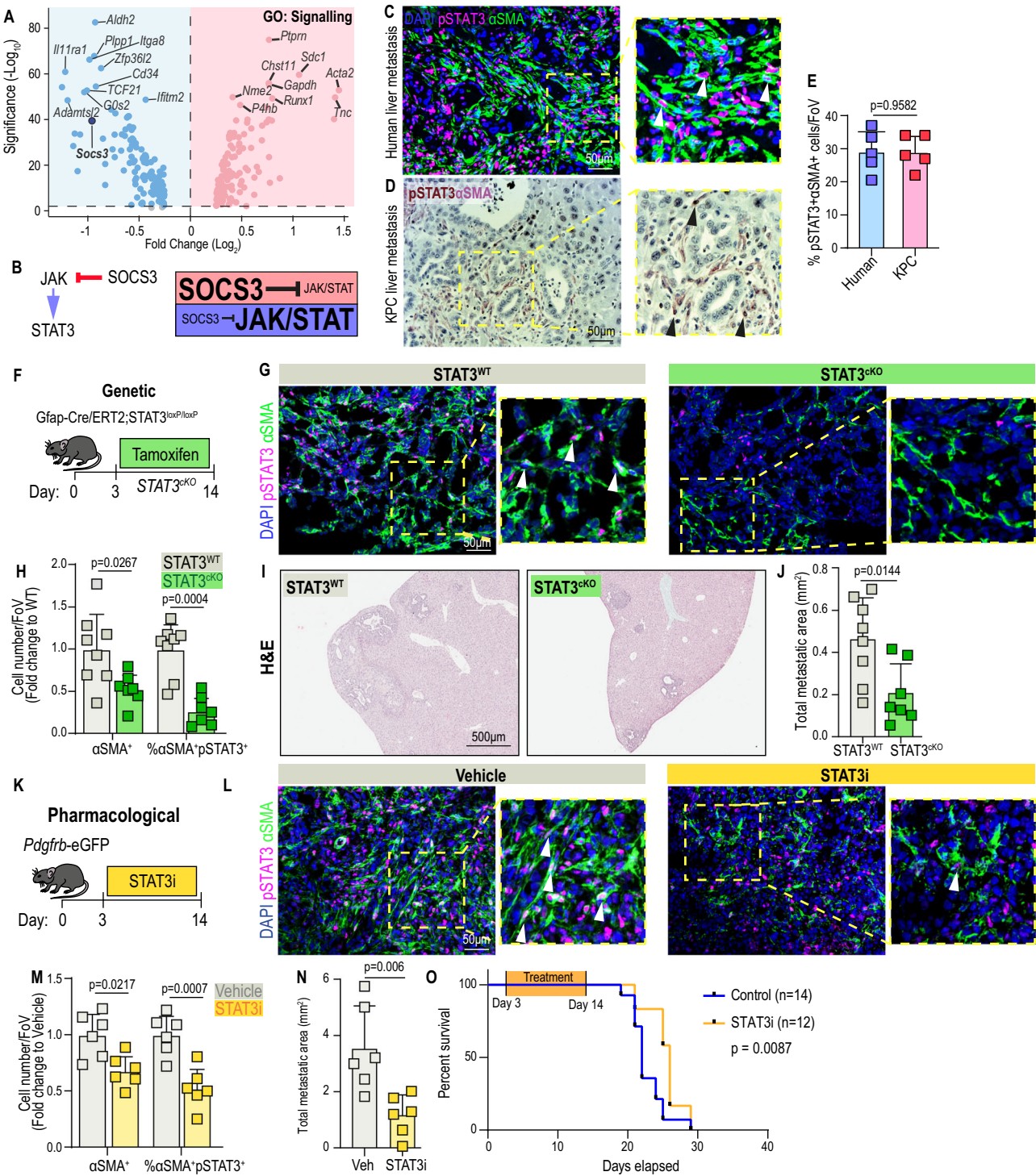

Taken together, our data suggest that the three functionally distinct MAF populations identified are conserved across human and murine PDAC liver metastasis and are spatially organised to regions proximal and distal to tumour vasculature. Meanwhile, depletion of macrophages reprograms the fibrotic landscape from a classically collagen-rich desmoplasia to an increasingly vascularised TME through a reduction in myMAFs.

### myMAFs support liver metastasis and their pro-metastatic function depends on activation of the JAK/STAT signalling pathway

We and others have previously shown that a collagen-dense extracellular matrix promotes cancer disease progression and impairs

treatment response[21,22,30]. Given that myMAFs are the predominant source of collagen-rich ECM and that macrophage depletion markedly reduced their abundance, we next aimed to elucidate how macrophages regulate myMAF activation and function.

To explore active signalling pathways, we queried up and down-regulated genes among myMAFs, compared to vMAFs, iMAFs, and cycMAFs, that are associated with the gene ontology term Signalling (GO:0023052). Here we identified that *Socs3*, a prominent and potent suppressor of the JAK/STAT pathway, was among the most down-regulated signalling-associated genes in myMAFs (Fig. 3A, B)[31]. Activated JAK phosphorylates STAT3 on Tyr[705], resulting in the homodimerization of two STAT3 molecules and subsequent nuclear translocation. Meanwhile, Socs3, via its SH2 domain, directly binds to

**Fig. 3 | myMAFs support liver metastasis and their pro-metastatic function depends on the activation of JAK/STAT signalling pathway.** **A** Volcano plot of DEG among myMAFs (vs vMAF;iMAF;cycMAF) enriching for Gene Ontology term: Signalling (GO:0023052). **B** Illustration of JAK/STAT regulation by SOCS3. High SOCS3, repressed JAK/STAT signalling. Low SOCS3, active JAK/STAT signalling. **C** Representative immunofluorescence image of JAK/STAT active (pSTAT3+) myMAFs (αSMA+) in human PDAC liver metastasis. Arrowheads, double positive cells. Scale bar, 50 μm. **D** Representative immunohistochemical image of JAK/STAT active (pSTAT3+; DAB, brown) myMAFs (αSMA+; VIP purple) in spontaneous liver metastasis of KPC mice. Arrowheads indicate double positive cells. Scale bar, 50 μm. **E** Mean percentage of pSTAT3+αSMA+ cells, among all αSMA+ cells, depicted in **C** and **D**. N = 5 independent samples. Error bars, SD. P value, two-tailed unpaired t-test. **F** Experimental design for generating metastasis bearing GFAP-STAT3 conditional knockout mice (STAT3$^{cKO}$). **G** Representative immuno-fluorescence images of JAK/STAT active (pSTAT3+) myMAFs (αSMA+) in metastatic tumours of STAT3$^{WT}$ and STAT3$^{cKO}$ mice. Arrowheads, double-positive cells. Scale

bar, 50 μm. **H** Mean percentage of αSMA+, and percentage of pSTAT3+αSMA+ cells, presented as fold change relative to STAT3$^{WT}$. n = 8 STAT3$^{WT}$, n = 7 STAT3$^{cKO}$ mice. Error bars, SD. P value, two-way ANOVA with Tukey's multiple comparisons. Representative H&E staining (**I**) and average sum of metastatic area per group (**J**). Scale bar, 500 μm. n = 8 STAT3$^{WT}$, n = 7 STAT3$^{cKO}$ mice. Error bars, SD. P value, two-tailed unpaired t-test. **K** Experimental design for pharmacological inhibition of pSTAT3 with Silibinin (STAT3i), in Pdgfrb-GFP mice. **L** Representative immuno-fluorescence images of JAK/STAT active (pSTAT3+) myMAFs (SMA+) in metastatic tumours of vehicle or STAT3i-treated mice. Arrowheads indicate double-positive cells. Scale bar, 50 μm. **M** Mean percentage of αSMA+, and percentage of pSTAT3+αSMA+ cells, presented as fold change relative to control. n = 6 mice per group. Error bars, SD. P value, two-way ANOVA with Tukey's multiple comparisons. **N** Average sum of metastatic area per group. n = 6 mice per group. Error bars, SD. P value, two-tailed unpaired t-test. **O** Kaplan–Meier survival analysis of metastasis-bearing mice treated with vehicle (n = 14) or STAT3i (n = 12). P value, log-rank test and Cox proportional hazards test. Source data are provided as a Source Data file.

phosphorylated tyrosine's of activated JAK, thereby inhibiting the recruitment, phosphorylation, and nuclear translocation of STAT3[31]. In contrast, *Socs3* expression was markedly increased in vMAFs and iMAFs, suggestive of repressed JAK/STAT signalling among these MAF subtypes (Supplementary Fig. 3A). In liver metastases of both PDAC patients and the autochthonous KPC mice, we observed that on average 30% of αSMA+ cells stained positive for pSTAT3, confirming that the JAK/STAT pathway is active within MAFs (Fig. 3C–E). To explore whether JAK/STAT signalling was predominantly active within myMAFs, we utilised the gradated expression of PDGFR to distinguish vMAFs and iMAFs (both GFP$^{pos}$PDGFRα$^{high}$) from myMAFs (GFP$^{pos}$PDGFRα$^{low}$), revealing that pSTAT3 activity was almost exclusively enriched among myMAFs (Supplementary Fig. 5E, F). Consistent with our transcriptomic data, depletion of macrophages by αCSF1R therapy significantly reduced the percentage of pSTAT3$^{pos}$αSMA$^{pos}$ MAFs, among all αSMA+ cells, within metastatic tumours, compared to IgG control treated mice (Supplementary Fig. 5G, H). Taken together, these results suggest that JAK/STAT is active in myMAFs, and its activity is linked to the presence of macrophages.

Our prior results suggest that myMAFs are of HStC-lineage (Supplementary Fig. 3G). *Gfap* promoter-driven Cre recombinase has previously been described as a model for selective deletion in HStCs[32–34]. Therefore, to test the biological relevance of pSTAT3+ in myMAFs to metastatic outgrowth, we induced experimental metastasis in an inducible GFAP-STAT3 KO mouse model (STAT3$^{cKO}$: *GFAP-Cre/ERT2;STAT3$^{loxP/loxP}$*) (Fig. 3F). Three days after intrasplenic injection of KPC derived cells, tamoxifen (Tmx) was administered to both control STAT3$^{WT}$ (Cre−) and STAT3$^{cKO}$ (Cre+) and 11 days later livers were harvested for histological interrogation (Fig. 3F). Successful depletion of STAT3 was validated by visualising pSTAT3 activity among desmin+ cells, a robust marker of HStCs[35], confirming a significant reduction in JAK/STAT activity among HStC-MAFs (Supplementary Fig. 5I, J). In agreement with reduced myMAF activation, STAT3$^{cKO}$ tumours showed an overall reduction in αSMA+ fibrosis, percentage of pSTAT3+αSMA+ cells (among αSMA+ cells), and collagen deposition (Fig. 3G, H and Supplementary Fig. 5K, L). Critically, STAT3$^{cKO}$ mice displayed a significant reduction in metastatic tumour burden, compared to control mice, suggestive of a tumour-promoting function for pSTAT3+ myMAFs (Fig. 3I, J).

As a more translational approach, we next tested whether the pro-tumorigenic functions of pSTAT3+myMAFs could be inhibited by the systemic administration of a pharmacological STAT3 inhibitor (STAT3i) (Fig. 3K). Silibinin, a plant-derived secondary metabolite, is a direct STAT3i binding with high affinity to the SH2 domain that is critical for phosphorylation and transcriptional activity of STAT3[36,37]. Daily administration of silibinin to metastasis-bearing *Pdgfrb*-GFP mice led to a significant reduction in αSMA+ fibrosis, percentage of pSTAT3+αSMA+ cells (among αSMA+ cells), and collagen deposition

(Fig. 3L, M and Supplementary Fig. 5M, N). Most importantly, pharmacological STAT3i significantly reduced metastatic tumour burden and associated with an increase in overall survival (Fig. 3. N & O and Supplementary Fig. 5O). Taken together, our data suggests that macrophages contribute to the accumulation of pro-tumourigenic pSTAT3+ myMAFs, and that HStC-specific genetic depletion of STAT3, or pharmacological STAT3 inhibition, is sufficient to disrupt myMAF activation and their associated pro-metastatic functions.

## Co-stimulation of progranulin and cancer cell-derived factors promote myMAF activation

Having identified JAK/STAT as an essential signalling pathway for myMAF activation, we next investigated how macrophages activate the JAK/STAT pathway in HStCs. We previously identified progranulin, secreted by monocyte derived macrophages, as a key regulator of hepatic fibrosis, therefore we probed whether progranulin was a key player in the activation of JAK/STAT signalling in HStCs[21,22]. To test this, we generated chimeric *Pdgfrb*-GFP mice harbouring WT bone marrow (BM), or BM lacking progranulin (*Grn−/−*) (Fig. 4A). Successful recon-stitution with WT or Grn−/− BM was confirmed by qPCR of isolated peripheral blood cells and mice were enroled into metastasis studies (Supplementary Fig. 6A). Histological interrogation of metastatic tumours revealed a significant reduction in both metastatic outgrowth and fibrosis (GFP+ cells within CK19+ lesions) in Grn−/− BM mice, compared to WT BM control mice (Fig. 4B–E). Strikingly, visualisation of STAT3 activity revealed a marked reduction in the proportion of pSTAT3+ cells among αSMA+ cells (Fig. 4F, G), which also coincided with a significant reduction in collagen deposition (Supplementary Fig. 6B, C). Taken together, our data suggests that genetic depletion of progranulin in the BM of mice is sufficient to reduce the accumulation of metastasis promoting pSTAT3+myMAFs in PDAC liver metastasis.

To better understand the contribution of macrophage-derived progranulin to the activation of pSTAT3+myMAFs, we adapted a three-dimensional culture system, previously described for PStCs, to culture primary murine HStCs, isolated by retrograde hepatic perfusion and adherence selection (Supplementary Fig. 6D)[18,38]. Primary HStCs are quiescent by nature, and 3D-matrigel embedding retains quiescence making it an efficient tool for monitoring phenotype transitions in response to external stimuli. Accordingly, HStCs were exposed to CM from tumour educated bone marrow-derived macrophages (BMDMs), isolated from WT and Grn−/− mice, cancer cell CM, or both, for 4 days prior to interrogation for myMAF (*Cthrc1, Spp1, Acta2, Col1a1, Col1a2, Postn*) and vMAF (*Socs3, Hgf, Pdgfra, Itga8*) gene signatures (Fig. 4H). While exposure to WT-BMDM CM alone induced the expression of several myMAF and vMAF genes, the combination of CM from both WT-BMDMs and cancer cells amplified the induction of *Cthrc1, Spp1*, and *Postn*, while suppressing several vMAF genes, indicating an induction of a myMAF phenotype (Fig. 4I). Notably, this phenotype was

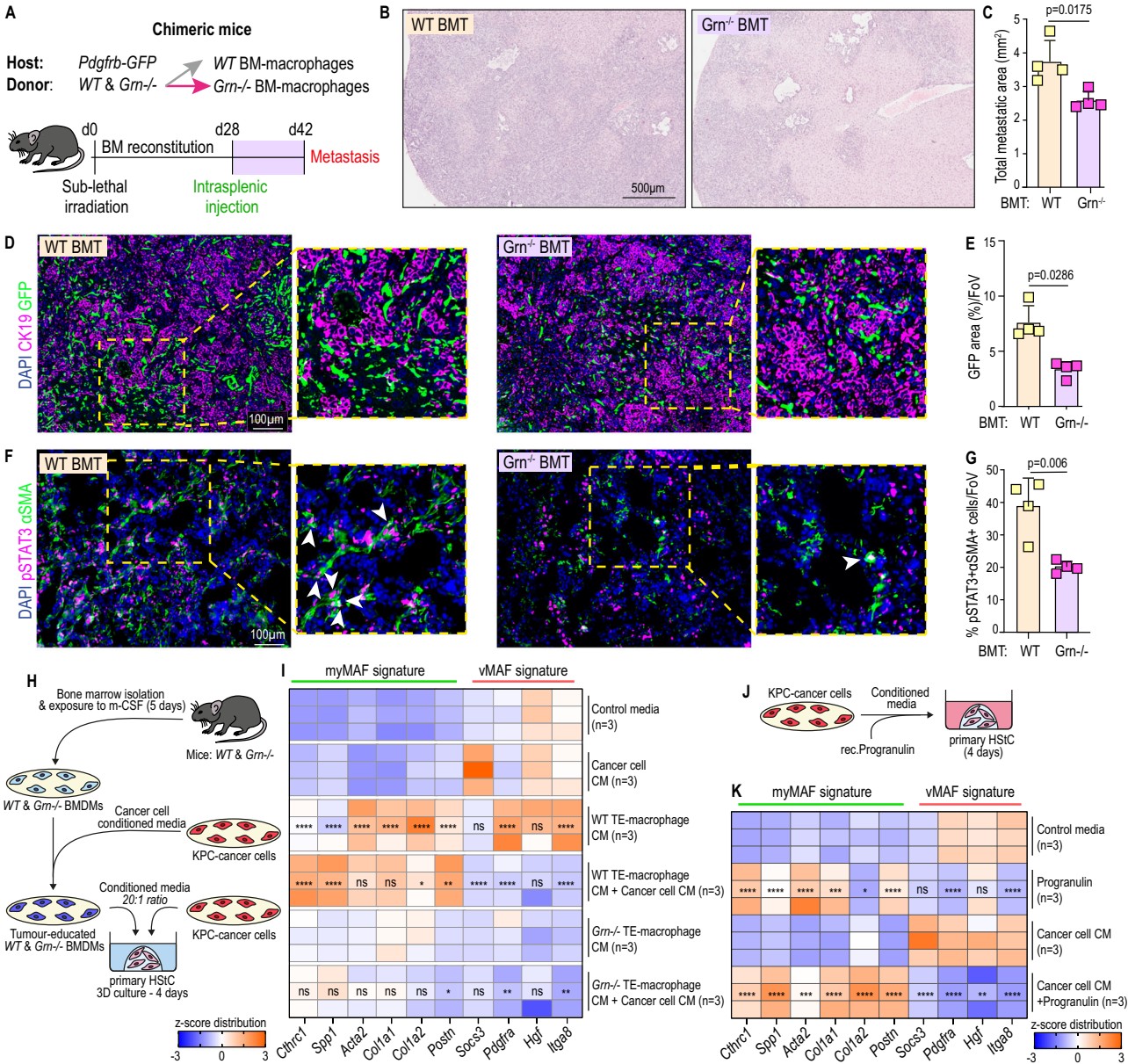

**Fig. 4 | Co-stimulation of progranulin and cancer-cell derived factors promote myMAF activation. A** Generation of chimeric *Pdgfrb*-GFP mice with either WT, or *Grn⁻/⁻*, bone marrow (BM) reconstitution. WT and *Grn⁻/⁻* BM transplanted mice were implanted with KPC cells and terminated after 14 days. Representative H&E staining (**B**) and quantification of the average sum of metastatic area per group (**C**). Scale bar, 500 μm. *n* = 4 mice per group. Error bars, SD. *P* value, two-tailed unpaired *t*-test. Representative immunofluorescence image (**D**) and quantification of MAFs (GFP+) among metastatic tumours (CK19+) of WT and *Grn⁻/⁻* BM *Pdgfrb*-GFP mice (**E**). Scale bar, 100 μm. *n* = 4 mice per group. Data is presented as mean GFP percentage area. Error bars, SD. *P* value, two-tailed Mann−Whitney test. Representative immunofluorescence image (**F**) and quantification (**G**) of JAK/STAT active (pSTAT3+) myMAFs (αSMA+). Arrowheads, double positive cells. Scale bar, 100 μm. *n* = 4 mice per group. Data is presented as average percentage of pSTAT3+αSMA+ cells, among αSMA+ cells. Error bars, SD. *P* value, two-tailed unpaired *t*-test. **H** Schematic of experimental design. Bone marrow-derived macrophages were generated from WT and *Grn⁻/⁻* mice and educated with cancer cell CM for 24 h. Primary HStCs were exposed to conditioned media from tumour educated macrophages and cancer cells, alone or in combination, for 4 days. **I** Heatmap of selected myMAF and vMAF gene signature expression in primary HStCs stimulated as outlined in **H**. *N* = 3 independent experiments. *P* value, one-way ANOVA with Tukey's multiple comparisons. Significance shown depict: WT TE-Macrophage CM vs Control media; WT TE-Macrophage CM+ Tumour CM vs WT TE-Macrophage CM; *Grn⁻/⁻* TE-Macrophage CM+ Tumour CM vs *Grn⁻/⁻* TE-Macrophage CM. TE Tumour educated. **J** Schematic of experimental design. Primary HStCs were exposed to cancer cell CM supplemented with recombinant progranulin for 4 days. **K** Heatmap of selected myMAF and vMAF gene signature expression in primary HStCs stimulated as outlined in **J**. *n* = 3 independent experiments. *P* value, one-way ANOVA with Tukey's multiple comparisons. Significance shown, Progranulin vs Control media; Cancer cell CM+ Progranulin vs Cancer cell CM. Source data and exact *p* values are provided as a Source Data file.

not recapitulated when HStCs were exposed to Grn−/− CM under similar conditions (Fig. 4I).

To further validate the contribution of progranulin, HStCs were stimulated with cancer cell CM in the presence and absence of recombinant progranulin (Fig. 4J). Likewise, while cancer cell CM alone suppressed a myMAF signature, the addition of recombinant

progranulin restored expression of several myMAF genes, including *Ctrhc1*, *Spp1*, *Acta2*, and *Postn*, while downregulating a vMAF signature (Fig. 4K).

Together, these results highlight a key role for macrophage-derived progranulin in the activation of pSTAT3+myMAFs in vivo, while the complementarity of macrophage-derived progranulin and

cancer-cell derived factors promotes the activation and acquisition of a myMAF phenotype in vitro.

## Neutralisation of cancer-cell derived LIF represses the activation of pSTAT3+myMAFs and inhibits metastatic outgrowth

The interleukin 6 (IL6) cytokine superfamily are known direct and potent activators of JAK/STAT, and leukaemia inhibitory factor (LIF) has previously been shown to induce the activation of pancreatic stellate cells (PStCs)[18]. Therefore, we queried whether LIF, and other IL6 family cytokines, were responsible for complementing progranulin to induce a myMAF phenotype in vitro.

Of the IL6-family cytokines tested, *Lif* mRNA transcripts were the most abundant in cancer cells, compared to *Il6* and *Osm* (Supplementary Fig. 6E). To test the contribution of cancer cell-derived LIF to the induction of myMAFs in vitro, primary HStCs were exposed to cancer cell CM, pre-treated with IgG or LIF neutralising antibody (LIF-nAb), and supplemented with recombinant progranulin (Fig. 5A). In the presence of IgG, cancer cell CM with progranulin induced the expression of myMAF genes, including *Spp1*, *Acta2*, and *Postn*, while the neutralisation of LIF partially suppressed this induction (Fig. 5B). Moreover, conditioned media generated from primary HStCs stimulated with recombinant progranulin and exposed to cancer cell CM in the presence of control IgG, supported KPC cancer cell colony formation compared to control media (Fig. 5C–E). As expected, CM media generated from HSCs stimulated with recombinant progranulin and cancer cell CM in the presence of LIF-nAb showed a significant reduction in promoting colony formation compared to IgG group (Fig. 5E).

To further study the role of progranulin and LIF in myMAF activation, HStCs were stimulated with recombinant progranulin, LIF, or both, for 4 days. Both progranulin and LIF alone were capable of inducing the expression of *Cthrc1*, *Spp1*, and *Acta2*, but to differing levels, while LIF alone suppressed *Socs3*, but induced *Pdgfra* (Fig. 5F). However, under co-stimulation of progranulin and LIF, HStCs abundantly expressed a myMAF signature, including a marked induction of *Cthrc1* and *Spp1*, while suppressing vMAF genes (Fig. 5F).

On protein level, while progranulin induced high expression of αSMA, independently of JAK/STAT (Fig. 5G–I), LIF induced persistent STAT3 phosphorylation but moderate αSMA expression, whereas co-stimulation with both progranulin resulted in both persistent STAT3 phosphorylation and high expression of αSMA (Fig. 5G–I), consistent with the pSTAT3+myMAF phenotype observed in vivo. Together, these results supported the complementary effect of LIF and progranulin for inducing pSTAT3+myMAFs in vitro.

Partial suppression of a pSTAT3+myMAF phenotype by neutralisation of LIF from cancer cell CM suggested that other cancer cell-derived factors may also play a role (Fig. 5B). Therefore, we extended our in vitro culture assay to other members of the LIF cytokine family, such as IL6. While co-stimulation of IL6 with progranulin enhanced the expression of the myMAF markers *Cthrc1* and *Spp1*, collagen genes and *Postn* were suppressed, suggesting a myMAF phenotype was not sufficiently induced (Supplementary Fig. 7B). LIF and IL6 are known to bind to their respective receptors, Il6 receptor (IL6R) and LIF receptor (LIFR), with high affinity, but no cross reactivity[39]. Therefore, the unique signature induced by LIF, in the presence of progranulin, suggests that a pSTAT3+myMAF phenotype could be mediated specifically through the LIFR. In support of this statement, siRNA-mediated knockdown of the IL6R (*Il6ra*) revealed a largely dispensable function, apart from *Col1a2* and *Postn*, to the induction of a myMAF phenotype by co-stimulation of LIF and progranulin (Supplementary Fig. 6G, H). Taken together, our results identify LIF as a key cancer cell derived factor, which combined with progranulin induces a pSTAT3+myMAF phenotype in vitro.

Next, we assessed the source of Progranulin and LIF expression in the metastatic TME in vivo. By isolating Epcam+ cancer cells, F4/80+ macrophages, and GFP+ MAFs through flow cytometry-based cell sorting from metastasis-bearing *Pdgfrb*-GFP mice, we confirmed that, in agreement with previous reports[22], tumour-associated macrophages were the main source of progranulin (*Grn*) (Fig. 5J), whereas cancer cells were the main source of *Lif* transcripts (Fig. 5K). To test the role of LIF in promoting metastatic outgrowth in vivo, metastasis-bearing *Pdgfrb*-GFP mice were treated with LIF-nAb, or IgG control (Fig. 5L). Neutralisation of LIF inhibited metastatic outgrowth (Fig. 5M, N), which coincided with a significant reduction in overall fibrosis (Fig. 5O, P), percentage of pSTAT3+αSMA+ cells (among αSMA+ cells) (Fig. 5Q, R), and collagen deposition (Supplementary Fig. 6I, J). Taken together, our results support that progranulin, mainly derived from macrophages, and LIF, mainly derived from cancer cells, drives pSTAT3+myMAF activation, and inhibition of either progranulin or LIF is sufficient to disrupt their associated pro-metastatic functions.

## The binding of progranulin to Sortilin enhances Sortilin-LIFR proximity, leading to STAT3 hyperactivation in HStCs

Having identified progranulin and LIF as key mediators of pSTAT3+myMAFs, we wanted to better understand the underlying molecular mechanism. The time taken to recover primary HStCs from matrigel meant this model was unsuitable for exploring signalling events that may occur early during phenotypical MAF transitions. Therefore, we utilised LX2 cells, a human HStC-derived cell line representing activated myofibroblasts when cultured in a 2D monolayer[40], and pharmacologically reverted these cells to a quiescent phenotype with calcipotriol, a vitamin D analogue, as previously reported[32,41]. Strong induction of *CYP24A1*, a direct vitamin D receptor target gene, confirmed efficient response to calcipotriol, and led to an accumulation of BODIPY+ lipid droplets and downregulation of *ACTA2* transcripts, both hallmarks of stellate cell quiescence, after only 24 h (Supplementary Fig. 6K–M). For all subsequent studies, LX2 cells were pre-treated with calcipotriol for 24 h prior use.

Next, we explored whether stimulation of quiescent LX2 cells with recombinant progranulin, LIF, or both, induced STAT3 activation. Consistent with our prior results, progranulin alone induced no STAT3 phosphorylation, suggesting that it does not directly activate STAT3 in HStCs (Fig. 5D, E and Supplementary Fig. 6N). When used as a single agent, LIF induced rapid STAT3 phosphorylation within 10 min, which peaked at 30 min (Fig. 6A, B). Remarkably, at each time point, the addition of progranulin to LIF amplified and prolonged STAT3 phosphorylation, compared to LIF stimulation alone (Fig. 6A, B). A similar induction was observed upstream at JAK1, within 5 min, however this was rapidly turned over (Supplementary Fig. 6O, P). These results suggest that, mechanistically, progranulin-mediated hyperactivation of LIF signalling was likely occurring upstream of JAK/STAT, possibly via LIFR at the cell surface.

The rapid internalisation of progranulin by neurons and astrocytes is mediated by the sortilin receptor, and the free C-terminal domain of progranulin has been mapped to be essential for this interaction[42]. To determine whether the interaction of sortilin with the C-terminal end of progranulin was essential for STAT3 hyperactivation, we generated two different recombinant progranulin constructs, one full length (FL-PGRN) and a truncated version, lacking the C-terminal end (Trunc-PGRN) (Fig. 6C, D). Both constructs were tagged with StreptagII for purification, and mCherry for fluorescent detection. To confirm that sortilin is necessary for progranulin uptake, LX2 cells were transfected with *SORT1* siRNA in combination with siGLO transfection indicator, followed by overnight loading of cells with recombinant FL-PGRN (Fig. 6E). Quantification of mCherry fluorescent intensities among siGLO+ cells revealed that sortilin knockdown markedly ablated FL-PGRN uptake, confirming that in HStCs the sortilin receptor is essential for progranulin uptake (Fig. 6F, G). Next, we assessed whether sortilin, and progranulin uptake, was necessary for progranulin-mediated STAT3 hyperactivation. Co-stimulation of FL-PGRN and LIF

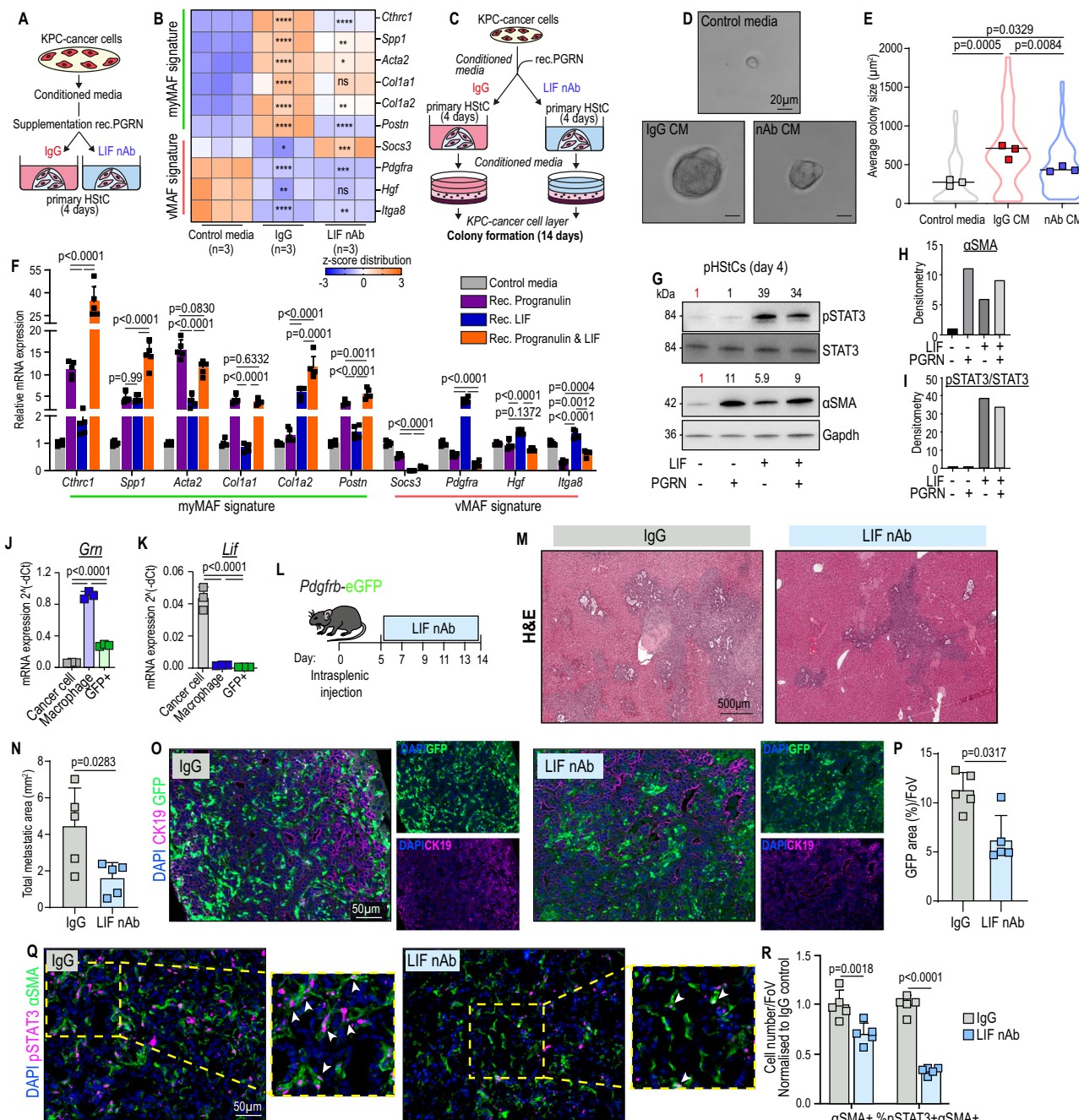

**Fig. 5 | Neutralisation of cancer-cell derived LIF represses the activation of pSTAT3+myMAFs and inhibits metastatic outgrowth.** Schematic of experiment (**A**) and qPCR (**B**) of vMAF and myMAF genes. Data is presented as heatmap of z-scores. $n = 3$ independent experiments. *P* value, one-way ANOVA with Tukey's multiple comparisons. Significance shown: IgG vs Control media; LIF-nAb vs IgG. Schematic of experimental design (**C**) and representative images (**D**) of colony formation. Scale bar: 20 μm. **E** Quantification of average colony size presented as average from $n = 3$ independent biological experiments. *P* value, one-way ANOVA with Tukey's multiple comparisons. **F** qPCR of myMAF and vMAF genes in primary HStCs stimulated with Progranulin, LIF, or both. Data represents average fold change. $n = 5$ independent experiments. Error bars, SD. *P* value, one-way ANOVA with Tukey's multiple comparisons. **G–I** Immunoblotting (**I**) of primary HStCs stimulated with Progranulin, LIF, or both, and densitometric analysis of (**H**) αSMA and (**I**) pSTAT3/STAT3 activity relative to Gapdh. Experiment was repeated three times with similar results. qPCR of (**J**) *Grn* and (**K**) *Lif* mRNA expression in Cancer cells, Macrophages, and MAFs isolated from metastasis bearing *Pdgfrb*-GFP mice. $n = 3$

independent mice. Error bars, SD. *P* value, one-way ANOVA with Tukey's multiple comparisons. **L** Schematic of experimental design to treat metastasis-bearing *Pdgfrb*-GFP mice with IgG or LIF-nAb. Representative H&E staining (**M**) and quantification (**N**) of average metastatic area from IgG and LIF-nAb treated mice. Scale bar: 500 μm. $n = 5$ mice per group. Error bars, SD. *P* value, two-tailed unpaired *t*-test. Representative immunofluorescence image (**O**) and quantification (**P**) of MAFs (GFP+) among metastatic tumours (CK19+) of IgG and LIF-nAb treated mice. Scale bar, 50 μm. $n = 5$ mice per group. Data represents average percentage area of GFP. Error bars, SD. *P* value, two-tailed Mann–Whitney test. Representative immunofluorescence image (**Q**) and quantification (**R**) of αSMA+, and percentage of pSTAT3+αSMA+ cells, presented as average fold change relative to control, in IgG and LIF-nAb treated mice. Scale bar: 50 μm. Arrowheads indicate double positive cells. $n = 5$ mice per group. Error bars, SD. *P* value, two-way ANOVA with Tukey's multiple comparisons. Source data and exact *p* values are provided as a Source Data file.

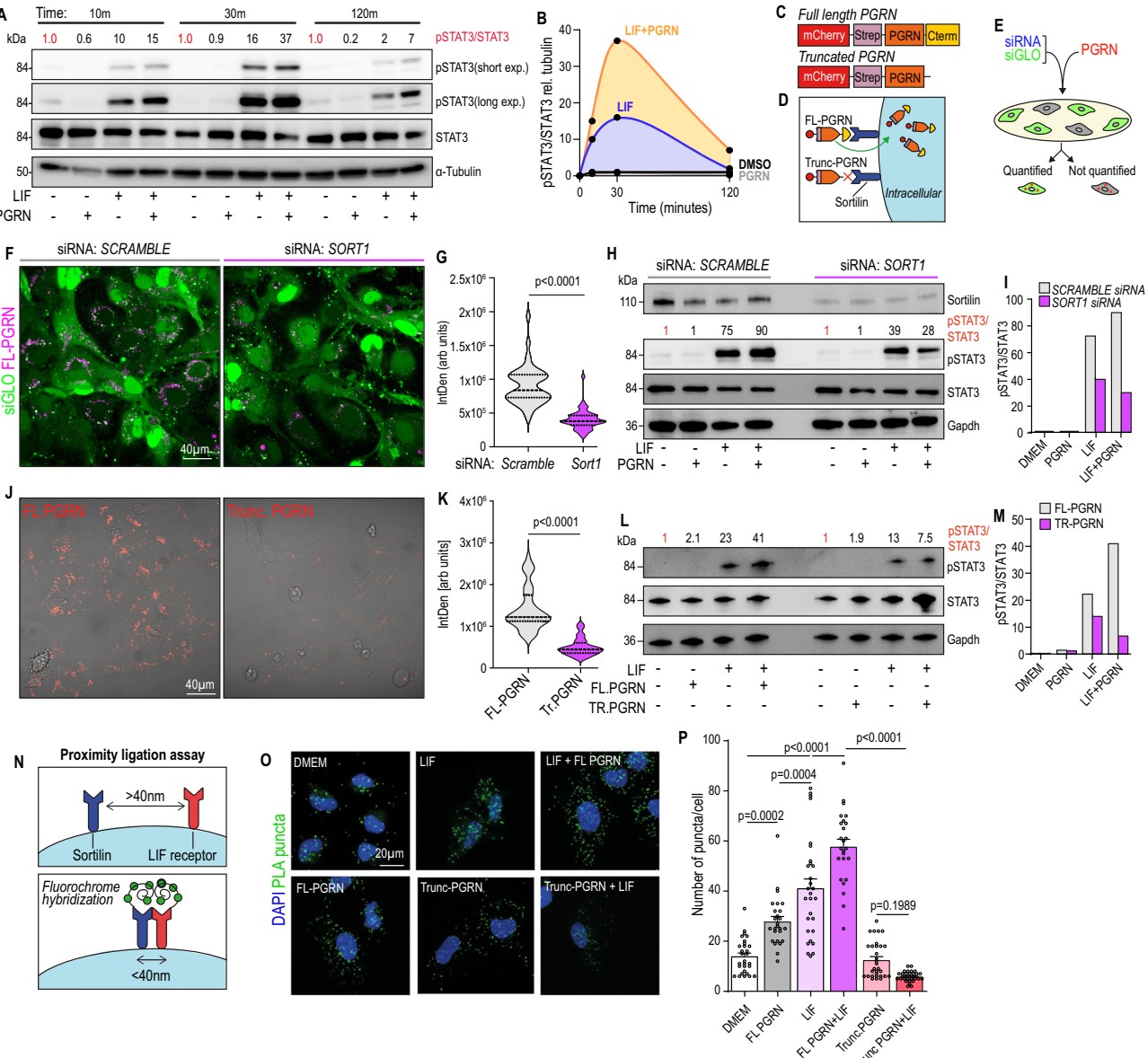

**Fig. 6 | The binding of progranulin to Sortilin enhances Sortilin-LIFR proximity, leading to STAT3 hyperactivation in HStCs.** Immunoblot (**A**) and densitometric quantification (**B**) of pSTAT3/STAT3, relative to α-tubulin. Experiment was performed three times with similar results. Schematic (**C**) of recombinant full length (FL) progranulin, or progranulin lacking the C-term tail (Trunc PGRN) constructs, with an mCherry-StreptagII, and (**D**) illustration depicting Sortilin-mediated uptake of progranulin, via C-terminus end. **E** Schematic of experimental design for pre-treatment of LX2 cells with *SCRAMBLE* or *SORT1* siRNA, and siGLO transfection reagent, prior stimulating with FL-progranulin. Representative images (**F**) and quantification (**G**) of FL-progranulin uptake (FL-PGRN+) in successfully transfected (siGLO+) LX2 cells. Scale bar, 40 μm. siRNA: *SCRAMBLE*, *n* = 48 cells. siRNA: *SORT1*, *n* = 47 cells. Distribution of data presented as violin plot. *P* value, two-tailed Mann–Whitney test. Immunoblot (**H**) and densitometric quantification of pSTAT3/STAT3 relative to GAPDH (loading control) (**I**) in LX2 cells pre-treated with *SORT1* siRNA and stimulated with LIF in the presence or absence of progranulin, for 30 min. Successful receptor knockdown was validated by protein detection of

sortilin. Experiment was performed three times with similar results. Representative images (**J**) and quantification (**K**) of the uptake of FL-progranulin (*n* = 51) and Trunc-progranulin (*n* = 35) (m-cherry; Red) in LX2 cells. Scale bar: 40 μm. Distribution of data presented as violin plots. *P* value, two-tailed Mann–Whitney test. Immunoblot (**L**) and densitometric quantification of pSTAT3/STAT3 relative to GAPDH (**M**) in LX2 cells stimulated with LIF in the presence or absence of FL- or Trunc-progranulin, for 30 min. Experiment was performed three times with similar results. **N** Illustration depicting the mechanism of duolink Proximity Ligation Assay. Two antibody-labelled proteins of interest, Sortilin and LIF receptor, generate a fluorescently detectable signal, visible as a single punctum, only when residing within 40 nm of each other. Representative fluorescence images (**O**) and quantification (**P**) of average number of puncta (PLA+) in LX2 cells stimulated with DMEM (*n* = 29), LIF (*n* = 29), FL-PGRN (*n* = 26), LIF + FL-PGRN (*n* = 25), Trunc-PGRN (*n* = 31), or LIF+Trunc-PGRN (*n* = 32). Scale bar: 20 μm. Error bars, SD. *P* value, one-way ANOVA with Tukey's multiple comparisons. Source data are provided as a Source Data file.

only induced STAT3 hyperactivation in *SCRAMBLE* siRNA, but not *SORT1* siRNA pre-treated LX2 cells (Fig. 6H, I).

To validate the role of progranulin-sortilin binding, LX2 cells were loaded overnight with Trunc-PGRN, lacking the C-terminal end, and mCherry fluorescent intensities quantified. Indeed, in the absence of a functional sortilin-binding domain, the uptake of Trunc-

PGRN was negligible, in comparison to FL-PGRN (Fig. 6J, K). Similarly, co-stimulation of Trunc-PGRN did not hyperactivate LIF-induced STAT3 signalling (Fig. 6L, M), suggesting that the interaction and uptake of progranulin, mediated by C-terminal binding to the sortilin receptor, is essential for the regulation of LIF-induced STAT3 signalling.

Sortilin expression has previously been described to enhance the level of pSTAT3 induced by cytokines of the IL6 family, specifically only those that bind the LIF receptor (LIFR), including LIF, but not IL6[43]. Mechanistically, both sortilin and LIFR have been shown to co-exist in subcellular fractions rich in lipid rafts[43], whereas sortilin has previously been implicated in modulating lipid raft clustering[44], a phenomenon known to compartmentalise and concentrate signalling events[45]. Meanwhile, an interaction between sortilin and LIFR has been demonstrated by both antibody-based proximity-ligation assay (PLA) and surface plasmon resonance, revealing the ectodomain of sortilin interacts with the extracellular domain of LIFR[43]. Accordingly, based on our prior results, we hypothesised that the binding of progranulin to sortilin modulates its proximity with LIFR, which in turn amplifies LIF/LIFR signalling.

To explore this further, we performed in situ visualisation of Sortilin-LIFR proximity, in the presence or absence of LIF and progranulin, using duolink PLA probes that generate a specific fluorescent signal when receptors are present within 40 nm of each other (Fig. 6N). Visualisation of PLA puncta in unstimulated LX2 cells confirmed that sortilin and LIFR exist in close proximity, while induction of STAT3 signalling with LIF increased the number of puncta per cell, suggesting a greater number of sortilin and LIFR proteins residing in close proximity (Fig. 6M, N). Notably, co-stimulation of LIF with FL-PGRN significantly increased the number of detectable puncta, compared to LIF alone, and a phenotype that could not be replicated in the presence of C-terminal truncated PGRN (Fig. 6O, P). Together, these results suggest that the binding of progranulin to sortilin modulates the proximity of sortilin-LIFR, which amplifies JAK/STAT signalling that facilitates the induction of a pSTAT3+myMAF phenotype.

### STAT3 activated myMAFs secrete Osteopontin (*Spp1*), which in turn supports immunosuppressive macrophage functions

Having proposed an underlying molecular mechanism driving pSTAT3+myMAF activation, we next sought to decipher their pro-metastatic functions with the underlying goal of identifying new therapeutic strategies. We went back to our scRNAseq analysis and enriched for up- and down-regulated genes encoding for secreted factors among myMAFs, compared to all MAFs, using the GO term "Extracellular Region" (Fig. 7A). Several ECM associated genes were identified, including *Spp1*, *Col1a1*, *Postn*, and *Tnc*, and their expression in pSTAT3+myMAFs generated in vitro was determined to be STAT3-dependent (Fig. 7B, C).

We and others have previously shown that periostin (*Postn*) promotes pancreatic cancer metastasis by supporting the outgrowth of disseminated cancer cells[22,46], and our prior results confirmed that pSTAT3+myMAFs promoted anchorage-independent outgrowth of cancer cells in vitro (Fig. 5D, E). Therefore, we tested whether periostin was also responsible for these results (Supplementary Fig. 7A).

Indeed, myMAF CM enhanced cancer cell colony formation in a periostin-dependent manner (Supplementary Fig. 7B, C). In support of our in vitro data, both HStC-specific genetic depletion and pharmacological inhibition of STAT3 markedly reduced the number of Ki67+ CK19+ cancer cells in vivo, suggesting that pSTAT3+myMAF-derived factors, such as periostin, mediate cancer cell proliferation in a STAT3-dependent manner (Supplementary Fig. 7D–G).

As key regulators of tumour immunity, macrophages can acquire immunostimulatory or immunosuppressive functions depending on their activation state and external stimuli. Given the dependency of pSTAT3+myMAFs on macrophages, we tested whether, reciprocally, pSTAT3+myMAFs communicate with and shape macrophage functions. To address this, we stimulated primary murine bone marrow-derived macrophages (BMM) with CM from myMAFs, cultured in the presence or absence of a STAT3i to inhibit myMAF conversion, and explored gene expression of a panel of immunostimulatory (*H2-Aa*, *Cxcl10*) and immunosuppressive (*Arg1*, *Chil3l3*, *Mrc1*) markers (Fig. 7D).

CM from pSTAT3+myMAFs significantly upregulated genes associated with an immunosuppressive macrophage phenotype, an induction that was partially suppressed by generating myMAFs in the presence of STAT3i (Fig. 7E). Notably, STAT3i alone also had a direct effect on macrophages, significantly inducing *Cxcl10* expression while suppressing *Arg1* (Fig. 7E). Together, these results suggest that myMAF-derived factors promote an immunosuppressive BMM phenotype in a STAT3-dependent manner.

To identify myMAF derived factors contributing to the observed immunosuppressive macrophage phenotype, we revisited our up and down-regulated genes encoding for secreted factors among myMAFs, compared to all MAFs (Fig. 7A). Encoding osteopontin (OPN), *Spp1* was among the most upregulated genes and emerging experimental data has identified OPN as an immunosuppressive chemokine in glioblastoma-infiltrating macrophages[47]. Furthermore, TCGA analysis confirmed that high OPN expression is associated with reduced overall and disease-free survival in pancreatic cancer (Supplementary Fig. 7H, I). To determine the main source of OPN in vivo and validate STAT3-dependency, we analysed OPN gene expression across MAFs (GFP+), cancer cells (Epcam+), immune cells (CD45+), isolated by flow cytometry-based cell sorting from metastasis bearing *Pdgfrb*-GFP mice treated with STAT3i (Fig. 7F and Supplementary Fig. 7J). We found that in metastatic PDAC tumours, MAFs and cancer cells express high levels of *Spp1*, and that STAT3i reduces *Spp1* expression in both cell types (Fig. 7G). A reduction of OPN and pSTAT3 on protein level from metastatic tumours treated with STAT3i confirmed response to systemic pharmacological administration (Supplementary Fig. 7K–M). Moreover, interrogation of secreted factors in the CM of pSTAT3+myMAFs revealed a marked reduction in OPN secretion when cultured in the presence of STAT3i (Fig. 7H), suggesting that an immunosuppressive macrophage phenotype could be induced by myMAF-secreted OPN.

To directly test its role in macrophage activation, we stimulated BMMs with recombinant OPN and interrogated immunostimulatory and immunosuppressive signatures, as above (Fig. 7I). Indeed, OPN-stimulated BMMs displayed a strong immunosuppressive phenotype, while also downregulating immunostimulatory genes (Fig. 7J). Meanwhile, neutralisation of OPN from pSTAT3+myMAF CM suppressed the induction of an immunosuppressive macrophage phenotype (Fig. 7K, L). Notably, pSTAT3+myMAF-CM induced *Cxcl10* expression in a STAT3-dependent (Fig. 7E), but osteopontin-independent manner (Fig. 7L), suggesting that other unidentified pSTAT3+myMAF-derived factors may have additional immunoregulatory functions.

Since, besides pSTAT3+myMAFs, cancer cells also expressed high levels of OPN (Fig. 7G), we further explored the contribution of KPC-cell derived OPN to promoting an immunosuppressive macrophage phenotype. We found that, as observed in myMAFs, inhibition of STAT3 reduced *Spp1* expression in KPC cells in vitro (Supplementary Fig. 7N). As expected, macrophages exposed to KPC-cell CM, pre-treated with IgG, acquired an immunosuppressive phenotype (Supplementary Fig. 7O, P). Meanwhile, neutralisation of OPN from KPC-CM was sufficient to partially suppress the induction of *Arg1* and *Mrc1* (Supplementary Fig. 7O, P). However, it was notable that these macrophages remained inherently immunosuppressive, suggesting that additional KPC-derived factors, beyond OPN, remain at play. For example, we have previously shown that KPC-derived macrophage colony stimulating factor (mCSF) has a central role in regulating an immunosuppressive phenotype[21].

Taken together, our data reveals an intricate interplay of cancer cells, fibroblasts, and macrophages that suggests the fostering of a hospitable microenvironment conductive to metastatic outgrowth.

### myMAFs orchestrate an immunosuppressive microenvironment in a STAT3-dependent manner in metastatic PDAC

CD8+ T cells are among the main effector cells of an anti-tumour immune response and immunosuppressive *Chil3* + (encoding YM1)

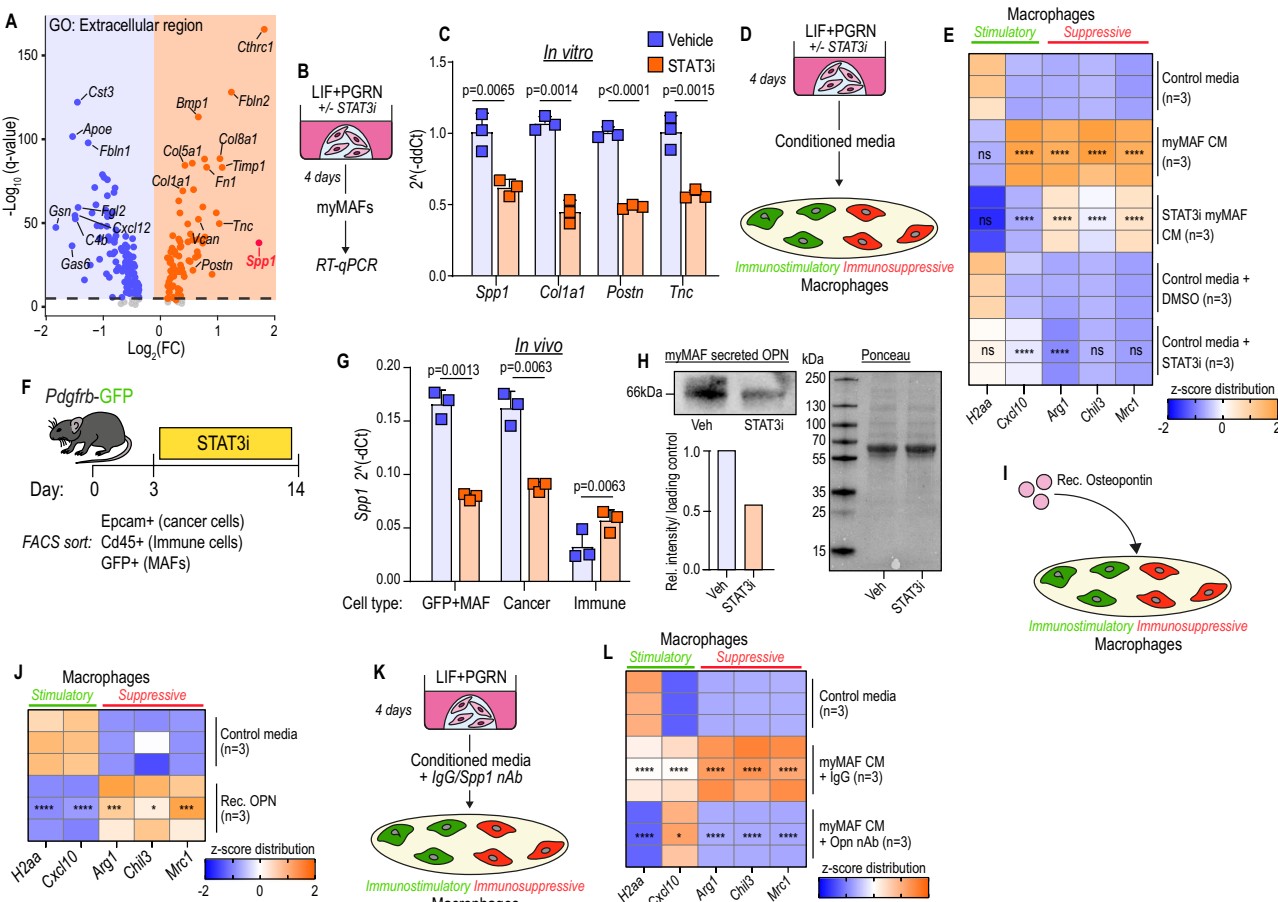

**Fig. 7 | STAT3 activated myMAFs secrete Osteopontin (*Spp1*), which in turn supports immunosuppressive macrophage functions. A** Volcano plot of DEG among myMAFs (vs vMAF;iMAF;cycMAF) enriching for Gene Ontology term: Extracellular region (GO:0005576). Schematic of experimental design (**B**) and qPCR (**C**) of selected upregulated genes identified in **A**. Data is presented as average fold change. *N* = 3 independent experiments. Error bars, SD. *P* value, two-tailed unpaired *t*-test. Schematic of experimental design (**D**) and qPCR (**E**) of immunostimulatory and immunosuppressive markers. Data is presented as a heatmap of z-scores. *N* = 3 independent experiments per group. *P* value, two-way ANOVA with Tukey's multiple comparisons. Significance shown depict: myMAF CM vs Control media; STAT3i myMAF CM vs myMAF CM; Control media + STAT3i vs Control media + DMSO. Schematic of experimental design (**F**) and qPCR (**G**) of relative *Spp1* expression, presented as mean fold change, in sorted cell populations. *N* = 3 independent experiments per group. Error bars, SD. *P* value, Two-way ANOVA with

Tukey's post hoc test. **H** Immunoblot of secreted osteopontin from pSTAT3+my-MAF CM, generated as illustrated in **D**. Below left, densitometric analysis of relative intensity normalised to ponceau stain (right). Experiment was performed three times with similar results. Illustration of the experimental design (**I**) and qPCR (**J**) of immunostimulatory and immunosuppressive markers in BMMs stimulated with recombinant Osteopontin. Data is presented as a heatmap of z-scores. *N* = 3 independent experiments per group. *P* value, two-tailed unpaired *t*-tests. Illustration of the experimental design (**K**) and qPCR (**L**) of immunostimulatory and immunosuppressive markers in BMMs exposed to myMAF CM in the presence or absence of Osteopontin nAb. Data is presented as a heatmap of z-scores. *N* = 3 independent experiments per group. *P* value, one-way ANOVA with Sidak's multiple comparisons. Significance shown depicts: myMAF CM + IgG vs Control media; myMAF CM + Opn nAb vs myMAF CM + IgG. Source data and exact *p* values are provided as a Source Data file.

macrophages have been shown to potently inhibit CD8+ T cell infiltration and their activation status in many cancer types[21,48,49]. To confirm that STAT3+myMAFs promote an immunosuppressive macrophage phenotype in vivo, we analysed the phenotypical status of metastasis associated F4/80+ macrophages, using YM1 as a marker for immunosuppressive phenotype, in metastatic tumours of mice lacking STAT3 expression in HStCs (STAT3^cKO mice) (Fig. 8A). In agreement with previous findings, advanced metastatic tumours derived from mice expressing STAT3 in HStCs (STAT3^WT, Cre-) were rich in YM1+ macrophages (Fig. 8B, C). However, metastatic tumours from mice lacking STAT3 expression in HStCs (STAT3^cKO, Cre+) showed a marked reduction of YM1+ macrophages, while overall macrophage numbers (F4/80+ cells) remained unchanged (Fig. 8B, C). Next, we explored whether a reduction of immunosuppressive YM1+ macrophages in STAT3^cKO mice associated with increased cytotoxic CD8+ T cell infiltration and activation (GzmB+). As expected, advanced metastatic tumours of STAT3^WT control mice were poorly infiltrated with CD8+

T cells, and those few CD8+ T cells displayed a dysfunctional state (GzmB-) (Fig. 8D–F and Supplementary Fig. 8A). On the contrary, in metastatic tumours of mice deficient in HStC STAT3 (STAT3^cKO), we observed an increase in the number of infiltrating CD8+ T cells and an increase in their activation state (GzmB+). Thus, our data suggests that genetic depletion of STAT3 in HStCs is sufficient to inhibit myMAF activation and is associated with an improved immunostimulatory anti-tumour response in PDAC liver metastasis (Fig. 8D–F).

As a more translational approach, we analysed metastatic tumours from mice treated with pharmacological STAT3i (Fig. 8G). Confirming our previous observations, while overall macrophage numbers (F4/80+) remained unaffected by STAT3i, the administration of STAT3i markedly reduced the presence of YM1+ immunosuppressive macrophages (Fig. 8H, I), and CD8+ T cell infiltration and activation (GzmB+) was significantly increased (Fig. 8J, K and Supplementary Fig. 8B). Flow cytometry analysis of metastasis infiltrating CD8+ T cells derived from STAT3i treated mice revealed a significant increase in

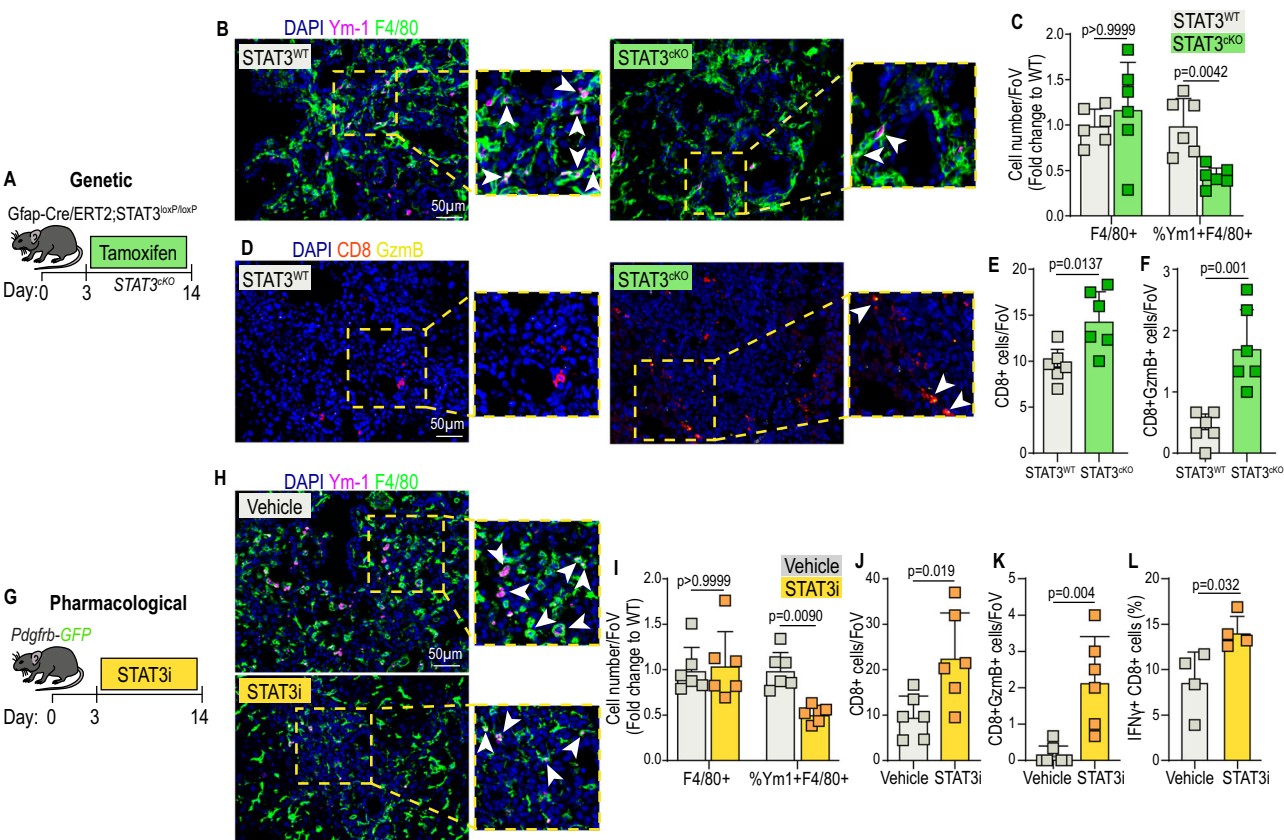

**Fig. 8 | myMAFs orchestrate an immunosuppressive microenvironment in a STAT3-dependent manner in metastatic PDAC. A** Experimental design for generating metastasis bearing GFAP-STAT3 conditional knockout mice (STAT3cKO). **B** Representative immunofluorescence images of immunosuppressive (Ym-1+) macrophages (F4/80+) in metastatic tumours of STAT3WT and STAT3cKO mice. Arrowheads indicate double-positive cells. Scale bar, 50 μm. **C** Quantification of F4/80+, and percentage of Ym-1 + F4/80+ cells, presented as average fold change relative to STAT3WT. *n* = 6 mice per group. Error bars, SD. *P* value, two-way ANOVA with Tukey's multiple comparisons. **D** Representative immunofluorescence images of cytotoxic (Gzmb+) T cells (CD8+) in metastatic tumours of STAT3WT and STAT3cKO mice. Arrowheads indicate double-positive cells. Scale bar, 50 μm. Quantification of total infiltrating (**E**) and cytotoxic (**F**) CD8+ cells, presented as average cells per field of view. *N* = 6 mice per group. Error bars, SD. *P* value, two-tailed unpaired *t*-test. **G** Experimental design for pharmacological inhibition of pSTAT3 with Silibinin (STAT3i), in *Pdgfrb*-GFP mice. **H** Representative immuno-fluorescence images of immunosuppressive (Ym-1+) macrophages (F4/80+) in metastatic tumours of vehicle or STAT3i-treated mice. Arrowheads indicate double-positive cells. Scale bar, 50 μm. **I** Quantification of F4/80+, and percentage of Ym-1 + F4/80+ cells, presented as average fold change relative to vehicle control. *n* = 6 mice per group. Error bars, SD. *P* value, two-way ANOVA with Tukey's multiple comparisons. Quantification of total infiltrating (**J**) and cytotoxic (**K**) CD8+ cells, presented as average cells per field of view. *N* = 6 mice per group. Error bars, SD. *P* value, two-tailed unpaired *t*-test. **L** Percentage of IFNγ+ cells, among CD8+ T cells, determined by flow cytometry analysis. Data is presented as mean from *n* = 4 mice per group. Error bars, SD. *P* value, two-tailed unpaired *t*-test. Source data are provided as a Source Data file.

IFNγ expression, compared to control mice, suggesting increased cytotoxic activity (Fig. 8L & Supplementary Fig. 8C).

In summary, our data suggests that the pathogenic crosstalk between cancer cells, macrophages, and fibroblasts is conductive to metastatic outgrowth of PDAC in the liver (Fig. 9). Macrophages and cancer cells promote pSTAT3+myMAF activation, via progranulin and LIF mediated JAK/STAT signalling. In turn, myMAF-secreted periostin directly promotes cancer cell proliferation, whereas myMAF and cancer-cell secreted Osteopontin fosters an immunosuppressive macrophage phenotype, thereby curbing an efficient anti-tumour immune response and permitting metastatic outgrowth (Fig. 9).

## Discussion

Herein, we employed scRNAseq to uncover the crosstalk between macrophages and fibroblasts driving metastatic outgrowth of pancreatic cancer. To achieve this, we utilised the *Pdgfrb*-GFP reporter mouse model to efficiently label and sequence the hepatic mesenchyme in tumour bearing livers, in the presence or absence of macrophages. Our transcriptomic analysis revealed a distinct population of cells that expressed common markers of fibroblasts, which we termed

as metastasis associated fibroblasts (MAFs) since these cells were only present in tumour bearing livers, and not in healthy control livers. The MAF cluster comprised four distinct populations of vMAFs (42%), myMAFs (43%), iMAFs (9.5%), and cycMAFs (5.5%), with myMAFs and iMAFs strongly aligning with myCAF and iCAF phenotypes previously identified at the primary site[12,13,18].

Tissue-resident mesenchymal cells are the primary sources of CAFs and MAFs, therefore it is essential to consider organ and lineage-dependent heterogeneity when defining diversity and function. In support of previous observations, our transcriptomic and histological analysis identifies HStCs, as the main source of MAFs in PDAC liver metastasis[15]. Our data indicates that vMAFs and myMAFs are both of HStC lineage, suggesting divergence into two distinct HStC populations that may regulate tumour vascularisation and extracellular matrix composition. In healthy livers, HStCs partition into two topographically diametric lobule regions, comprising portal vein-associated HStCs (paHStC) and central vein-associated HStCs (cvHStC), with the latter giving rise to myofibroblasts in a model of centrilobular liver injury[19]. Similarly, the divergence of HStCs into two distinct populations of cytokine-expressing HStCs (cyHStCs) and

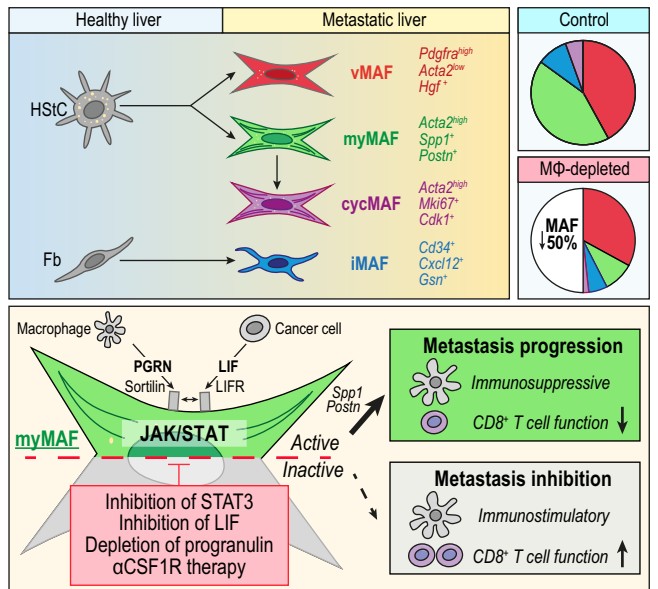

**Fig. 9 | Schematic depicting MAF diversity and the pro-metastatic functions of pSTAT3+myMAFs.** Illustration depicting the proposed cellular origin, subtyping, distribution, and mechanism of myMAF activation. In established liver metastatic PDAC, HStCs give rise to vMAFs, myMAFs, and cycMAFs, while portal fibroblasts give rise to iMAFs. In tumour bearing control mice, vMAFs and myMAFs are dominant, whereas iMAFs and cycMAFs are minor populations. In macrophage depleted mice (αCSF1R-treated) an overall reduction in fibrosis is primarily driven by the loss of myMAFs, resulting in an imbalance of vMAF/myMAF ratio. Mechanistically, progranulin and LIF, mainly derived from macrophages and cancer cells, respectively, co-opt to promote a myMAF phenotype via activation of JAK/STAT signalling. Progranulin binding to sortilin enhances the proximity of sortilin to LIFR, leading to JAK/STAT hyperactivation. Reciprocally, myMAF secreted periostin directly promotes cancer cell proliferation, whereas myMAF secreted osteopontin promotes an immunosuppressive macrophage phenotype. Pharmacological or genetic interference of cancer cell-macrophage-MAF crosstalk ablates the metastasis promoting functions of pSTAT3+myMAFs in metastatic PDAC.

myofibroblastic HStCs (myHStCs) has been recently described in hepatocellular carcinoma (HCC)[50]. cyHStCs-derived *Hgf* is tumour-restraining, while myHStCs, through increased tissue stiffness, are tumour-promoting[50]. Interestingly, we show that in PDAC liver metastasis *Hgf* is highly expressed by vMAFs, and we observe an increased vMAF/myMAF ratio in response to macrophage depletion. Further investigation is required to assess whether vMAF-derived *Hgf* also has a tumour-restraining function and whether topographically separated HStC populations give rise to distinct MAF and CAF subpopulations.

Using the *Pdgfrb*-GFP reporter mouse model, we further identify that the iMAF subtype derives from portal fibroblasts (Fig. 9), while in a recent transcriptional analysis using an *Lrat*-tdTomato reporter mouse model, iMAFs were identified among HStCs[15]. These discrepancies are likely explained by the use of different reporter mouse models, whereas the *Lrat*-tdTomato mouse model is restricted to HStC labelling, the *Pdgfrb*-GFP reporter model encapsulates a broader spectrum of different cell types, including HStCs, portal fibroblasts, and VSMCs[15,19]. Our findings are further supported by recent evidence from the primary PDAC site indicating that pancreatic iCAFs, similar to what we observed with iMAFs, can also arise from a pool of resident fibroblasts[17]. However, whether pancreatic and hepatic fibroblasts give rise to similarly functioning iCAF and iMAF populations warrants further investigation. While myMAF and iMAF clusters strongly aligned with myCAF and iCAF phenotypes identified at the primary site of PDAC, vMAFs did not. Further investigation is necessary to determine whether vMAFs are uniquely present in PDAC liver

metastasis, or whether vMAFs are also detectable in other organs or cancer types.

The plasticity of distinct CAF and MAF subtypes is a question often debated and our data suggests the possibility that, based on their distinct cellular origin, fibroblasts activate linearly into iMAFs, whereas vMAFs and myMAFs are of HStC origin and could exist on a spectrum of interconvertible states driven by external stimuli. In further support of this statement, we observe that distinct MAF phenotypes do not express unique markers, but rather that gradated expression of gene signatures results in, overall, enrichment within populations. Accordingly, gradated expression could indicate that MAF phenotypes, within their lineages, are interconvertible and that cells at the interphase, which co-express markers of both subtypes, are in an intermediate state. In support of this statement, plasticity of PStC-CAFs has also been highlighted both in vitro and in vivo[8,13,18]. Further, our findings agree with other recently published data showing that the use of advanced technologies, such as scRNAseq or multiplexed mass cytometry, permits the identification of MAF and CAF populations based on their transcriptional phenotype or expression levels of a combination of markers[12,28], whereas basic immunofluorescence can be applied, where appropriate, with surrogate markers for identifying distinct populations.

Previously, we uncovered a significant role for macrophages, through secretion of progranulin, in regulating fibrosis and ECM deposition at the metastatic site[21,22]. Our transcriptomic data consolidate these findings, revealing an intricate co-dependent relationship between macrophages and ECM-producing myMAFs within the TME. Using genetic and pharmacological approaches, we reveal that active JAK/STAT signalling, driven by complementarity of progranulin and cancer cell-derived LIF, promotes myMAF activation and function. STAT3 inhibition, like macrophage, progranulin, or LIF depletion, reduces myMAF activation and leads to significant stromal remodelling. We previously identified periostin as a macrophage-induced HStC-derived factor that drives cancer cell proliferation[22]. This study provides further molecular insight into the regulation of periostin expression and reveals that periostin expression is driven by myMAFs, in a macrophage and STAT3 dependent manner.

Mechanistically, we observe that Progranulin-Sortilin binding is implicated specifically in LIF-LIFR mediated JAK/STAT signalling to induce a myMAF phenotype. Our results did not extend to IL6, which instead activates JAK/STAT by binding to IL6R. Notably, the addition of IL6 to progranulin enhanced the expression of *Spp1* but did not recapitulate a myMAF phenotype. Although we did not observe abundant expression of IL6 by cancer cells in our model, the level of IL6 in the blood of advanced PDAC patients is high[51], which suggests that circulating IL6 could also contribute to the expression of Osteopontin (*Spp1*) and, accordingly, the pro-metastatic functions of myMAFs.

The binding of progranulin to sortilin has been extensively characterised, and progranulin is shown to induce a unique conformational change that stabilises the membrane-proximal 10CC-b domain[52]. This conformational change is dependent on the C-terminal end of progranulin, which we observe to also be necessary for increased STAT3 activation. An interaction between Sortilin and LIFR has previously been described[43], and both receptors are known to co-exist in subcellular fractions rich in lipid rafts[43], whereas sortilin has previously been described to modulate lipid raft clustering[44], a phenomenon known to compartmentalise and concentrate signalling events[45]. While our results further support an interaction and reveals the binding of progranulin contributes to Sortilin-LIFR proximity, further investigation is necessary to discern whether this is a direct or indirect interaction.

The JAK/STAT pathway has previously been associated with driving a tumour-promoting iCAF phenotype at the primary site of PDAC, and in vitro studies have identified autocrine-LIF and paracrine-OSM as

critical mediators of this phenotype[18,53]. Meanwhile, our data supports that progranulin/LIF/STAT3 signalling promotes a myMAF phenotype at the metastatic site. Accordingly, our findings highlight that the cellular composition of fibroblasts differs between the primary and metastatic TME and that STAT3-mediated signalling leads to different fibroblast subtypes that are dependent on their local microenvironment and cellular origin. In the liver, JAK/STAT has also been shown to play a critical role in the pathogenesis of fibrosis and inhibition of STAT3, including exosome delivery of STAT3 siRNA selectively into myofibroblasts, inhibits fibrosis[54]. These results indicate that inhibition of JAK/STAT will disrupt the pro-tumorigenic functions of both pancreatic iCAFs and hepatic myMAFs, highlighting its potential as a systemic treatment strategy for metastatic PDAC patients. Beyond PDAC, tumour-promoting functions of STAT3 signalling in fibroblasts has also been described in other organs. Tissue resident fibroblasts in the brain, known as reactive astrocytes, have recently shown to promote breast cancer brain metastasis in a STAT3-dependent manner[37]. A phase 2 trial evaluating the efficacy of STAT3i (silibinin) in preventing brain metastasis recurrence in lung and breast cancer patients is currently ongoing (NCT05689619).

Beyond tumour promotion through direct cancer cell interaction, myMAFs are critical mediators of stromal architecture, known to be essential for immune cell trafficking[55]. Collagen deposition has been shown to directly associate with tumour stiffness, which enhances tumour growth directly through activation of mechanosensitive pathways, or indirectly through inhibition of CD8$^+$ T cell infiltration[21,50]. A concomitant increase in tumour vascularisation may permit increased cellular trafficking and sustain a biomechanical mechanism for enhanced CD8$^+$ T cell infiltration. Thereby, whether vMAFs, which dominate the fibroblast landscape in macrophage depleted tumours, also play a role in promoting or sustaining enhanced vascularisation warrants further investigation. Previous efforts to enhance CD8$^+$ T cell infiltration have been largely ineffective due to persistent dysfunction within the TME[21]. Our data suggests that pSTAT3+myMAFs not only sustain stromal architecture but are also immune-regulatory. Through secretion of Osteopontin, both pSTAT3+myMAFs and cancer cells induce an immunosuppressive macrophage phenotype that is known to potently suppress cytotoxic T cell activity[21,55]. Increased infiltration and activation of CD8$^+$ T cells in STAT3 inhibited tumours, in comparison to genetic inhibition of pSTAT3+myMAFs alone, suggests an additive effect, likely mediated by the additional suppression of cancer-cell derived Osteopontin.

It is important to also note that although pSTAT3+myMAF-educated macrophages were overwhelmingly immunosuppressive, we did also observe an induction of *Cxcl10* expression, which has been implicated in promoting anti-tumour immunity and efficacy of immune checkpoint therapy (ICT)[56]. Accordingly, our data suggests that pSTAT3+myMAFs may have additional immunoregulatory functions beyond Osteopontin. Further investigation is necessary to better understand and develop strategies to selectively harness functions that may be tumour-restraining, while inhibiting tumour-promoting mechanisms. Beyond the metastatic site, Osteopontin is highly expressed within myCAF populations of numerous scRNAseq datasets, including recently defined immunoregulatory Lrrc15+ myCAFs[13,27]. While we could not detect *Lrrc15* within our dataset, several genes including *Cthrc1*, *Acta2*, and *Postn* are conserved across both pSTAT3+myMAFs and Lrrc15+ myCAFs. Whether myCAF-derived Osteopontin also regulates macrophage function at the primary site requires further study.

Although we see increased levels of active CD8+ T cells in both HStC-STAT3$^{cKO}$ and STAT3i tumours, the majority of CD8+ T cells remain dysfunctional. Recent efforts combining ICT with stromal reprogramming has shown promising results in preclinical mouse models of PDAC[30,57]. However, the JAK/STAT pathway also plays a central role in type 1 interferon-driven induction of T cell cytotoxicity[58], rendering STAT3i largely incompatible in combination with ICT. Accordingly, our data supports several alternative approaches for inhibiting immune-modulatory pSTAT3+ myMAFs, including progranulin, LIF, and osteopontin, which warrants further investigation for combination with ICT to further harness and unleash the anti-tumour immune response.

The limitations of our study include the utilisation of an experimental metastasis model to conduct the initial scRNA sequencing in *Pdgfrb*-GFP reporter mice. To address this, we have validated our observations in spontaneous liver metastases derived from autochthonous KPC mouse and patient derived biopsies. The utilisation of αCSF1R therapy is a pan-macrophage depletion approach, inhibiting macrophages that are both embryonically developed (Kupffer cells) and recruited (bone marrow-derived macrophages). Thus, our approach does not discern the origin or subtype of fibrosis-regulating macrophages. Additional studies are needed to specifically target macrophage subpopulations.

Together, our data reveals molecular and cellular insights into MAF heterogeneity, describes a complex interaction between cancer cells, macrophages, and fibroblasts, and identifies several druggable targets that could be further developed to treat metastatic PDAC.

## Methods
### Ethics statement
This study complies with all relevant ethical regulations. Studies involving the use of liver metastasis biopsy and blood samples from patients with treatment-naive, advanced PDAC were accessed using the PINCER platform study, approved by the National Research Ethics Service Committee North West, Greater Manchester REC15/NW/0477. All individuals provided written informed consents on approved institutional protocol. All animal studies were conducted in accordance with UK Home Office regulations under project license P16F36770. The maximum tumour burden limit of 1.5 cm mean diameter was not exceeded in the studies. In all animal studies, the severity was limited to moderate.

### Clinical samples
Human studies using liver biopsy samples were approved by the National Research Ethics (NRES) Service Committee North West – Greater Manchester REC15/NW/0477. Liver biopsies were collected from patients with treatment-naïve, advanced PDAC with pathologically confirmed liver metastasis. All individuals provided informed consents for tissue donation on approved institutional protocol.

### Cell lines and cell culture
Murine pancreatic cancer cells KPC FC1199, from here on referred as KPC, were generated in the Tuveson laboratory (Cold Spring Harbor Laboratory, New York, USA) isolated from PDAC tumour tissue of Kras$^{G12D/+}$; p53$^{R17H/+}$; Pdx1-Cre mice of a pure C57BL/6 background and authenticated as previously reported[24]. KPC$^{Luc/ZsGreen}$ cells were generated by using pHIV Luc-ZsGreen (gift from B. Welm, University of Utah, Salt Lake City, UT; Addgene plasmid no.39196) through lentivirus infection. Infected cells were selected for high ZsGreen expression levels using FACSAria III cell sorter (Becton Dickinson). HEK293T cells (CRL-3216) were purchased from ATCC. The human hepatic stellate cell line LX2 was kindly provided by J. Mann's laboratory, University of Newcastle, UK. All cells were maintained in DMEM supplemented with 10% FBS and antibiotic antimycotic solution (10 U/mL penicillin, 0.1 mg/mL streptomycin, and 0.25 ug/mL amphotericin B) (Sigma) and routinely tested negative for the presence of mycoplasma contamination. The cell lines used in this article are not listed in the International Cell Line Authentication Committee and National Center for Biotechnology Information Biosample database of misidentified cell lines.

## Mouse strains

All animal procedures were conducted in accordance with the UK Home Office regulations under the project license P16F36770 (M.C. Schmid). Mice were housed under specific-pathogen-free conditions at the Biomedical Science Unit at the University of Liverpool. Mice were housed under 12 h dark/light cycle, 20–24 °C, and 45–65% relative humidity. Mice were maintained with environmental enrichment, access to standard chow and water ad libitum. At experimental end points, all mice were euthanized by Schedule 1 method of cervical dislocation. C57BL/6 mice were purchased from Charles River Laboratories. Grn$^{-/-}$ mice (B6(Cg)-$Grn^{tm1.1Aidi}$/J) were purchased from the Jackson Laboratory. $Pdgfrb$-GFP mice on the C57BL/6 genetic background was kindly gifted by Prof. Neil C. Henderson, Edinburgh. STAT3$^{cKO}$ mice, housed at the CNIO (Madrid), were generated by breeding $GFAP$-$Cre/ERT2$ (B6.Cg-Tg(GFAP-Cre/ERT2)505Fmv/J; 012849, Jackson laboratory) with STAT3$^{loxP/loxP}$ mice. All animal experiments with the STAT3$^{cKO}$ mice were performed at the CNIO (Madrid) and in accordance with a protocol approved by the CNIO, Instituto de Salud Carlos III and Comunidad de Madrid Institutional Animal Care and Use Committee. For all animal studies utilising transgenic mouse models, both female and male mice aged 6–8 weeks old were used. The pharmacological STAT3 inhibitor (STAT3i) survival study used only female mice of 6–8 weeks old.

## Study design

Pancreatic cancer does not disproportionally affect males or females; therefore, sex and gender of patients or mice was not considered in the study design. Accordingly, no sex or gender analysis was performed.

## Autochthonous KPC model

KPC on a mixed background were bred in-house at the CRUK Beatson institute and maintained in conventional caging with environmental enrichment, access to standard chow, and water ad libitum. All animal experiments were performed under a UK Home Office Licence and approved by the University of Glasgow Animal Welfare and Ethical Review Board. Genotyping was performed by Transnetyx (Cordoba, TN, USA). Tissues were harvested at humane timepoint and analysed at the University of Liverpool.

## Experimental liver metastasis

Liver metastasis was induced by implanting $1 \times 10^6$ KPC cells in 25 μL of PBS into the spleen of immunocompetent syngeneic C57BL/6 mice using a Hamilton 29 G syringe, as previously described[22]. Metastatic tumour burden was measured by histological interrogation.

## In vivo drug treatments

For macrophage depletion, αCSF1R neutralising antibody (BioXCell, BE0213, Clone AFS98) was administered by intraperitoneal injection (1 mg) from day 7, and every 48 h (400 μg) into $Pdgfrb$-GFP mice. Control mice received IgG alone (BioXCell, BE0089 Clone 2A3). For STAT3 inhibition studies, Silibinin (Sigma, S0417) was administered at 200 mg/Kg dose via oral gavage once daily, from day 3. Control mice received only vehicle of 0.5% (w/v) Carboxymethyl cellulose and 0.025% Triton X-100. Mice were treated up until day 14. For generating STAT3$^{cKO}$ mice, Tamoxifen was administered by intraperitoneal injection at 1 mg/mouse, starting from day 3 post KPC-implantation every 48 h. Tamoxifen was administered to both control group (Cre-) and cKO group (Cre+). AZD7507 (CSF1Ri) was administered at 100 mg/kg twice daily as previously described[23]. LIF neutralising antibody (AF499, R&D Systems), or IgG control (AB-108-C, R&D Systems) was administered by intraperitoneal injection at 0.5 mg/Kg, diluted in PBS, starting from day 5, every 48 h, up until day 14.

## Bone marrow transplantation

Bone marrow transplantation was performed by injection of $5 \times 10^6$ bone marrow cells isolated from donor WT or Grn$^{-/-}$ mice into the tail vein of recipient $Pdgfrb$-GFP mice that had received a lethal dose of irradiation (10 Gy). After 4 weeks, successful engraftment was assessed by genomic DNA PCR according to The Jackson Laboratory protocol on peripheral blood cells from fully recovered bone marrow transplanted mice. After confirmation of successful bone marrow reconstitution, mice were enroled in tumour studies and implanted with KPC cells via intrasplenic injection.

## Liver cell isolation

Single-cell suspensions from murine livers were prepared by mechanical and enzymatic disruption with 1 mg/mL Collagenase P (Roche) in Hanks Balanced Salt Solution (HBSS) at 37 °C for 30–40 min. Cells were then incubated with 0.05% trypsin at 37 °C for 5 min. After removal of debris by filtering cell suspension through a 70 μm strainer, red blood cells were removed using RBC Lysis Buffer (Biolegend) as previously described[21,22,25].

## Flow cytometry and cell sorting

Single-cell suspensions from murine livers were prepared as outlined above, followed by resuspension in MAC buffer (0.5% BSA, 2 mM EDTA, PBS). Cells were blocked for 10 min on ice with FC block (CD16/CD32, BD Biosciences), and then stained with Sytox-blue viability marker (Thermo Fisher) and fluorophore-conjugated antibodies (Biolegend), listed in Supplementary Table 1.

To assess IFNγ expression levels in metastasis-derived CD8+ T cells, magnetically isolated CD8a+ T cells from metastatic livers were stimulated with 50 ng/mL phorbol 12-myristate 13-acetate (Sigma Aldrich) and 1 μg/mL of ionomycin (Sigma Aldrich) for 5 h at 37 °C in the presence of brefeldin A (eBioscience; 1:100). For intracellular staining, cells were first fixed (eBioscience, IC fixation buffer) and permeabilized (eBioscience, 1× permeabilization buffer). Cells were stained with fluorophore-conjugated IFNγ antibody (Biolegend, clone XMG1.2), CD8 antibody (biolegend) and LIVE/DEAD™ fixable Aqua Dead Cell Stain Kit (Thermo Fisher).

Flow cytometry was performed on a FACS Canto II (BD Biosciences), and fluorescence- activated cell sorting (FACS) was carried out using FACS Aria IIIu (BD Biosciences). Cells were sorted directly into RLT buffer + β-mercaptoethanol according to the manufacturers instruction for RNA isolation (Qiagen).

## Generation of primary murine macrophages and hepatic stellate cells

Primary murine macrophages were generated by flushing the bone marrow from the femur and tibia of C57BL/6 mice followed by incubation for 5 days in DMEM containing 10% FBS and 10 ng/mL murine M-CSF (Peprotech). Primary hepatic stellate cells (HStCs) were isolated from C57BL/6 mice by retrograde hepatic perfusion and in vitro mechanical digestion with a cocktail of pronase (Sigma, P5147), collagenase D (Roche, 110888820001), and DNAse1 (Roche, 10104159001), as previously described[59]. HStCs were selected by 11.5% Optiprep (Sigma, D-1556) density centrifugation and further enriched by culturing in DMEM containing 10% FBS on plastic dishes for 48 h, before trypsinisation and 3D embedding growth factor reduced matrigel (Corning) and culturing in 2% FBS + DMEM in the presence of outlined stimuli.

## Cell stimulations

Where indicated, cells were treated with 100 nM calcipotriol (Stratech), 1 ng/mL murine (Sigma Technologies, L5158) or human (Peprotech, 300-05) LIF, 1 ng/mL murine IL6 (216-6, Peprotech), 2 μg/mL murine (Adipogen Life Sciences, AG-40A-0189Y-C050) progranulin, 2 μg/mL human mCherry-StrepTagII-PGRN (in house) or mCherry-

StrepTagII-PGRN-Truncated (in house), or 10 ng/mL murine Osteopontin (R&D Systems, 441-OP-050). For in vitro studies, STAT3i (Silibinin) was dissolved in DMSO and used at 100 µM.

## Generation of myMAF conditioned media

3D cultured primary hepatic stellate cells were generated as above and polarised into myMAFs with 1 ng/mL of LIF, or cancer cell CM, and 2 µg/mL Progranulin. For some studies, myMAFs were generated in the presence of Silibinin, or vehicle. After 4 days, media was discarded and replaced with 2% FBS DMEM. After 24 h, the media was collected and spun at 1200 RPM for 5 min to remove cellular debris. For periostin (AF2955 – R&D Systems) and osteopontin (AF808 – R&D Systems) neutralisation, myMAF CM was generated as above and incubated with 1 µg/mL neutralising antibody, or IgG control, for 1 h at 37 °C prior exposure to primary macrophages.

## Generation of cancer cell conditioned media

Cancer cells were seeded in a 2D monolayer at 80% confluence, and the next day washed with PBS, and media replaced with 2% FBS DMEM. After 24 h, the CM was collected and spun at 1200 RPM for 5 min to remove cellular debris. For experimental use, cancer cell CM was diluted 20:1 in all assays. For LIF (AF499 – R&D Systems) neutralisation, cancer cell CM was generated as above and incubated with 1 µg/mL neutralising antibody, or IgG control, for 1 h at 37 °C prior experimental use.

## Generation of tumour-educated macrophage conditioned media

Primary bone-marrow derived macrophages (BMDMs) from WT and $Grn^{-/-}$ mice were isolated as described above. After 5 days exposure to m-CSF, BMDMs were washed with PBS and exposed to cancer cell CM (generated as described above), supplemented with fresh 2% FBS, for 24 h. Media was aspirated and cells were washed with PBS twice, prior to addition of fresh 2% FBS DMEM. After 24 h, CM was collected and spun at 1200 RPM for 5 min to remove cellular debris. For experimental use, macrophage CM was utilised at 95% concentration.

## Immunostimulatory & immunosuppressive education of macrophages

BMMs were generated as described above and exposed to the described stimuli, as outlined in the experimental illustrations within the figures, for an 8-h period prior lysing in RLT buffer for interrogation of immunostimulatory and immunosuppressive markers by qPCR.

## In vitro colony formation assay

A 24-well plate was coated with a layer of 0.5% agar solution made of phenol-free DMEM without FBS. KPC$^{luc/zsGreen}$ cells were embedded at a concentration of 10,000 cells/well in a 0.3% agar mix consisting of myMAF CM supplemented with 2% FBS. Following polymerisation, a layer of CM was added on top of the agar. Colony quantification was performed after 14 days by measuring bioluminescence signal (IVIS, Perkin Elmer) following the addition of 150 µg/mL of Beetle Luciferin solution (Promega).

## siRNA knockdown

LX2 cells were transfected using 2.5 µL/mL DharmaFECT 1 transfection reagent following manufacturers protocol with either 50 nM *SCRAMBLE* ON-TARGETplus Non-targeting Pool, ON-TARGETplus Human *SORT1* siRNA, ON-TARGETplus Mouse *Il6ra* siRNA SMARTPool or siGLO Green Transfection Indicator (Dharmacon). Cells were used for signalling or protein uptake studies 48 h post transfection.

## Cloning

To generate mCherry-StreptagII-hPGRN and pHIV-mCherry-StrepTagII-hPGRNΔ5aa, the insert DNA fragments, mCherry and hPGRN, was prepared using KOD polymerase. Fragment 1 was amplified from LC3-EGFP-mCherry plasmid using mCherry-hPGRN-F1.F (CAGGGCTGGTGGCTGGAACAGCTAGCATGGTGAGCAAGGGCGAGGAG) and mCherry-hPGRN-F1.R (GGTGGCTCCAACTACCTCCACCACCGCTGGATCCCTTGTACAGCTCGTCCATGCCG) primers. Fragment 2 was amplified from hPGRN cDNA ORF Clone (SinoBiological, cat#HG10826-M) using mCherry-hPGRN-F2.F (CCAGCGGTGGTGGAGGTAGTTGGAGCCACCCCCAGTTCGAGAAAGGATCCACGCGGTGCCCAGATGGTCA) combined with mCherry-hPGRN-F2.R (GATCGAGGTCGACGGTATCGATTCACAGCAGCTGTCTCAAGGCTGG) for full length PGRN construct or mCherry-hPGRNΔ5aa–F2.R GATCGAGGTCGACGGTATCGATTCAGGCTGGGTCCCTCAAAGGGGC for truncated PGRN. Destination vector pHIV-Luciferase (Plasmid #21375) was double digested using NheI and ClaI enzymes (NEB) in Cutsmart Buffer for 1 h at 37 °C. Amplification and digestion were confirmed by resolving DNA on 1% agarose gel in 1xTAE buffer. Fragments were extracted from the gel and purified using Wizard SV Gel and PCR Clean-Up System (Promega) following manufacturer's protocol. The SLiCE Extract and 10× Slice Buffer was prepared following protocol previously described by Zhang et al.[60]. SLiCE reaction was set up by mixing 1 ul of 10× SLiCE buffer, 1ul SLiCE extract, 50 ng of linearised vector DNA and both DNA inserts in molar ratio 10:1. Reaction was mixed and incubated for 1 h at 37 °C. 2 ul of SLiCE reaction mix was used for transformation of One Shot® TOP10 Competent Cells (Thermo Fisher) following manufacturer's protocol. Transformed cells were plated on agar containing Ampicillin and incubated overnight at 37 °C. 10 randomly selected colonies were screen by PCR using MyFi DNA Polymerase (Bioline) to confirm the insertion of both fragments. Positive clones were used to inoculate Ampicillin containing LB and cultured overnight at 37 °C with shaking. Bacterial cultures were then pelleted by centrifugation at $4300 \times g$, 10 min and plasmid DNA was isolated using QIAprep Spin Miniprep Kit (Qiagen).

## Lentivirus production

Plasmids were purified using Endotoxin-Free Maxiprep Kit (Qiagen). HEK293T cells were seeded at $3.8 \times 10^6$ cells per plate in DMEM complete in 10 cm tissue culture plates and incubated at 37 °C, 5% CO$_2$ overnight. The next morning, media was replaced with fresh complete DMEM containing 25 µM chloroquine diphosphate and incubated for 5 h. 27 µg of total plasmid DNA as a mixture of the transfection plasmids RRE:VSVG:REV:pHIV in ratio of 2:1:2:4 was diluted in 500 µl of OptiMEM (Gibco) and mixed with jetPEI (Polyplus)-OptiMEM (µg DNA:µg PEI – 1:3). After 20 min incubation transfection mix was added to 293 T packaging cells, incubated overnight and media replaced the next morning. Virus was harvested at 48 and 72 h, centrifuged at $1000 \times g$ for 5 min and filtered through a 0.45 µm PES filter, pooled and added to HEK-293T cells. The media was replaced after 48 h and cells were cultured prior to reaching confluency. Stable expression of recombinant proteins was confirmed by fluorescent microscopy and immunoblotting.

## Recombinant protein production and purification

HEK-293T cells stably expressing pHIV-mCherry-StrepTagII-hPGRN or pHIV-mCherry-StrepTagII-hPGRNΔ5aa were grown to confluency in T-175 flasks in complete DMEM and overnight media was harvested, and filtered through a 0.45 µm PES filter and pre-concentrated on Amicon Ultra-15 Centrifugal Filter Units (Merck) 50 kDa.

All recombinant protein was purified on a Strep-Tactin®XT 4Flow® high capacity column (IBA Lifesciences) with 5 ml resin bed. 5 ml of 10× Buffer W (100 mM Tris/HCl pH 8.0, 150 mM NaCl, 1 mM EDTA) was added to 45 ml of each pre-concentrated supernatant which then were loaded on the pre-equilibrated column in 4 °C protected from light, washed with 1X Buffer W and eluted Buffer BXT. All the fractions were collected, sterilised by filtration using Millex-GP 0.22 µM filter units (Merck) and concentrated down to 500 uL in sterile PBS on Amicon Ultra-15 Centrifugal Filter Units. Recombinant protein concentration

was measured using BCA Protein Assay Kit Following manufacturer's protocol.

### Recombinant protein uptake

LX2 cells were seeded in Cellview 4-compartment glass bottom imaging cell culture dish (Greiner), pre-treated with 100 nM calcipotriol and siRNA knockdown, as above, before loading overnight with 2 µg/mL of fluorescently tagged recombinant progranulin. Images were acquired using Zeiss LSM 800 Microscope (Carl Zeiss). Fluorescent intensities for individual cells were measured using ImageJ. For uptake in siRNA knockdown cells, only siGLO+ cells were measured.

### Proximity ligation assay

Proximity ligation assay (PLA) was performed using the Duolink PLA kit (Sigma) according to the manufacturer's guideline. Briefly, LX2 cells were seeded onto cover slips and left overnight to adhere. Stimulated LX2 cells were fixed in 4% paraformaldehyde and permeabilized with 0.025% Triton-X/PBS, followed by LIFR (Santa Cruz, sc-515337, 1:100) and Sortilin (Abcam, ab16640, 1:100) primary antibody incubation overnight at 4 °C, followed by incubation with PLA probes, and ligation and amplification steps according to the manufacturer's guideline. Imaging was performed using Zeiss LSM800 (Zeiss) confocal microscope. Image analysis of PLA puncta (greater than 10 pixels) was performed on ImageJ.

### Lipid droplet staining

LX2 cells were cultured on glass coverslips in the presence, or absence, of 100 nM calcipotriol for 24 h, fixed in 4% paraformaldehyde for 30 min followed by permeabilization with 0.1% Triton X-100 in PBS for 10 min. Cells were blocked with 1% BSA, 0.05% Tween-20 in PBS for 30 min at room temperature and stained with 1 µg/mL BODIPY (Thermo Fisher, D3922) for 20 min, counterstained with DAPI (Life Technologies) and mounted using Dako Fluorescent Mounting Medium. Cells were imaged using an Axio Observe Light Microscope with the Apotome.2 (Zeiss) and quantified using ImageJ software.

### Haematoxylin and Eosin staining

Paraffin embedded murine liver samples were dewaxed and hydrated using xylene followed by a graded ethanol series. Tissues were treated with haematoxylin (5 min), washed, followed by Eosin staining (1 min). After a final wash, the slides were rapidly dehydrated in in a graded ethanol series, cleared in xylene, and mounted.

### Picrosirius red staining

Paraffin embedded murine liver samples were de-waxed and hydrated using a graded ethanol series. Tissue sections were then treated with 0.2% phosphomolybdic acid and subsequently stained with 0.1% sirus red F3B (Direct red 80) (Sigma Aldrich) in saturated picric acid solution for 90 min at room temperature. Tissues were then rinsed twice in acidified water (0.5% glacial acetic acid) before and after the staining with 0.033% fast green FCF (Sigma Aldrich). Finally, tissues were dehydrated in three changes of 100% ethanol, cleared in xylene and mounted. Picrosirius red staining was imaged using an Axio Observe Light Microscope with the Apotome.2 (Zeiss) and quantified using ImageJ software.

### Immunoblotting

Protein lysates were prepared using 62.5 mM Tris-Cl pH 6.8; 10% glycerol; 2% SDS; 1% β-Mercaptoethanol supplemented with complete protease inhibitor mixture (Sigma), phosphatase inhibitor cocktail (Invitrogen), 1 mmol/L phenylmethysufonylfluoride and 0.2 mmol/L sodium orthovanadate. Cell lysates were incubated on ice for 10 min, before sonication and clarification. Protein concentration was determined using Pierce Protein BCA Assay Kit – Reducing Agent Compatible (Thermo Fisher) according to the manufacturer's guidelines.

Secreted proteins were concentrated using StrataClean Resin (Agilent Technologies) according to the manufacturer's guidelines. Proteins were separated on TGX Precast Gels (Bio-Rad) and transferred using Trans-blot Turbo Transfer System (Bio-Rad). Membranes were blocked in 5% BSA in Tris-buffered saline containing 0.1% Tween-20 (TBST), and incubated with primary antibodies, followed by HRP-conjugated secondary antibodies, diluted in 5% BSA-TBST. Antibodies used are listed in Supplementary Table 1. Protein bands were visualised using Pierce ECL Western Blotting Substrate (Thermo Fisher) on a ChemiDoc MP (Bio-Rad) imaging system.

### RT qPCR

Total RNA purification was performed with the RNeasy kit (Qiagen) and cDNA was generated using QuantiTect Reverse Transcription kit (Qiagen) according to the manufacturer's instructions. 500 ng of total RNA was used to generate cDNA. Quantitative polymerase chain reaction (qPCR) was performed using 5× HOT FIREPol EvaGreen qPCR Mix Plus (ROX) (Solis Biodyne) on an AriaMX instrument (Agilent). Three step amplification was performed (95 °C for 15 s, 60 °C for 20 s, 72 °C for 30 s) for 45 cycles. Relative expression levels were normalised to *Gapdh* expression according to the formula 2^- (Ct *gene of interest* – Ct *Gapdh*). Fold increase in expression levels were calculated by comparative Ct method 2^-(ddCt). Heatmaps were generated from standardised values of 2^(-ddCt). Statistical analysis was calculated between dCt values. Primer assays used are listed in Supplementary Table 2.

### Immunohistochemistry

Deparaffinization and antigen retrieval was carried out using the PT-Link System (DAKO), followed by immunostaining using Envision Plus System (DAKO). Tissue sections were incubated with primary antibodies at 4 °C overnight followed by incubation with HRP-conjugated polymer (Agilent). Antibodies are listed in Supplementary Table 1. Staining was developed using either diaminobenzidine (Agilent) or VIP (Vector) substrate and counterstained with haematoxylin (Sigma). Tissue sections were imaged using an Axio Observe Light Microscope with the Apotome.2 (Zeiss) and quantified using ImageJ software. A minimum of three fields of view, per section, was quantified and averaged for each data point representing a mouse.

### Immunofluorescence

Deparaffinisation and antigen retrieval was carried out using the PT-Link System (DAKO), followed by blocking in 10% of either donkey or goat serum. Tissue sections were incubated with primary antibodies at 4 °C overnight followed by incubation with fluorophore-conjugated secondary antibodies or tyramide signal amplification using Tyramide Superboost Kit (Thermo Fisher) and nuclear dye DAPI (Thermo Fisher) at room temperature for 2 h. Antibodies used are listed in Supplementary Table 1. Tissue sections were mounted using Fluorescent Mounting Media (DAKO). For staining using two antibodies raised from the same host species, tyramide signal amplification was performed using Tyramide Superboost Kit (Thermo Fisher) according to manufacturer's guidelines. For pSTAT3 (CST#9145, 1:100) staining, fresh frozen OCT embedded tissue was sectioned and pre-treated with methanol for 10 min at −20 °C before proceeding with immunofluorescence staining. Tissue sections were imaged using an Axio Observe Light Microscope with the Apotome.2 (Zeiss) and quantified using ImageJ and Zen Black software. From each mouse or patient, a minimum of three fields of view, per section, were quantified and averaged for each data point.

### Single-cell RNA sequencing sample preparation

Single-cell suspension from murine livers enriched for viable Epcam-, CD45-, CD31-, *Pdgfrb*-GFP+ cells were prepared by cell sorting as outlined above and immediately processed for library preparation using

10× Genomics Chromium Chip B Single Cell kit and Single Cell 3′ GEM, Library & Gel Bead kit (10× Genomics), according to manufacturer protocols. Paired-end sequencing was performed using the Illumina NextSeq 500 instrument.

### Alignment and quantification of counts

Raw FASTQ files were aligned to the mouse reference genome (mm10) using the CellRanger count pipeline[61]. UMI counts for aligned reads of each gene were quantified across individual cells and read into R (4.0.2) for downstream analysis using functionality within the Seurat package (V4)[62]. To exclude cell doublets or dead cells, cells with gene counts >6500, <500, or with over 10% of their reads mapping to mitochondrial genes were removed from the matrix.

### Sample integration, normalisation, and clustering

Filtered and quantified UMI counts for Naïve, IgG and αCSF1R samples were normalised and integrated with SCTransform, implementing a regularised negative binomial regression model with gene counts as a covariate to remove technical variance associated to read depth[63]. The integrated sample cell matrix was clustered using a shared nearest neighbours (SNN) graph constructed across the first 50 principal components of the dataset. The resolution for the Louvain clustering algorithm was set to 0.8. The goal of the sample integration was to assess the cellular heterogeneity between Naïve, IgG and aCSF1R samples and facilitate the identification of distinct metastasis-associated fibroblast subclusters. Clusters were visually and spatially represented using Uniform Manifold Approximation and Projections (UMAPs).

### Differential expression

Differentially expressed genes were determined across clusters using a one vs all approach with a Wilcoxon rank sum test implemented in the FindAllMarkers function in Seurat[62]. Genes were filtered for a minimum absolute log fold change of 0.25. Significantly differentially expressed genes were ranked in terms of absolute log fold change.

### Extraction of metastasis-associated fibroblasts

MAF cluster subpopulations were extracted from the integrated sample matrix of Naïve, IgG and aCSF1R based on the following two rules;
  1. Cells within a cluster originated from IgG or aCSF1R samples
  2. Cells for the cluster had <1% of cells from the Naïve samples

Extracted MAF cells were reclustered using a shared nearest neighbours (SNN) graph constructed across the first 20 principal components of the dataset with a resolution of 0.3. The clustering of extracted MAF cells resulted in a total of four subpopulations. The remaining Naïve cells were reclustered using a shared nearest neighbours (SNN) graph constructed across the first 50 principal components of the dataset with a resolution of 1. The clustering of extracted Naïve cells resulted in a total of 3 cell populations. Differentially expressed genes for MAF subpopulations and Naïve cell populations were determined using a Wilcoxon test in a one vs all approach using the FindAllMarkers function[62].

### Pathway enrichment analysis

The Gene Ontology (GO) pathway enrichment analysis of differentially expressed genes was performed using g:Profiler and REVIGO. To identify signalling pathways that are active within myMAFs, genes enriching within GO: Extracellular region was filtered out and the remaining cellular gene set list filtered against GO: Signalling. To identify genes that are likely secreted, myMAF DEG were filtered against GO: Extracellular region.

### Gene set enrichment analysis

Gene set enrichment analysis (GSEA) was used to iteratively test the enrichment of published myCAF and iCAF signatures against each of our identified clusters. GSEA was performed in R using the clusterProfiler package[64]. All P values were corrected using the Benjamini–Bochberg (BH) method[65].

### Quantification and statistical analysis

Statistical significance was analysed with GraphPad Prism v8 software. The Shapiro–Wilk test was used to determine the normality of the data. When comparing differences between two experimental groups, statistical significance was determined in normally distributed data using two-tailed unpaired Student's $t$ test, or two-tailed Mann–Whitney test for data that is not normally distributed. For multiple comparisons, one-way ANOVA coupled with Sidak's *post hoc*, or two-way ANOVA coupled with Tukey's *post hoc* test was performed. A $p$-value < 0.05 was considered statistically significant. Statistical details of specific experiments can be found in the corresponding figure legends. Where statistical significance is annotated on heatmaps, the exact $p$-values can be found in the source data.

### Statistics and reproducibility

Each experiment was repeated independently at least three times with similar results.

### Reporting summary

Further information on research design is available in the Nature Portfolio Reporting Summary linked to this article.

## Data availability

The publicly available scRNAseq data of MAFs derived from metastasis-bearing *Pdgfrb*-GFP mice generated in this study has been deposited in the Gene Expression Omnibus (GEO) database under accession code GSE232335. The remaining data are available within the Article, Supplementary Information, and Source Data. Source data are provided with this paper as a Source Data file. Any further information is available from the corresponding author upon request. Source data are provided with this paper.

## Code availability

Analysis for the manuscript was completed using an RStudio docker (rocker/tidyverse:4.0.2), and all code used is made available on GitHub [https://github.com/CBFLivUni/Raymant-et-al-2023].

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

## Acknowledgements

We thank the Liverpool Shared Research Facilities including the Centre for Cellular imaging (CCI), Centre for Genomic Research (CGR), Computational Biology Facility (CBF), Cell Sorting Facility (CSF), and the Biomedical Services Unit (BSU) for provision of equipment and technical assistance. We thank Elzbieta Boyd for the generation of reagents. We thank Ruth Stafferton for consenting of patients. We thank the patients and their families who contributed with tissue samples to these studies. We thank both Professor Scott Friedman for agreeing to share the LX2 cell line, and Professor Jelena Mann for donation of the LX2 cell line. These studies were supported by grants from Cancer Research UK (A25607, A26978, A26979), Medical Research Council (MR/P018920/1) and North West Cancer Research Fund for M.C.S, Wellcome Trust (102521/Z/13/Z) and North West Cancer Research Funds for A.M., Cancer Research UK A17196, A2996, and A25233 for J.P.M. N.C.H. is supported by a Wellcome Trust Senior Research Fellowship in Clinical Science (ref. 219542/Z/19/Z).

## Author contributions

M.R. designed and performed most of the experiments, analysed and interpreted the data, and wrote the manuscript. Y.A, L.A, V.Q, and G.B helped with in vivo experiments. V.C.G helped with in vitro experiments. M.G provided technical support. D.G performed bioinformatic analysis. D.P, C.H, and P.G provided patient samples. N.C.H provided *Pdgfrb*-GFP mice. J.P.M provided KPC-derived tissue samples. M.V provided *GFAP-Cre/ERT2;STAT3^{loxP/loxP}* mice and helped with in vivo experiments. A.M provided conceptual advice, interpreted the data and wrote the manuscript. M.C.S conceived and supervised the project, interpreted data, and wrote the manuscript. All authors critically analysed and approved the manuscript.

## Competing interests

The authors declare no competing interests.
