## [Peer Review File · Nature Communications]

Macrophage-fibroblast JAK/STAT dependent crosstalk promotes liver metastatic outgrowth in pancreatic cancerREVIEWER COMMENTS

Reviewer #1 (Remarks to the Author):

PDAC is a highly metastatic disease with dismal prognosis. In the recent years, studies exploiting scRNA seq techniques started to reveal the heterogeneous stromal microenvironment within primary PDAC tumors. However, when cancer cells metastasize to a secondary organ, the unique local environment will form a distinct cancer-associated stroma that results in cancer cell seeding and metastases formation. Therefore, understanding such metastatic niche is very important if we want to address the metastasis problem in PDAC. In the current study, by using scRNA seq, Raymant et al have identified the CAF heterogeneity within the liver metastasis and characterized an important interaction between macrophage and the myMAF subtypes. This article was well written and easy to follow. In addition, the studies were mostly well-designed and thorough. However, there are still some details that need to be addressed to further improve the quality of the study.

1. Four major populations of MAFs were described including vMAF, myMAF, iMAF, cycMAF. Are any of these clusters were actually pericytes, given some population seem to be closely associated with vasculature and PDGFRB is also a pericyte marker? In Figure 1G, markers for pericytes need to be shown such as CSPG4 and MCAM. If any of these populations turn out to be actually pericytes, I think instead of creating a new term for these populations, it is better to just call them pericytes and adjust the conclusions accordingly. This will be less confusing and more consistent with what has been known in the field about how pericyte regulate angiogenesis and thus affect metastasis. And therefore, besides staining for markers such as aSMA, PDGFRa etc in the metastasis, pericyte markers also need to be stained in the metastasis.

2. Many PDGFRA antibodies also cross-react with PDGFRB antibodies, therefore, the specificity of PDGFRA antibody used in this study needs to be validated. For example, compare PDGFRA staining pattern VS PDGFRB in the metastatic stroma see if they overlap mostly. Another reason to validate is that PDGFRA+ and aSMA+ populations seem to be two distinct MAF populations according to scRNA seq, however, in the staining, there are many PDGFRA+ aSMA+ double positive cells. This inconsistency needs to be addressed instead of simply calling these cells vMAF & iMAF (aSMA+PDGFRA-high cells). Alternatively, other

more MAF subtype markers should be used for staining if PDGFR α and α SMA are not good ones to differentiate vMAF and iMAF apart.

3. Figure 2F, is this a representative figure? Quality of staining is not good and the change after treatment is modest.

4. “we induced experimental metastasis in a genetic conditional mutant mouse model lacking STAT3 expression in HStCs, the cellular origin of myMAFs (cKO-STAT3: GFAP-Cre/ERT2;STAT3loxP/loxP)”, claiming a cell of origin will require robust lineage tracing study, since in this article such data was not provided, please change such claim. In addition, more information about GFAP needs to be provided. Evidence of why GFAP is a lineage marker of HStCs or myMAFs need to be provided.

5. Bone-marrow-derived macrophages need to be induced from Grn KO mouse bone marrow and look at their conditioned media effect on fibroblast STAT3 activation and myMAF phenotype induction in vitro.

6. Interaction between sortilin and LIFR needs to be investigated through immunoprecipitation.

Reviewer #2 (Remarks to the Author):

In this manuscript, the authors used the PDGFRB-GFP reporter mouse model and scRNAseq techniques to study the crosstalk between macrophages and metastasis-associated fibroblasts (MAFs) that promotes metastatic outgrowth of pancreatic cancer. They proposed a model in which macrophage-derived progranulin and cancer cell-secreted leukemia inhibitory factor (LIF) act together to promote the generation of pro-metastatic myofibroblastic-MAFs (myMAFs) through a STAT3-dependent mechanism. myMAF-specific genetic depletion of STAT3 or pharmacological blockade of STAT3 restored anti-tumor responses and inhibited tumor metastases. These findings are consistent with numerous earlier studies that have demonstrated a key functional role of STAT3 in cancer-associated fibroblasts (CAFs) or MAFs. In a reciprocal manner, the authors showed that myMAF-secreted osteopontin promoted the immunosuppressive properties of macrophages and the functional inhibition of cytotoxic CD8⁺ T cells. These studies enhanced our understanding of the complex interactions among macrophages, fibroblasts, and tumor cells in the tumor microenvironment, and support the ongoing efforts of targeting the JAK/STAT signaling

pathway in CAFs/MAFs for cancer treatment.

There are a few important issues that should be addressed as outlined below.

1) The conclusion that LIF secreted by cancer cells is important for promoting myMAFs is not justified by the data provided in this manuscript. Simply showing the high expression of LIF in cancer cells is not sufficient. The authors should provide experimental data that directly test the role of cancer-secreted LIF in the generating of myMAFs. Based on the authors' model, knockdown or knockout or inhibition of LIF in cancer cells would suppress the generation of myMAFs and repress tumor metastasis. Is this true? Have the authors tested cytokines in the LIF family such as IL-6? Does IL-6 play a role in the generation of myMAFs?

2) A key point of this manuscript is that progranulin induces the association of its receptor sortilin with the LIF receptor LIFR, resulting in the activation of STAT3. Progranulin does not bind to LIF or LIFR, how can the binding of progranulin to its receptor can cause the interaction of sortilin and LIFR? What is the molecular basis that can account for such interactions? Can progranulin promote the LIF-induced JAK1 phosphorylation?

3) Both myMAFs and cancer cells express high levels of Osteopontin (OPN) (Fig. 6). What is the role of OPN produced by cancer cells in the promoting of immunosuppressive macrophages in the tumor microenvironment?

Other points:

1) In Figure 3B, the left diagram, SOCS3 is known to directly inhibit JAK1, not STAT3. The diagram would make more sense by showing this fact.

2) In Figure 4K, the experiments should include a control of LIF treatment alone. The PGRN-mediated enhancement of STAT3 phosphorylation in response to LIF was not obvious (Fig. 4H), thus, it is important to show if PGRN has any effect on the LIF-induced gene activation.

Reviewer #3 (Remarks to the Author):

Metastasis is a primary cause of death from pancreatic cancer and it is extremely important to investigate how the cancer cells interact with the stromal cells in the liver and find ways to restrict or limit PDAC-derived liver metastasis. The paper "Macrophage-fibroblast JAK/STAT dependent crosstalk promotes pancreatic cancer liver metastasis" includes a mechanistic investigation of the interactions between cancer cells macrophages and fibroblasts. It described different types of metastasis-associated fibroblasts and claim a critical dependency of myMAF in the STAT3 pathway. STAT3 activation in myMAF is induced by cancer cells and macrophages, and it controls fibrosis, metastasis size, and the induction of an immunosuppressive program in macrophages. Overall, the paper is interesting but I have several concerns that should be addressed:

- 1) The author claims for new fibroblast subtype that is unique for metastasis, the vMAF. However, the split of c0+c1 cells to two clusters does not seem convincing. The authors should color the UMAP in Fig. 1g according to the genes that are listed in Fig. 1i (similar to Fig. 2a). It is important to observe the distribution of these genes without any scaling.
- 2) The discovery of vMAF is an important contribution to the manuscript. Continuing the previous concern, the author should use uniquely expressed genes in vMAF and stain human and mouse sections with relevant antibodies. Both myMAF and vMAF express aSMA and Pdgfra and these proteins can not efficiently be used to discriminate between the subtypes.
- 3) Many of the findings rely on tissue section staining. It is critical to give the reader detailed information on the quantification and on the number of sections that were quantified in each mouse. Also, the total metastatic area is a critical measure to evaluate the importance of the presented mechanism, adding X4 photos is important to evaluate the results. In addition, does STAT3 inhibitor increase mice's life span?
- 4) The authors invested a large amount of effort and performed a large set of experiments, but they need to emphasize which of the findings are new. For example, the role of fibrosis in metastasis formation is known and the role of STAT3 in myCAF biology is also known, the

same is the role of LIF and progranulin in myCAF activation.

5) It is not clear which of the interactions and effects between macrophages, cancer cells, and fibroblasts are specific for metastasis in comparison to the primary tumor site.

6) The authors claim for co-regulation of the JAK/STAT pathways by progranulin and LIF but the contribution of progranulin seems quite minor according to the data in Fig 4. Repeating the experiments in Fig 4j-k separately with LIF and progranulin can uncover the relative contributions. In addition, the level of IL6, a known inducer of STAT3 pathway, in PDAC patients' blood, is high. The actual contribution of LIF and progranulin to STAT3 activation is not clear. Does LIF and progranulin have an effect in IL6R KO background?

7) In the experiment shown in SUPPLEMENTARY FIGURE.S1 (KPC mice treated with AZD7507), is there a reduction in the metastatic area? It is important to show also in this model.

Reviewer #4 (Remarks to the Author):

Raymant et al investigate how the macrophage-fibroblast crosstalk promotes liver metastasis outgrowth in pancreatic ductal adenocarcinoma (PDAC). They identify 4 metastasis-associated fibroblast (MAF) populations: inflammatory, myofibroblastic (myMAFs), vascular-associated, and cycling. Mechanistically, the authors demonstrate that macrophages activate the JAK/STAT pathway in myMAFs, which in turn promote pancreatic metastatic outgrowth in the liver. Specifically, STAT3 activation in myMAFs is mediated by malignant cell-secreted LIF and boosted by macrophage-secreted progranulin. In turn, myMAF-secreted periostin appears to at least partially mediate myMAF pro-metastatic role by increasing malignant cell proliferation. Finally, osteopontin from both myMAFs and malignant cells induces an immunosuppressive phenotype in macrophages. Depletion of myMAFs, and consequential downregulation of osteopontin, appears to be at least partially responsible for increased infiltration and activity of cytotoxic T cells due to a loss of immunosuppressive macrophages.

Considering that most PDAC patients are metastatic at diagnosis, this study has the strong rationale of investigating late-stage PDAC for the development of new therapies. Moreover, the conclusions are largely supported by the data and the analyses shown are typically rigorous and well-controlled (e.g. single-cell RNA-seq results have been validated by IF; whenever possible autochthonous mouse model and human tissue analyses have been performed to corroborate the results obtained with the intrasplenic injection model; complementary genetic and pharmacological strategies are employed). Altogether this is a well-done study that provides novel insights into an important area. My comments are minor and meant to strengthen the study prior to publication.

1. In Figure S1K, the difference in CK19 staining in the autochthonous model is rather modest compared to the approximate 50% reduction in total metastatic area observed in the intrasplenic model (Figure S1H), even if the depletion of PDGFRbeta-positive cells and macrophages is comparable. Can the authors comment on this difference and/or provide total metastatic area for the autochthonous model and/or CK19 staining quantification for the intrasplenic model? Additionally, can the authors comment on whether the metastatic burden is comparable between these models?
2. Please indicate the sample number in all figure legends (e.g. number of mice for healthy liver and PDAC metastasis single-cell RNA-sequencing).
3. In the conclusions related to Figure 1, the authors state that the macrophages promote the presence of myMAFs. However, this has not been shown in figure 1, especially considering that the cancer cell burden is also reduced following macrophage depletion. I thus suggest to rephrase with “macrophage depletion leads to less myMAFs”, which is a faithful description of the data.
4. In addition to the heatmap shown in Figure S2I, it would be relevant to show whether the in vivo iCAF and myCAF signatures derived from KPC PDAC tumours (ref 12) are upregulated/downregulated/unchanged in the 4 MAF populations identified – they can be run as GSEA.
5. In Figure 4K, iMAF genes should also been shown. This is especially important considering that in this scenario PSCs would acquire an inflammatory CAF state.
6. Figures 6C/6D should also include results with control media + STAT3i to evaluate whether STAT3 inhibition has a direct effect on the macrophages, especially considering the

MHCII (H2Aa) expression data shown.

7. In the final summary, prior to the discussion, I suggest to include that Spp1 from the malignant cells may also contribute to the macrophage immunosuppressive phenotype. This is also suggested by the more dramatic CD8 infiltration and activation in STAT3i tumours compared to STAT3cko tumours (Figures 7J-K and 7E-F). I suggest the authors discuss this point as well.

8. In the discussion, when talking about primary PDAC, the authors say that iCAFs “originate from a pool of fibroblast, not tissue resident stellate cells” (ref 17). However, the study cited does mention that some PSCs become iCAFs, it just says that not all iCAFs come from PSCs. Thus, this sentence should be rephrased. Similarly, when discussing about JAK/STAT signalling in iCAFs in primary PDAC, the authors seem to suggest that this signalling has only been confirmed in iCAFs in vitro using PSCs (see “Whereas iCAF polarisation of PStCs has been shown to occur ... in vitro, the contribution of PStCs to the iCAF pool in vivo has recently been revealed to be minor”.) However, JAK/STAT signalling activation in iCAFs has also been confirmed in iCAFs in vivo, thus independently from the specific PSC origin (ref 18 and 12). Thus, this sentence should be rephrased to reflect previous evidence.

9. I strongly suggest changing the title with “...promotes liver metastatic outgrowth in pancreatic cancer” to more faithfully reflect the data shown and models used.

10. CAF heterogeneity has been shown to go beyond iCAFs, myCAFs, apCAFs. In the discussion, the authors mention that Lrrc15+ CAFs in PDAC also express Spp1. Can the authors show whether Lrrc15 is more highly expressed in myMAFs compared to other MAFs in their SCA datasets and/or in their sorted CAFs (with STAT3i)? Similarly, since Endoglin(Eng)-positive and -negative CAFs have also been characterised, and Eng-negative CAFs are associated with anti-tumour immunity (Hutton et al Cancer Cell), can the authors show expression of Eng in their SCA datasets and/or sorted CAFs (with STAT3i)?

11. As a small suggestion, I am not sure whether the authors should propose a combo of JAK/STAT inhibition and immunotherapy. My understanding is that JAK2 inhibition will impair INF-gamma production and that JAK1 inhibition will impair IL2 signalling, and thus T cell proliferation.

Giulia Biffi

Reviewer #5 (Remarks to the Author):

In the current paper, the authors addressed the interactions between macrophages, fibroblasts and cancer cells and the effects of this crosstalks on the metastatic potential of pancreatic cancer cells to colonize the liver. They identified the mechanisms of how macrophages regulates fibroblast heterogeneity in a metastatic niche, by giving rise to activation of metastasis-associated fibroblasts (MAFs). They reported that LIF (Leukemia inhibitory factor) from cancer cells and Progranulin from macrophages activates myMAFs (myofibroblastic-MAFs) through JAK/STAT signaling. Reciprocally, activated myMAFs also secrete osteopontin which promoting macrophage immunosuppression and inhibition of cytotoxic T cell functions.

Use of techniques such as proximity ligation assay and 3D cultures; or some applications in different mouse models like intrasplenic injection-induced metastasis, bone marrow transplant, macrophage depletion and in vivo inhibitor treatments put an emphasis on the novelty of findings in the current manuscript. Additionally involving data from human biopsy material also enriched paper's impact. Overall the experimental strategy was satisfactory to evaluate hypothesis.

Regarding to the contribution to the field, the reported outcomes can be useful for related future studies. To target identified molecular and cellular interactions in complex metastatic PDAC microenvironment can be promising to restrain PDAC liver metastasis.

The writing quality of the paper is clear, easy to follow and interpretable by target audience. The story is generally convincing. By considering the reported outcomes and the contribution to the field, the study is worthy of publication, but the authors must address the given commentaries and requirements to present a stronger impact in the complete study.

1) I found few typos in figure legends such as 3H, S2I paragraph startings, and in Figure 1 legends at 1E there are two dots as "experimental design.." and 1F "(lef)" is "left" should be corrected. I kindly ask the authors to proof-read whole manuscript carefully. Additionally, during the submission of the manuscript and figures to the system, figures were submitted twice and the order of Figure 2 and 3 is replaced in one of them. More, In Figure S6 legends, instead of A, B, C etc. for the subfigures, some dots appeared. Please check this typo exist in your original draft or it happened during the submission of the supplementary figures.

Please be aware of that.

- 2) What is the reason of use of *Pdgfrb* mice strain to label cells of mesenchymal lineage but not *Pdgfra*? Because *Pdgfra* strain was also many times reported to track mesenchymal cell populations.
- 3) Picrosirius red staining in the Figure S4K and M is not convincing by comparing other picrosirius staining images from literature or the other images shown in this manuscript like in figure S4A. Red stainings are barely visible difficult to compare with control. Please change these pictures with better versions.
- 4) Regarding JAK/STAT signaling, when the authors showed regulation of STAT3 phosphorylation with western blot, did they also check JAK expression levels? Can Progranulin or LIF also regulate JAK expression? If yes, please also involve them in the corresponding figures.
- 5) In Figure 3M, the number of α SMA⁺ cells are shown to be reduced by STAT3 inhibitor. It was also shown that α SMA⁺pSTAT3⁺ cell number also was reduced. Isn't it likely α SMA⁺pSTAT3⁺ cell number was reduced since the number of α SMA⁺ cells were reduced? What is the statistical significance between α SMA⁺ and α SMA⁺pSTAT3⁺ cell number (orange bars)?
- 6) Is there a specific reason to use CSF1R nAb to deplete macrophages in this study rather than other macrophage-depletion approaches?
- 7) Is there any severe phenotype of *Grn*^{-/-} mice?
- 8) Is there a better version of images in Figure S6B with a better resolution? It would be good to replace if exist. Because the colonies are not visible and distinguishable.
- 9) What could be the reason behind additive effect of PGRN to LIF regarding improving the phosphorylation of STAT3 in LX2 cells as it shown in Figures 4H and I? Was this experiment also performed for mouse system?
- 10) In Figure 5C and D, why is siRNA-mediated knockdown shown as “-/-” ? Because it is well known that “-/-” is always used for knock-out. Please correct this.
- 11) In Figure 6D, can authors explain why *Cxcl10* does not show similar expression pattern with *H2aa* in heatmap and give similar expression level like immunosuppressive genes although it is indicated as a immunostimulatory gene? Same also in Figure 6K, why the expression between immunostimulatory and immunosuppressive genes is not clear as in Fig 6I?
- 12) In Figure S6K, it is better to add a graph showing the quantification of WB data.

We thank everyone for their positive, fair, and constructive comments. As outlined in detail in the following pages, we have carefully addressed all the comments and provide additional experimental data.

REVIEWER COMMENTS

Reviewer #1 (Remarks to the Author):

PDAC is a highly metastatic disease with dismal prognosis. In the recent years, studies exploiting scRNA seq techniques started to reveal the heterogeneous stromal microenvironment within primary PDAC tumors. However, when cancer cells metastasize to a secondary organ, the unique local environment will form a distinct cancer-associated stroma that results in cancer cell seeding and metastases formation. Therefore, understanding such metastatic niche is very important if we want to address the metastasis problem in PDAC. In the current study, by using scRNA seq, Raymant et al have identified the CAF heterogeneity within the liver metastasis and characterized an important interaction between macrophage and the myMAF subtypes. This article was well written and easy to follow. In addition, the studies were mostly well-designed and thorough. However, there are still some details that need to be addressed to further improve the quality of the study.

1. Four major populations of MAFs were described including vMAF, myMAF, iMAF, cycMAF. Are any of these clusters were actually pericytes, given some population seem to be closely associated with vasculature and PDGFRB is also a pericyte marker? In Figure 1G, markers for pericytes need to be shown such as CSPG4 and MCAM. If any of these populations turn out to be actually pericytes, I think instead of creating a new term for these populations, it is better to just call them pericytes and adjust the conclusions accordingly. This will be less confusing and more consistent with what has been known in the field about how pericyte regulate angiogenesis and thus affect metastasis. And therefore, besides staining for markers such as aSMA, PDGFRa etc in the metastasis, pericyte markers also need to be stained in the metastasis.

Response: Thank you for this comment and we also initially wondered whether the vMAFs are pericytes due to their highly related functional annotations. We would like to explain that the four MAF populations are only present in tumour bearing livers, but not in healthy livers (Fig 1F). Thus, overall, the MAFs are distinct clusters of cells which are absent in healthy livers. However, as suggested by this reviewer we have now included additional scRNA seq data into this revised manuscript showing that cells expressing the classical pericyte markers Cspg4 and Mcam are captured by our Pdgfrb-GFP reporter mouse model (New Supplementary Figure S3B & S3C), but that none of these pericyte markers are expressed by the MAF clusters. The defined subcluster c0 (Fig. 1G) displays a functional signature for vascular-, tube, and blood vessel development (Fig. 1J) and we have therefore annotated this cluster as vascular-MAFs.

Additional IF staining's further confirming the presence of MCAM+ cells among Pdgfrb-GFP positive cells in healthy liver tissue surrounding blood vessel-like structures, while MCAM+ cells were only sparsely found within metastatic tumour lesions (New Supplementary Fig. S3E&F and highlighted text, lines 166-173).

Taken together, based on our single cell analysis and IF tissue section staining's, our data suggest that vMAFs are a distinct cluster of cells from classical Mcam+, Cspg4+ pericytes and that vMAFs are only present in tumour-bearing livers.

2. Many PDGFRA antibodies also cross-react with PDGFRB antibodies, therefore, the specificity of PDGFRA antibody used in this study needs to be validated. For example, compare PDGFRA staining pattern VS PDGFRB in the metastatic stroma see if they overlap mostly. Another reason to validate is that PDGFRA+ and aSMA+ populations seem to be two distinct MAF populations according to scRNA seq, however, in the staining, there are many PDGFRA+ aSMA+ double positive cells. This inconsistency needs to be addressed instead of simply calling these cells vMAF & iMAF (aSMA+PDGFRA-high cells). Alternatively, other more MAF subtype markers should be used for staining if PDGFRA and aSMA are not good ones to differentiate vMAF and iMAF apart.

Response: As this reviewer correctly raises, many PDGFRA and PDGFRB antibodies have been shown to cross-react. We can confirm that the PDGFRA antibody used herein, ab203491, has been knockout validated, and was chosen for this purpose. For clarification, we do not use a PDGFRB antibody since we are using our Pdgfrb-GFP+ reporter mouse model to identify PDGFRb+ positive cells.

Immunofluorescent characterisation shows that only a proportion of Pdgfrb-GFP+ cells do co-stain with the PDGFRA antibody ab203491 and that a proportion of GFP-negative PDGFRA-positive cells are also visible (New Supplementary Figure S1A-B). Together, these results support that the PDGFRA antibody we use in this study does not cross-react with PDGFRB.

Our single cell transcriptional analysis revealed four significantly distinct MAF clusters based on entire gene expression signatures of each cell. Based on their gene signatures, the data shows that three of the 4 MAF clusters originate from Lrat+ HStCs (vMAF, myMAF, cMAF), while one originates from CD34+ portal fibroblast (iMAF). The data also shows that there are no unique fibroblast markers that are completely absent or present among these four MAFs subclusters, but that MAFs show graded expression of fibroblast specific markers (Fig. 2A). Comparing gene expression levels among different MAF clusters using uMAPs and violin plots (Fig. 2A and New Supplementary Fig. S4A), our data suggests that Acta2 (α SMA) is expressed among all MAFs, but particularly high in myMAFs, while Pdgfra shows higher expression levels in vMAFs and iMAFs (New Supplementary Fig. S4A). Thus, in response to the reviewer, the PDGFRA+ aSMA+ double positive cells detected in tissue sections most likely represent cells at the interface of two clusters, and thus appear double positive. In the revised manuscript, we are now addressing this observation in the results (highlighted lines 226-239 & 241-243) and discussion (highlighted lines 656-667) sections and clearly state that CD34, PDGFRA, and aSMA do not represent unique MAF cluster markers, but are rather differentially enriched among the different MAF subtypes.

4. “we induced experimental metastasis in a genetic conditional mutant mouse model lacking STAT3 expression in HStCs, the cellular origin of myMAFs (cKO-STAT3: GFAP-Cre/ERT2;STAT3loxP/loxP)”, claiming a cell of origin will require robust lineage tracing study, since in this article such data was not provided, please change such claim. In addition, more information about GFAP needs to be provided. Evidence of why CFAP is a lineage marker of HStCs or myMAFs need to be provided.

Response: As suggested by this reviewer, the text has been modified to better reflect the data included in this manuscript (Highlighted lines 301-305). The revised text now reads “ Our prior results suggest that myMAFs are of HStC-lineage (Supplementary Fig. S3G). Gfap promoter-driven Cre

recombinase has previously been described as a model for selective deletion in HStCs (30-32). Therefore, to test the biological relevance of pSTAT3+myMAFs to metastatic outgrowth, we induced experimental metastasis in an inducible GFAP-STAT3 KO mouse model (cKO-STAT3: GFAP-Cre/ERT2;STAT3loxP/loxP) (Fig. 3F)."

Despite our best efforts, we have been unable to identify an antibody against GFAP that efficiently works on liver tissue and that is compatible with the specialised protocol needed for pSTAT3 detection (fresh frozen tissue and methanol fixation & permeabilization). Consequently, as an alternative approach to validating that GFAP-Cre efficiently targets hepatic stellate cells, we used Desmin, a defined marker for HStCs (1) to visualise the presence of pSTAT3+ cells among Desmin+ cells (Sup Fig. S5I&J), which confirmed efficient depletion of pSTAT3+Desmin+ cells.

5. Bone-marrow-derived macrophages need to be induced from Grn KO mouse bone marrow and look at their conditioned media effect on fibroblast STAT3 activation and myMAF phenotype induction in vitro.

Response: As proposed by this reviewer, we have performed additional studies exploring the contribution of conditioned media derived from bone-marrow-derived macrophages to the induction of a STAT3 active myMAF phenotype in vitro (New Fig 4Q&R). Our new data reveal that exposure of 3D cultured primary hepatic stellate cells to conditioned media (CM) from tumour-educated WT bone-marrow derived macrophages, mixed with KPC-tumour cell conditioned media, induces a gene expression signature that resembles STAT3-dependent myMAFs (Fig 4R). Notably, exposure of stellate cells to CM from tumour-educated Grn KO macrophages mixed with KPC tumour cell conditioned media failed to recapitulate the same phenotype observed with WT bone-marrow-derived macrophages. These additional studies have been incorporated into the revised manuscript in Fig.4Q&R and the main text (Lines 348-362).

6. Interaction between sortilin and LIFR needs to be investigated through immunoprecipitation.

Response: As suggested by this reviewer, we have investigated the interaction between sortilin and LIFR by immunoprecipitation. Unfortunately, despite several different approaches, we were unable to successfully co-IP sortilin and LIFR. Briefly, we attempted to IP sortilin from LX2 cells twice using two different detergents for cell lysis: Triton X-100 and NP-40. When lysed in Triton X-100, little to no sortilin was extracted in the whole cell lysate, while LIFR was robustly detectable. Unsurprisingly, we were unable to IP sortilin under these lysis conditions. In contrast, lysis of cells in NP-40 permitted robust extraction, detection, and immunoprecipitation of the sortilin receptor, but in these conditions, LIFR was not detectable in whole cell lysates nor the IP fraction. In agreement with our unsuccessful IP attempts, the interaction of sortilin and LIFR has previously been described based on proximity ligation assays (2). We were able to confirm this close proximity of sortilin/LIFR upon LIF and progranulin stimulation using the same method, but due to the nature of the proximity ligation assay, we cannot discern whether this is a direct or in-direct interaction. Thus, we have modified the text to reflect this by replacing the word interaction with proximity. These points are also raised in further detail in the discussion (highlighted lines 687-696).

Reviewer #2 (Remarks to the Author):

In this manuscript, the authors used the PDGFRB-GFP reporter mouse model and scRNAseq techniques to study the crosstalk between macrophages and metastasis-associated fibroblasts (MAFs) that promotes metastatic outgrowth of pancreatic cancer. They proposed a model in which macrophage-derived progranulin and cancer cell-secreted leukemia inhibitory factor (LIF) act together to promote the generation of pro-metastatic myfibroblastic-MAFs (myMAFs) through a STAT3-dependent mechanism. myMAF-specific genetic depletion of STAT3 or pharmacological blockade of STAT3 restored anti-tumor responses and inhibited tumor metastases. These findings are consistent with numerous earlier studies that have demonstrated a key functional role of STAT3 in cancer-associated fibroblasts (CAFs) or MAFs. In a reciprocal manner, the authors showed that myMAF-secreted osteopontin promoted the immunosuppressive properties of macrophages and the functional inhibition of cytotoxic CD8+ T cells. These studies enhanced our understanding of the complex interactions among macrophages, fibroblasts, and tumor cells in the tumor microenvironment, and support the ongoing efforts of targeting the JAK/STAT signaling pathway in CAFs/MAFs for cancer treatment.

There are a few important issues that should be addressed as outlined below.

1) The conclusion that LIF secreted by cancer cells is important for promoting myMAFs is not justified by the data provided in this manuscript. Simply showing the high expression of LIF in cancer cells is not sufficient. The authors should provide experimental data that directly test the role of cancer-secreted LIF in the generating of myMAFs. Based on the authors' model, knockdown or knockout or inhibition of LIF in cancer cells would suppress the generation of myMAFs and repress tumor metastasis. Is this true? Have the authors tested cytokines in the LIF family such as IL-6? Does IL-6 play a role in the generation of myMAFs?

Response: As suggested by this reviewer, we have now performed additional studies to address the contribution of cancer-cell secreted LIF to the generation of myMAFs and metastatic outgrowth, both in vitro and in vivo.

"The authors should provide experimental data that directly test the role of cancer-secreted LIF in the generating of myMAFs"

In vitro: Exposure of 3D culture primary hepatic stellate cells (HStCs) to tumour-educated macrophage-conditioned media (CM) together with cancer cell conditioned media recapitulated a myMAF phenotype in vitro (New Fig. 4H&I). In addition, cancer cell CM supplemented with recombinant progranulin was also sufficient to induce a myMAF phenotype (New Fig. 4J&K), while neutralisation of LIF from cancer cell-CM, supplemented with progranulin, was sufficient to partially disrupt the induction of a myMAF phenotype (New Fig. 5A&B). Together, these results support a role for cancer cell-derived LIF in contributing to a myMAF phenotype.

“Based on the authors’ model, knockdown or knockout or inhibition of LIF in cancer cells would suppress the generation of myMAFs and repress tumor metastasis. Is this true?”

As suggested by this reviewer, we have included additional studies evaluating the contribution of cancer-cell derived LIF to myMAF generation, and consequently, tumour metastasis. To determine whether cancer-cell derived LIF contributed to pro-metastatic functions of myMAFs, cancer cell CM was pre-treated with LIF neutralising antibody, or IgG, and supplemented with progranulin prior generating myMAFs. Conditioned media was harvested from myMAFs generated in the presence or absence of LIF, and exposed to cancer cells suspended in an agar layer to determine their effects on anchorage independent outgrowth (New Fig. 5C). Cancer cells exposed to IgG-myMAFs grew the largest colonies, while LIF neutralisation during the generation of myMAFs suppressed their cancer growth promoting functions (New Fig. 5D&E). Together, these studies provide evidence of the critical role for cancer cell-secreted LIF in the formation of myMAFs, which reciprocally promote anchorage-independent outgrowth of cancer cells.

To further validate the role of cancer-cell derived LIF in tumour metastasis, metastasis bearing Pdgfrb-GFP mice were treated with LIF neutralising antibody, or IgG (New Fig. 5L). LIF neutralisation repressed metastatic outgrowth, and coincided with significant reduction in fibrosis, pSTAT3+ α SMA+ cells, and collagen deposition, indicative of a repressed myMAF phenotype (New Fig. 5M-R and New Supplementary Fig. S6I&J).

Have the authors tested cytokines in the LIF family such as IL-6? Does IL-6 play a role in the generation of myMAFs?

As suggested by this reviewer, we have performed additional studies to address the role of IL6 in generating myMAFs. Cancer cells also expressed IL6, but at considerably lower levels in comparison to LIF (New Supplementary Fig. S6E). To test the contribution of IL6, we co-stimulated HStCs with progranulin and IL6 alone, or in combination. Adding IL-6 to progranulin stimulated HStCs further increased the expression of Spp1, but it failed to induce the expression of Cthrc1, Col1a1, Col1a2, and Postn, the markers used to define myMAFs (New Supplementary Figure. S6F). Thus, our data suggest that IL6 is not promoting myMAF formation, but it is plausible that IL6 contributes to the induction of Spp1 and the pro-tumourigenic functions of HStC, by inducing Spp1 expression in the presence of progranulin.

2) A key point of this manuscript is that progranulin induces the association of its receptor sortilin with the LIF receptor LIFR, resulting in the activation of STAT3. Progranulin does not bind to LIF or LIFR, how can the binding of progranulin to its receptor can cause the interaction of sortilin and LIFR? What is the molecular basis that can account for such interactions? Can progranulin promote the LIF-induced JAK1 phosphorylation?

Response: An interaction between sortilin and LIFR has previously been described, shown by the close proximity of both receptors within lipid rafts and the use of proximity ligation assays (1). In this study, Larsen et al. also state that they were unable to successfully conduct co-IPs of these receptors (1) and propose that the interaction between sortilin and LIFR causes conformational changes that increase LIFR’s affinity for cytokines.

To address this reviewers’ comment, we have now conducted a series of co-IPs of both receptors, but we were unsuccessful due to technical challenges (See comments related to reviewer 1, point 6).

However, using a proximity ligation assay, we were able to show that sortilin and LIFR are found in close proximity in fibroblasts in response to PGRN/LIF stimulation, but not under unstimulated conditions, suggesting an interaction of these two receptors (see Figure 6O&P). Due to the nature of the proximity ligation assay, we cannot discern whether this is a direct or in-direct interaction. Thus, we have modified the text to reflect this by replacing the word interaction with proximity. These points are also raised in further detail in the discussion (highlighted lines 687-696).

In regard of the molecular basis that can account for such interactions, the binding of progranulin to sortilin has been extensively characterised, and progranulin is shown to induce a unique conformational change to sortilin that stabilises the membrane-proximal 10CC-b domain (3). This conformational change is dependent on the C-term end of progranulin, which we observe to also be necessary for increased STAT3 activation (Fig. 6J-M). Thus, based on these observations we hypothesise that PGRN-induced conformational change of sortilin may transcend and stabilise receptors that are in close proximity of sortilin, such as LIFR, resulting in enhanced downstream signalling. However, proving this hypothesis will require further in-depth investigations that are beyond the scope of the current work. We have included this point as a future direction in our revised discussion (highlighted lines 687-696).

“Can progranulin promote the LIF-induced JAK1 phosphorylation?”

To address this question, in addition to enhanced pSTAT3 signalling, we have included new data showing that co-stimulation of LIF with progranulin also increases JAK1 phosphorylation at a 5-minute timepoint (New Supplementary Fig. S6O&P). Notably, in our model JAK1 phosphorylation is rapidly turned over and absent by 30 minutes.

3) Both myMAFs and cancer cells express high levels of Osteopontin (OPN) (Fig. 6). What is the role of OPN produced by cancer cells in the promoting of immunosuppressive macrophages in the tumor microenvironment?

Response: As suggested by this reviewer, we have included additional experiments to investigate the role of cancer cell-derived OPN in promoting an immunosuppressive macrophage phenotype (New Supplementary Fig. S7N-P). We found that indeed cancer cells also express Spp1 in a STAT3-dependent manner, and while cancer-cell CM potently induced an immunosuppressive macrophage phenotype, neutralisation of OPN was sufficient to partially restore Cxcl10 expression, while repressing induction of Arg1 in macrophages. These results suggest that cancer-cell derived OPN also plays a role in promoting immunosuppressive macrophages in the TME.

Other points:

1) In Figure 3B, the left diagram, SOCS3 is known to directly inhibit JAK1, not STAT3. The diagram would make more sense by showing this fact.

Response: As suggested by this reviewer, we have modified the diagram in the revised manuscript to accurately reflect the function of SOCS3 (revised Fig. 3B).

2) In Figure 4K, the experiments should include a control of LIF treatment alone. The PGRN-mediated enhancement of STAT3 phosphorylation in response to LIF was not obvious (Fig. 4H), thus, it is important to show if PGRN has any effect on the LIF-induced gene activation.

Response: *In response to this reviewer, we have performed additional studies delineating the relative contribution of LIF and progranulin to the observed myMAF signature (New Fig. 5F). While progranulin alone induces the expression of several myMAF genes, this occurs in a STAT3-independent manner (New Fig. 5F-I). LIF alone is also capable of inducing myMAF gene expression, while co-stimulation with progranulin further amplifies myMAF gene expression and suppresses a vMAF signature.*

Reviewer #3 (Remarks to the Author):

Metastasis is a primary cause of death from pancreatic cancer and it is extremely important to investigate how the cancer cells interact with the stromal cells in the liver and find ways to restrict or limit PDAC-derived liver metastasis. The paper "Macrophage-fibroblast JAK/STAT dependent crosstalk promotes pancreatic cancer liver metastasis" includes a mechanistic investigation of the interactions between cancer cells macrophages and fibroblasts. It described different types of metastasis-associated fibroblasts and claim a critical dependency of myMAF in the STAT3 pathway. STAT3 activation in myMAF is induced by cancer cells and macrophages, and it controls fibrosis, metastasis size, and the induction of an immunosuppressive program in macrophages. Overall, the paper is interesting but I have several concerns that should be addressed:

1) The author claims for new fibroblast subtype that is unique for metastasis, the vMAF. However, the split of c0+c1 cells to two clusters does not seem convincing. The authors should color the UMAP in Fig. 1g according to the genes that are listed in Fig. 1i (similar to Fig. 2a). It is important to observe the distribution of these genes without any scaling.

Response: *As suggested by this reviewer, we have included uMAP projections of the genes highlighted for each cluster in Fig. 11 to observe the distribution of these genes without any scaling (Revised Supplementary Fig. S3A, and copy provided below). Our additional analysis further*

supports the hypothesis that these are two distinct populations. In agreement with this, our reported GO analysis based on gene expression signature among c0 and c1 reveals distinct functional annotations (Fig 1J), further suggesting that c0 and c1 are distinct cell clusters.

Figure legend: UMAP displaying expression levels of selected genes in metastasis associated fibroblasts (MAFs). Upper row: Genes enriched in c0 (vMAF). Lower row: Genes enriched in c1 (myMAF)

2) The discovery of vMAF is an important contribution to the manuscript. Continuing the previous concern, the author should use uniquely expressed genes in vMAF and stain human and mouse sections with relevant antibodies. Both myMAF and vMAF express α SMA and Pdgfra and these proteins cannot efficiently be used to discriminate between the subtypes.

Response: In response to this comment, we would like to explain that our single cell transcriptional analysis shows that there are no unique markers that are completely absent or present within MAF clusters, but that identified and used markers are rather expressed as gradients and are enriched within one of the MAF clusters (Fig. 2A). In line with published guidance, we aimed to define common fibroblast markers that were enriched in each cluster and would not likely be present on other non-mesenchymal cells (4). Comparing the uMAPs and violin plots (Fig 2A and New Supplementary Fig. S4A), our data suggest that Acta2 (α SMA) is expressed among all MAFs, but is particularly high in myMAFs, while Pdgfra is mostly expressed in vMAFs and iMAFs. As a consequence, the cells at the interphase of two clusters will stain double positive for two markers as observed for α SMA+PDGFR α cells in our IF stainings. In the revised text, we now added additional details regarding the observed gradient expression of markers (highlighted lines 226-239 & 241-243) and also clearly state in the discussion that the here used single markers are enriched in the corresponding MAF clusters, but are not uniquely expressed, and that immunofluorescent staining was used in an attempt to visualise MAF subtypes identified by scRNAseq in tissue sections (highlighted lines 656-667).

Accordingly, our findings agree with other recent published data showing that the use of advanced technologies, such as scRNA seq or multiplexed mass cytometry, permits the identification of new MAF clusters based on their transcriptional phenotype or expression levels of a combination of markers (5, 6).

3) Many of the findings rely on tissue section staining. It is critical to give the reader detailed information on the quantification and on the number of sections that were quantified in each mouse. Also, the total metastatic area is a critical measure to evaluate the importance of the presented mechanism, adding X4 photos is important to evaluate the results. In addition, does STAT3 inhibitor increase mice's life span?

Response: As suggested by this reviewer, we have provided additional information in the figure legends & methods to include additional details of what has been quantified and the number of fields of view averaged per mouse.

As advised, for better evaluation of the results, the representative images have been replaced for x4 objective images for all Haematoxylin & Eosin staining throughout.

As suggested by this reviewer, we have performed an additional study evaluating the survival benefit of STAT3 inhibition in metastasis-bearing mice (New Fig. 3O). We found that STAT3 inhibition significantly increases the survival of metastasis-bearing mice, compared to vehicle control.

4) The authors invested a large amount of effort and performed a large set of experiments, but they need to emphasize which of the findings are new. For example, the role of fibrosis in metastasis formation is known and the role of STAT3 in myCAF biology is also known, the same is the role of LIF and progranulin in myCAF activation.

Response:

We would like to explain that in pancreatic cancer, cancer associated fibroblast subtypes have mainly been studied at the primary tumour site (5-9). Our work characterises the fibroblast landscape at the metastatic site in the liver. Our data reveals that myofibroblastic and inflammatory CAF subtypes are also present in metastatic PDAC livers, while the presence of vMAF has not been described at the primary tumour site in PDAC and is a new cluster that we have identified in the metastatic liver. While LIF/STAT3 signalling axis has been linked to iCAF activation at the primary tumour site, our data show that PRGN/LIF/STAT3 signalling promotes a myCAF phenotype at the metastatic site. Thus, our findings highlight that the cellular composition of fibroblasts differs between the primary and metastatic TME and that STAT3-mediated signalling leads to different fibroblast subtypes, that depend on the local TME and the cellular origin of the CAFs/MAFs. In our revised discussion, we are now emphasising these important new findings in the context of previous knowledge.

5) It is not clear which of the interactions and effects between macrophages, cancer cells, and fibroblasts are specific for metastasis in comparison to the primary tumor site.

Response: As described in the above paragraph, our study focuses on the metastatic liver TME in PDAC. Thus, all our findings relate to the secondary tumour site only. However, comparing our data to previous studies performed by other groups that have focused on the primary TME in PDAC, our findings suggest that there are significant differences in the cellular heterogeneity of fibroblasts at the primary and metastatic site. Moreover, at the primary site, STAT3-mediate activation of iCAFs (7), while in the liver, STAT3-promotes a myofibroblast phenotype and myMAFs promote metastatic outgrowth. These differences are most likely due to differences in cell origin and environmental cues provided by the organ specific TME.

6) The authors claim for co-regulation of the JAK/STAT pathways by progranulin and LIF but the contribution of progranulin seems quite minor according to the data in Fig 4. Repeating the experiments in Fig 4j-k separately with LIF and progranulin can uncover the relative contributions. In addition, the level of IL6, a known inducer of STAT3 pathway, in PDAC patients' blood, is high. The actual contribution of LIF and progranulin to STAT3 activation is not clear. Does LIF and progranulin have an effect in IL6R KO background?

Response: In response to this reviewer, we have performed additional studies delineating the relative contribution of LIF and progranulin to the observed myMAF signature (New Fig. 5F). While progranulin alone induces the expression of several myMAF genes, this occurs in a STAT3-independent manner (Fig. 5F-I). LIF alone is also capable of inducing myMAF gene expression, while co-stimulation with progranulin further amplifies the expression of myMAF genes and suppresses a vMAF signature.

To test the role of the IL6R and contribution of IL6, we knocked down *il6ra* using siRNA-mediated silencing prior exposure to LIF and progranulin. With the exception of *Col1a2* and *Postn*, the IL6R was dispensable for the induction of pSTAT3+ myMAF signature in this model (New Supplementary Figure S6G&H). To test the contribution of IL6, we next co-stimulated HStCs with progranulin and IL6 alone, or in combination. Adding IL-6 to progranulin stimulated HStCs further increased the expression of *Spp1*, but it failed to induce the expression of *Cthrc1*, *Col1a1*, *Col1a2*, and *Postn*, the markers used to define myMAFs (New Supplementary Figure. S6F). Thus, our data suggest that IL6 is not promoting myMAF formation, but it is plausible that IL6 derived from patients' blood contributes to the induction of *Spp1* and the pro-tumourigenic functions of HStC, by inducing *Spp1* expression in the presence of progranulin. We have now included these points in our revised discussion (Highlighted line 683-686).

7) In the experiment shown in SUPPLEMENTARY FIGURE.S1 (KPC mice treated with AZD7507), is there a reduction in the metastatic area? It is important to show also in this model.

Response: As suggested by this reviewer, we have included additional analysis of the total metastatic area in KPC mice with detectable metastases (New Supplementary Fig. S1J&K). Our additional analysis confirms significant reduction in metastatic burden and agree with the previously reported survival benefit for KPC mice treated with AZD7507 (10).

Reviewer #4 (Remarks to the Author):

Raymant et al investigate how the macrophage-fibroblast crosstalk promotes liver metastasis outgrowth in pancreatic ductal adenocarcinoma (PDAC). They identify 4 metastasis-associated fibroblast (MAF) populations: inflammatory, myofibroblastic (myMAFs), vascular-associated, and cycling. Mechanistically, the authors demonstrate that macrophages activate the JAK/STAT pathway in myMAFs, which in turn promote pancreatic metastatic outgrowth in the liver. Specifically, STAT3 activation in myMAFs is mediated by malignant cell-secreted LIF and boosted by macrophage-secreted progranulin. In turn, myMAF-secreted periostin appears to at least partially mediate myMAF pro-metastatic role by increasing malignant cell proliferation. Finally, osteopontin from both myMAFs and malignant cells induces an immunosuppressive phenotype in macrophages. Depletion of myMAFs, and consequential downregulation of osteopontin, appears to be at least partially responsible for increased infiltration and activity of cytotoxic T cells due to a loss of immunosuppressive macrophages.

Considering that most PDAC patients are metastatic at diagnosis, this study has the strong rationale of investigating late-stage PDAC for the development of new therapies. Moreover, the conclusions are

largely supported by the data and the analyses shown are typically rigorous and well-controlled (e.g. single-cell RNA-seq results have been validated by IF; whenever possible autochthonous mouse model and human tissue analyses have been performed to corroborate the results obtained with the intrasplenic injection model; complementary genetic and pharmacological strategies are employed). Altogether this is a well-done study that provides novel insights into an important area. My comments are minor and meant to strengthen the study prior to publication.

1. In Figure S1K, the difference in CK19 staining in the autochthonous model is rather modest compared to the approximate 50% reduction in total metastatic area observed in the intrasplenic model (Figure S1H), even if the depletion of PDGFRbeta-positive cells and macrophages is comparable. Can the authors comment on this difference and/or provide total metastatic area for the autochthonous model and/or CK19 staining quantification for the intrasplenic model? Additionally, can the authors comment on whether the metastatic burden is comparable between these models?

Response: As suggested by this reviewer, we have included additional analysis of total metastatic area for the autochthonous model, for faithful comparison to our experimental metastasis model. Our new data confirms a significant reduction in metastatic burden among mice treated with AZD7505, compared to the control group (New Supplementary Fig. S1J&K).

In response to whether metastatic burden is comparable, the intrasplenic model generates highly reproducible macro-metastatic tumours within 2-3 weeks, while naturally, the metastatic burden of KPC mice is unpredictable due to their spontaneously arising tumours. At end point, KPC mice tend to have fewer metastatic foci, but those lesions that are present have larger metastatic area, when directly compared to lesions in the intrasplenic model. In contrast, the intrasplenic model has markedly greater number of metastatic foci, which, on average, corresponds to similar overall metastatic burden. These differences in lesion sizes are most likely driven by the differing incubation time for metastatic outgrowth, with KPC mice being maintained until near humane end point whereas the intrasplenic model is maintained for 14 days in this study.

2. Please indicate the sample number in all figure legends (e.g. number of mice for healthy liver and PDAC metastasis single-cell RNA-sequencing).

Response: As suggested by this reviewer, we have included further information indicating sample numbers in all figure legends.

3. In the conclusions related to Figure 1, the authors state that the macrophages promote the presence of myMAFs. However, this has not been shown in figure 1, especially considering that the cancer cell burden is also reduced following macrophage depletion. I thus suggest to rephrase with “macrophage depletion leads to less myMAFs”, which is a faithful description of the data.

Response: As suggested by this reviewer, the text has been modified to better reflect the description of the data from figure 1. The text now reads: “Taken together, our data reveals four distinct MAFs populations in PDAC liver metastatic tumours, which originate either from HStC (myMAF, cycMAF, vMAF) or portal fibroblasts (iMAF), and that macrophage depletion leads to less myMAFs”. See highlighted text, line 216-218).

4. In addition to the heatmap shown in Figure S2I, it would be relevant to show whether the in vivo iCAF and myCAF signatures derived from KPC PDAC tumours (ref 12) are upregulated/downregulated/unchanged in the 4 MAF populations identified – they can be run as GSEA.

Response: As proposed by this reviewer, we have now performed the additional analysis by GSEA of iCAF and myCAF signatures derived from KPC PDAC tumours (New Sup Fig. S3H). These additional data reveal that myMAFs and iMAFs significantly enrich for myCAF and iCAF gene signatures, respectively, while vMAFs enrich for neither. cycMAFs also did not significantly enrich for either signature, however it is likely this is driven by the small number of cells present in the cluster. Together, these results further support our observation that myMAF/myCAF and iMAF/iCAF phenotypes are conserved across tumour sites, whereas vMAFs are unique to the metastatic microenvironment.

5. In Figure 4K, iMAF genes should also been shown. This is especially important considering that in this scenario PSCs would acquire an inflammatory CAF state.

Response: In response to this comment, we would like to explain that our single cell transcriptomic data strongly suggests that iMAFs in mouse PDAC liver metastasis originate from portal fibroblast due to their expression of Ly6a (Supplementary FigS3K), while vMAF and myMAF are mainly from HStC origin based on Lrat expression. Since in our in vitro assays we are using HStCs, we haven't tried to recapitulate an iMAF phenotype and only compared vMAF versus myMAF.

6. Figures 6C/6D should also include results with control media + STAT3i to evaluate whether STAT3 inhibition has a direct effect on the macrophages, especially considering the MHCII (H2Aa) expression data shown.

Response: As suggested by this reviewer, we have included additional controls to evaluate the direct effect of STAT3 inhibition on macrophages (New Fig. 7E). While there is a trend towards reduced H2aa expression, these changes are not significant. On the other hand, STAT3i stimulates Cxcl10 expression and suppresses both Arg1 and Mrc1, indicative of reprogramming macrophages towards a more immunostimulatory phenotype. For increased clarity, we have also projected the results of statistical tests on heatmaps throughout the manuscript.

7. In the final summary, prior to the discussion, I suggest to include that Spp1 from the malignant cells may also contribute to the macrophage immunosuppressive phenotype. This is also suggested by the more dramatic CD8 infiltration and activation in STAT3i tumours compared to STAT3cko tumours (Figures 7J-K and 7E-F). I suggest the authors discuss this point as well.

Response: As suggested by this reviewer, we have included further discussion highlighting the role of cancer cell-derived Spp1 in contributing to an immunosuppressive macrophage phenotype, and

how this may contribute to increased CD8 infiltration and activation in STAT3i-tumours vs STAT3cKO tumours (highlighted lines 726-731).

In addition, in response to reviewer 2, comment 3, we have performed additional studies to explore these observations in vitro, revealing that neutralisation of cancer cell-derived OPN partially suppresses an immunosuppressive macrophage phenotype (New Supplementary Fig. S7N-P).

8. In the discussion, when talking about primary PDAC, the authors say that iCAFs “originate from a pool of fibroblast, not tissue resident stellate cells” (ref 17). However, the study cited does mention that some PSCs become iCAFs, it just says that not all iCAFs come from PSCs. Thus, this sentence should be rephrased. Similarly, when discussing about JAK/STAT signalling in iCAFs in primary PDAC, the authors seem to suggest that this signalling has only been confirmed in iCAFs in vitro using PSCs (see “Whereas iCAF polarisation of PStCs has been shown to occur ... in vitro, the contribution of PStCs to the iCAF pool in vivo has recently been revealed to be minor”.) However, JAK/STAT signalling activation in iCAFs has also been confirmed in iCAFs in vivo, thus independently from the specific PSC origin (ref 18 and 12). Thus, this sentence should be rephrased to reflect previous evidence.

Response: As suggested by this reviewer, we have rephrased the wording in the text to correctly reflect previously published evidence.

The revised text now reads:

“Our findings are further supported by recent evidence from the primary PDAC site indicating that pancreatic iCAFs, similar to what we observed with iMAFs, can also arise from a pool of resident fibroblasts, (17)” (Highlighted line 648-650).

“The JAK/STAT pathway has previously been associated with driving a tumour-promoting iCAF phenotype at the primary site of PDAC, and in vitro studies have identified autocrine-LIF and paracrine-OSM as critical mediators of this phenotype (18, 56)” (Highlighted line 697-699).

9. I strongly suggest changing the title with “...promotes liver metastatic outgrowth in pancreatic cancer” to more faithfully reflect the data shown and models used.

Response: In agreement with this reviewer, we have altered the title to better reflect the data and models used. Our updated title is: Macrophage-fibroblast JAK/STAT dependent crosstalk promotes liver metastatic outgrowth in pancreatic cancer

10. CAF heterogeneity has been shown to go beyond iCAFs, myCAFs, apCAFs. In the discussion, the authors mention that Lrrc15+ CAFs in PDAC also express Spp1. Can the authors show whether Lrrc15 is more highly expressed in myMAFs compared to other MAFs in their SCA datasets and/or in their sorted CAFs (with STAT3i)? Similarly, since Endoglin(Eng)-positive and -negative CAFs have also been characterised, and Eng-negative CAFs are associated with anti-tumour immunity (Hutton et al Cancer Cell), can the authors show expression of Eng in their SCA datasets and/or sorted CAFs (with STAT3i)?

Response: As suggested by the reviewer, we have included further analysis from our scRNA seq to visualise Lrrc15 and Eng expression across MAF subpopulations (New Sup Fig. S3I&J). While Lrrc15 was undetectable in our dataset, Eng is expressed across all four MAF subpopulations, with some

potential enrichment in vMAF and iMAFs. These results provide further evidence of discrepancies and similarities across primary and metastatic site, highlighting organ-specific MAF heterogeneity.

11. As a small suggestion, I am not sure whether the authors should propose a combo of JAK/STAT inhibition and immunotherapy. My understanding is that JAK2 inhibition will impair INF-gamma production and that JAK1 inhibition will impair IL2 signalling, and thus T cell proliferation.

Response: As suggested by this reviewer, we removed the proposal of combining JAK/STAT inhibition and immunotherapy and further discuss the potential impact of JAK2 inhibition on T cells (Highlighted lines 747-752).

Reviewer #5 (Remarks to the Author):

In the current paper, the authors addressed the interactions between macrophages, fibroblasts and cancer cells and the effects of this crosstalks on the metastatic potential of pancreatic cancer cells to colonize the liver. They identified the mechanisms of how macrophages regulates fibroblast heterogeneity in a metastatic niche, by giving rise to activation of metastasis-associated fibroblasts (MAFs). They reported that LIF (Leukemia inhibitory factor) from cancer cells and Progranulin from macrophages activates myMAFs (myofibroblastic-MAFs) through JAK/STAT signaling. Reciprocally, activated myMAFs also secrete osteopontin which promoting macrophage immunosuppression and inhibition of cytotoxic T cell functions.

Use of techniques such as proximity ligation assay and 3D cultures; or some applications in different mouse models like intrasplenic injection-induced metastasis, bone marrow transplant, macrophage depletion and in vivo inhibitor treatments put an emphasis on the novelty of findings in the current manuscript. Additionally involving data from human biopsy material also enriched paper's impact. Overall the experimental strategy was satisfactory to evaluate hypothesis.

Regarding to the contribution to the field, the reported outcomes can be useful for related future studies. To target identified molecular and cellular interactions in complex metastatic PDAC microenvironment can be promising to restrain PDAC liver metastasis.

The writing quality of the paper is clear, easy to follow and interpretable by target audience. The story is generally convincing. By considering the reported outcomes and the contribution to the field, the study is worthy of publication, but the authors must address the given commentaries and requirements to present a stronger impact in the complete study.

1) I found few typos in figure legends such as 3H, S2I paragraph startings, and in Figure 1 legends at 1E there are two dots as "experimental design.." and 1F "(lef)" is "left" should be corrected. I kindly ask the authors to proof-read whole manuscript carefully. Additionally, during the submission of the manuscript and figures to the system, figures were submitted twice and the order of Figure 2 and 3 is replaced in one of them. More, In Figure S6 legends, instead of A, B, C etc. for the subfigures, some dots appeared. Please check this typo exist in your original draft or it happened during the submission of the supplementary figures. Please be aware of that.

Response: As correctly identified by this reviewer, some typos have passed through undetected and errors in the order of the figures has appeared in one of the versions uploaded. We thank this reviewer for highlighting these issues and have rectified the errors identified, and further typos that we detected throughout. Regarding Figure S6 legends, we thank this reviewer for highlighting these issues that emerged during PDF conversions of the documents. These have also been addressed.

2) What is the reason of use of Pdgfrb mice strain to label cells of mesenchymal lineage but not Pdgfra? Because Pdgfra strain was also many times reported to track mesenchymal cell populations.

Response: The Pdgfrb-eGFP mouse strain was selected to label cells of mesenchymal lineage due to prior publication of the extensive characterisation of the hepatic mesenchyme in this mouse model using scRNA seq technologies (11). Accordingly, we utilised this available dataset as quality control for our own analysis (Sup Fig. S2D). While Pdgfra has also been reported to track mesenchymal cell

populations, it has also been used to identify sub-populations of CAFs in different organs (6). With this in mind, we opted for the *Pdgfrb* reporter mouse model to study metastasis associated fibroblasts in the liver.

3) Picrosirius red staining in the Figure S4K and M is not convincing by comparing other picrosirius staining images from literature or the other images shown in this manuscript like in figure S4A. Red stainings are barely visible difficult to compare with control. Please change these pictures with better versions.

Response: As suggested by this reviewer, the representative picrosirius red stainings have been replaced with higher quality images for better interpretation of the red collagen staining (revised Sup Fig. S5K and M).

4) Regarding JAK/STAT signaling, when the authors showed regulation of STAT3 phosphorylation with western blot, did they also check JAK expression levels? Can Progranulin or LIF also regulate JAK expression? If yes, please also involve them in the corresponding figures.

Response: As suggested by this reviewer, we have included additional data showing that the addition of progranulin to LIF induces a modest increase in JAK1 phosphorylation (New Supplementary Fig, S6O&P). Notably, in our model JAK1 phosphorylation is rapidly turned over and absent by 30 minutes. The membranes that are already populated in the figures are at the 30-minute timepoint, and accordingly, we were unable to detect JAK1 phosphorylation for these.

5) In Figure 3M, the number of α SMA+ cells are shown to be reduced by STAT3 inhibitor. It was also shown that α SMA+pSTAT3+ cell number also was reduced. Isn't it likely α SMA+pSTAT3+ cell number was reduced since the number of α SMA+ cells were reduced? What is the statistical significance between α SMA+ and α SMA+pSTAT3+ cell number (orange bars)?

Response: In response to this comment, we would like to clarify that the α SMA+pSTAT3+ cell number reported in the graph was quantified as a percentage change of pSTAT3+ α SMA+ cells, among α SMA+ cells, but this information was absent from the graph (as originally reported in Fig. 3E and Fig. 4G). We have improved the labelling on the figure and in the legend, while the raw analysis is accessible within the source data.

6) Is there a specific reason to use CSF1R nAb to deplete macrophages in this study rather than other macrophage-depletion approaches?

Response: As highlighted by this reviewer, there are numerous approaches available to deplete macrophages. We have prior experience with several techniques, including utilisation of PI(3)K $\gamma^{-/-}$ mice, chlodronate liposomes, and targeting both CSF1 and CSF1R with neutralising antibodies (12-14). Since CSF1R targeting strategies are more widely tested in the clinical setting, we utilised this approach for the highest potential for translation.

7) Is there any severe phenotype of Grn^{-/-} mice?

Response: The Grn^{-/-} mice have been extensively evaluated by others and aged mice (>13 months) are reported to have decreased overall survival, social engagement, learning, and memory deficits (15). These symptoms are reported to reflect key aspect of frontotemporal dementia (FTLD), for which the Grn^{-/-} mice are used as a model. Meanwhile, young mice (<12 months old) do not display discernible symptoms of FTLD. This reference has now been included in the manuscript. However, for our studies mice aged 2-3 months were used which do not show any severe phenotype at this age.

8) Is there a better version of images in Figure S6B with a better resolution? It would be good to replace if exist. Because the colonies are not visible and distinguishable.

Response: The data and representative images in this figure were captured on the PerkinElmer IVIS imaging spectrum for the bioluminescence detection of cancer cells. As this imaging system was designed for in vivo imaging, unfortunately there is not enough resolution to distinguish individual colonies that are residing in different planes within the wells. Detection of bioluminescence offers a sensitive and unbiased measure of colony burden within each treatment condition.

9) What could be the reason behind additive effect of PGRN to LIF regarding improving the phosphorylation of STAT3 in LX2 cells as it shown in Figures 4H and I? Was this experiment also performed for mouse system?

Response: An interaction between sortilin and LIFR has previously been described by Larsen et al Furthermore, it has been shown that the C-term of PGRN induces a conformational change in sortilin (3).

Based on our data, we propose a model that the additive effect of PGRN to LIF is because PGRN stabilises the LIF Receptor (LIFR), resulting in prolonged and sustained activation of STAT3. The increased stability of LIFR is most likely mediated by the involvement of sortilin, a receptor for PGRN, which interacts with LIFR as previously shown (1).

While these points are speculative, we believe deeper insights into the molecular mechanism are beyond the scope of the current work, but is something we are planning to follow up on in future projects. We have included this point as a future direction in the discussion.

With regards to performing the experiment in the mouse system, we would like to explain that recovering primary HStCs from the matrigel takes over 1 hour, and thus it is not possible to monitor the early signalling events we observed using the LX2 cell line. However, we included additional studies at the 4-day timepoint to explore pSTAT3 activity which confirms that the co-stimulation of primary murine HStCs with LIF and progranulin results in sustained activation of pSTAT3 and high α SMA expression, consistent with a pSTAT3+myMAF phenotype (New Fig. 5D-F). This phenotype is not recapitulated under separate stimulation with LIF or progranulin alone.

10) In Figure 5C and D, why is siRNA-mediated knockdown shown as “-/-” ? Because it is well known that “-/-” is always used for knock-out. Please correct this.

Response: As highlighted by this reviewer, we have incorrectly labelled siRNA-mediated knockdown as -/-. This mislabelling has been rectified throughout with a labelling that says “siRNA KD”

11) In Figure 6D, can authors explain why Cxcl10 does not show similar expression pattern with H2aa in heatmap and give similar expression level like immunosuppressive genes although it is indicated as an immunostimulatory gene? Same also in Figure 6K, why the expression between immunostimulatory and immunosuppressive genes is not clear as in Fig 6I?

Response: In response to this comment, we would like to explain that our data suggests that a myMAF-derived factor induces the expression of Cxcl10 in a STAT3-dependent manner. In the simplest experimental approach, whereby macrophages are stimulated with recombinant OPN only, macrophages conform to the classical immunostimulatory and immunosuppressive dichotomy, with downregulation of both H2aa and Cxcl10, but upregulation of Arg1, Chi3l3, and Mrc1 (was Figure 6I, now revised Fig. 7J). Our data suggests that an additional unidentified myMAF derived factor, that is expressed in a STAT3-dependent manner, has additional immunoregulatory role of inducing Cxcl10 expression in macrophages. Our data supports that this factor is not osteopontin (was Figure 6K, now revised Fig. 7L).

These results offer an exciting hypothesis: although STAT3-active myMAFs are overwhelmingly pro-tumourigenic and suppress anti-tumour immunity, there is evidence here that a myMAF-derived factor is also capable of inducing immunostimulatory macrophage functions. It is plausible that some functions of myMAFs are tumour-restraining, and identifying any divergent functions is critical for developing methods that can selectively target pro-tumourigenic functions, while harnessing tumour-restraining mechanisms. We have included these points as future direction in the discussion so that attention is drawn to the regulation of Cxcl10.

12) In Figure S6K, it is better to add a graph showing the quantification of WB data.

Response: As suggested by this reviewer, we have now included densitometric analysis of the WB data (revised Sup Fig. S8K-M).

References rebuttal letter

1. L. Shang, M. Hosseini, X. Liu, T. Kisseleva, D. A. Brenner, Human hepatic stellate cell isolation and characterization. *J Gastroenterol* **53**, 6-17 (2018).
2. J. V. Larsen *et al.*, Sortilin facilitates signaling of ciliary neurotrophic factor and related helical type 1 cytokines targeting the gp130/leukemia inhibitory factor receptor beta heterodimer. *Mol Cell Biol* **30**, 4175-4187 (2010).
3. E. Trabjerg *et al.*, Investigating the Conformational Response of the Sortilin Receptor upon Binding Endogenous Peptide- and Protein Ligands by HDX-MS. *Structure* **27**, 1103-1113 e1103 (2019).

4. E. Sahai *et al.*, A framework for advancing our understanding of cancer-associated fibroblasts. *Nat Rev Cancer* **20**, 174-186 (2020).
5. E. Elyada *et al.*, Cross-Species Single-Cell Analysis of Pancreatic Ductal Adenocarcinoma Reveals Antigen-Presenting Cancer-Associated Fibroblasts. *Cancer Discov* **9**, 1102-1123 (2019).
6. C. Hutton *et al.*, Single-cell analysis defines a pancreatic fibroblast lineage that supports anti-tumor immunity. *Cancer Cell* **39**, 1227-1244 e1220 (2021).
7. G. Biffi *et al.*, IL1-Induced JAK/STAT Signaling Is Antagonized by TGFbeta to Shape CAF Heterogeneity in Pancreatic Ductal Adenocarcinoma. *Cancer Discov* **9**, 282-301 (2019).
8. C. X. Dominguez *et al.*, Single-Cell RNA Sequencing Reveals Stromal Evolution into LRRC15(+) Myofibroblasts as a Determinant of Patient Response to Cancer Immunotherapy. *Cancer Discov* **10**, 232-253 (2020).
9. A. T. Krishnamurty *et al.*, LRRC15(+) myofibroblasts dictate the stromal setpoint to suppress tumour immunity. *Nature* **611**, 148-154 (2022).
10. J. B. Candido *et al.*, CSF1R(+) Macrophages Sustain Pancreatic Tumor Growth through T Cell Suppression and Maintenance of Key Gene Programs that Define the Squamous Subtype. *Cell Rep* **23**, 1448-1460 (2018).
11. R. Dobie *et al.*, Single-Cell Transcriptomics Uncovers Zonation of Function in the Mesenchyme during Liver Fibrosis. *Cell Rep* **29**, 1832-1847 e1838 (2019).
12. S. R. Nielsen *et al.*, Macrophage-secreted granulin supports pancreatic cancer metastasis by inducing liver fibrosis. *Nat Cell Biol* **18**, 549-560 (2016).
13. V. Quaranta *et al.*, Macrophage-Derived Granulin Drives Resistance to Immune Checkpoint Inhibition in Metastatic Pancreatic Cancer. *Cancer Res* **78**, 4253-4269 (2018).
14. G. Bellomo *et al.*, Chemotherapy-induced infiltration of neutrophils promotes pancreatic cancer metastasis via Gas6/AXL signalling axis. *Gut* **71**, 2284-2299 (2022).
15. N. Ghoshal, J. T. Dearborn, D. F. Wozniak, N. J. Cairns, Core features of frontotemporal dementia recapitulated in progranulin knockout mice. *Neurobiol Dis* **45**, 395-408 (2012).

REVIEWERS' COMMENTS

Reviewer #1 (Remarks to the Author):

I appreciate the effort the authors spent on addressing my comments. My previous concerns have been addressed and the article has been greatly improved.

Reviewer #2 (Remarks to the Author):

I am satisfied with the revised manuscript that has addressed my previous concerns.

Reviewer #3 (Remarks to the Author):

The authors responded to all the points that I raised and made a significant effort to answer all the concerns, but it is still not clear if they claim that vMAF can be detected ONLY in metastasis and not in primary tumors. It is important to show if vMAF can be detected in other data sets. Macrophages and fibroblasts are plastic cells and scRNA-seq data analysis results in a "cloud" of cells that can then be divided in different ways. Combinations of markers can be chosen for each subclassification but it happens that the resulting sub-classification is not robust. Looking at the answer to my first two concerns I am still not sure that this is not the case.

Reviewer #4 (Remarks to the Author):

The authors have sufficiently addressed my comments and original concerns.

Reviewer #5 (Remarks to the Author):

All of my points have been adequately addressed by the authors, and I have no additional comments.

Rebuttal letter

REVIEWERS' COMMENTS

Reviewer #1 (Remarks to the Author):

I appreciate the effort the authors spent on addressing my comments. My previous concerns have been addressed and the article has been greatly improved.

Response: We would like to thank this reviewer for reviewing our manuscript.

Reviewer #2 (Remarks to the Author):

I am satisfied with the revised manuscript that has addressed my previous concerns.

Response: We would like to thank this reviewer for reviewing our manuscript.

Reviewer #3 (Remarks to the Author):

The authors responded to all the points that I raised and made a significant effort to answer all the concerns, but it is still not clear if they claim that vMAF can be detected ONLY in metastasis and not in primary tumors. It is important to show if vMAF can be detected in other data sets. Macrophages and fibroblasts are plastic cells and scRNA-seq data analysis results in a "cloud" of cells that can then be divided in different ways. Combinations of markers can be chosen for each subclassification but it happens that the resulting sub-classification is not robust. Looking at the answer to my first two concerns I am still not sure that this is not the case.

Response: In response to this reviewer's comment, we have now revised our text in the result section referring to vMAFs. We now emphasise the vMAFs are detectable in PDAC tumour bearing livers, but not in tumour free livers (line 174-175). In addition, we now include in the discussion that further studies are required to define whether vMAFs are uniquely present in PDAC liver metastasis or whether vMAFs can also be detected in other organs, or cancer types (line 647-650).

Reviewer #4 (Remarks to the Author):

The authors have sufficiently addressed my comments and original concerns.

Response: We would like to thank this reviewer for reviewing our manuscript.

Reviewer #5 (Remarks to the Author):

All of my points have been adequately addressed by the authors, and I have no additional comments.

Response: We would like to thank this reviewer for reviewing our manuscript.